# Primary and secondary clarithromycin resistance in *Helicobacter pylori* and mathematical modeling of the role of macrolides

Éva Kocsmár [1], György Miklós Buzás[2], Ildikó Szirtes[1], Ildikó Kocsmár [1], Zsófia Kramer[1], Attila Szijártó[3], Petra Fadgyas-Freyler[4], Kató Szénás[5], Massimo Rugge[6,7], Matteo Fassan [6], András Kiss[1], Zsuzsa Schaff[1], Gergely Röst [8,9,10] & Gábor Lotz [1,10 ✉]

Clarithromycin is a macrolide antibiotic widely used for eradication of *Helicobacter pylori* infection, and thus resistance to this antibiotic is a major cause of treatment failure. Here, we present the results of a retrospective observational study of clarithromycin resistance (Cla-res) in 4744 *H. pylori*-infected patients from Central Hungary. We use immunohistochemistry and fluorescence in situ hybridization on fixed gastric tissue samples to determine *H. pylori* infection and to infer Cla-res status, respectively. We correlate this information with macrolide dispensing data for the same patients (available through a prescription database) and develop a mathematical model of the population dynamics of Cla-res *H. pylori* infections. Cla-res is found in 5.5% of macrolide-naive patients (primary Cla-res), with no significant sex difference. The model predicts that this primary Cla-res originates from transmission of resistant bacteria in 98.7% of cases, and derives from spontaneous mutations in the other 1.3%. We find an age-dependent preponderance of female patients among secondary (macrolide-exposed) clarithromycin-resistant infections, predominantly associated with prior use of macrolides for non-eradication purposes. Our results shed light into the sources of primary resistant cases, and indicate that the growth rate of Cla-res prevalence would likely decrease if macrolides were no longer used for purposes other than *H. pylori* eradication.

[1] 2nd Department of Pathology, Semmelweis University, Budapest, Hungary. [2] Department of Gastroenterology, Ferencváros Health Center, Budapest, Hungary. [3] 1st Department of Surgery and Interventional Gastroenterology, Semmelweis University, Budapest, Hungary. [4] National Health Insurance Fund Administration, Budapest, Hungary. [5] Department of Pathology, Péterfy Hospital, Budapest, Hungary. [6] Department of Medicine (DIMED), Surgical Pathology & Cytopathology Unit, University of Padua, Padua, Italy. [7] Veneto Tumor Registry (RTV), Veneto Regional Authority, Padua, Italy. [8] Wolfson Center for Mathematical Biology, University of Oxford, Oxford, United Kingdom. [9] Bolyai Institute, University of Szeged, Szeged, Hungary. [10]These authors jointly supervised this work: Gergely Röst, Gábor Lotz. ✉email: lotz.gabor@med.semmelweis-univ.hu

*H*elicobacter pylori (H. pylori) is one of the most successful human pathogens and is estimated to be present in more than 50% of the global human population[1,2]. Although *H. pylori* causes only an asymptomatic infection in the vast majority of cases[3], it nonetheless a topic of interest in several fields of medicine, including gastroenterology, oncology and microbiology, chiefly due to its potentially severe sequalae, such as the development of chronic atrophic gastritis, peptic ulcer disease, gastric adenocarcinomas and B-cell mucosa-associated lymphoid tissue (MALT) lymphomas[2]. H. pylori infection usually persists for life and can be cleared only by medical eradication treatment[3]. The increasing failure rate of eradication treatment due to the appearance of resistant H. pylori strains contributes to the worldwide prevalence of this infection and subsequent inflammatory and neoplastic disorders[2].

Clarithromycin, a member of the macrolide family, is still the most potent antibiotic used in H. pylori eradication treatment protocols[4]. Clarithromycin binds to the 50S bacterial ribosomal subunit of H. pylori, and the consequent inhibition of protein synthesis results in a bacteriostatic effect[4]. Clarithromycin resistance (Cla-res) fundamentally depends on specific point mutations of certain codons in the peptidyl transferase region of the 23S rRNA, which are able to negatively influence the affinity of clarithromycin for the bacterial ribosome[4]. Moreover, there is an efflux pump mechanism, but it is able to effectively excrete clarithromycin only in 23S rRNA mutant strains, suggesting that it is not an independent resistance system but acts synergistically with mutational resistance[4,5]. Albeit the horizontal gene transfer of mutant genetic sequences is also an existing mechanism contributing to the high genetic diversity of H. pylori, this mechanism can only facilitate the spread of clarithromycin resistance within the H. pylori population of an infected person (consisting of one or multiple H. pylori strains) but, given the current state of our knowledge, it cannot transmit resistance between different individuals[4,6–8].

The increasing prevalence of Cla-res strains significantly contributes to the decreasing success rates of H. pylori eradication[2]. The efficacy of the conventionally used empirical triple therapy (a combination of clarithromycin, amoxicillin (or metronidazole), and an oral proton pump inhibitor (PPI), administered without susceptibility testing) is being dramatically reduced by the increase in Cla-res[9–11]. Therefore, Cla-res H. pylori ranks as the most problematic community-acquired non-mycobacterial infection and is listed as a high priority for research on the World Health Organization (WHO) priority list of antibiotic-resistant bacteria[12]. Additionally, the international guidelines for the management of H. pylori infection state that when the Cla-res rate is equal to or more than 15% of all diagnosed H. pylori infections in a population, eradication treatment with a clarithromycin-containing protocol is not recommended without prior susceptibility testing[2,13].

Previous studies have confirmed the existence of cross-resistance among all macrolides in H. pylori in vitro, as well as a causal link between Cla-res H. pylori infections and consumption of macrolide antibiotics[14–16]. Based on this, the notion of "no macrolide, no clarithromycin resistance" has become common among microbiologists and gastroenterologists[17]. In gastroenterology, primary Cla-res is commonly defined as resistance in patients with no prior history of eradication treatment with a clarithromycin-containing regimen. However, the use of clarithromycin or other macrolides for other purposes (including treatment of upper and lower respiratory tract infections, pelvic inflammatory disease, skin, and soft tissue infections)[18] may also lead to Cla-res[19].

Heteroresistant infections are characterized by mixed populations of susceptible and resistant H. pylori bacteria and can be associated with macrolide use for non-eradication purposes[20]. Nevertheless, the exact role of heteroresistant infections and non-eradication-motivated macrolide therapies in maintaining the population level of Cla-res is largely unclear. However, epidemiological modeling can shed light on the role of these factors in the population dynamics of Cla-res H. pylori[21]. To the best of our knowledge, the clarithromycin resistance of H. pylori has not been mathematically modeled thus far[22]. Although an analytical framework was created by Rupnow et al. to model H. pylori population dynamics, they did not include antibiotic resistance among the investigated parameters[23].

In this work, our aim was to determine the true primary Cla-res rate in a large cohort by analyzing the prior macrolide consumption of each patient based on a national medicine-dispensing database. Herein, we also establish the role of macrolides in high Cla-res rates, with special emphasis on the importance of macrolide use for non-eradication purposes. However, we also provide evidence for the existence of resistance that is not caused by macrolide consumption, and we analyze the population dynamics of Cla-res H. pylori infections by mathematical modeling to reveal the source of these primary resistance cases and the effect of non-eradication-motivated macrolide use on the prevalence of H. pylori and Cla-res.

## Results

**Epidemiology of clarithromycin resistance in Central Hungary.** Fluorescence in situ hybridization (FISH)-based clarithromycin susceptibility testing (Fig. 1) of 4744 H. pylori-infected patients identified a 17.2% overall prevalence of Cla-res in the studied period. Females showed a significantly higher Cla-res rate than males (19.8% vs 13.7%, $p < 0.001$). Heteroresistant infection was detected in nearly half of the resistant cases (47.2%). No significant sex difference in the heteroresistance rate was observed (males: 49.1%; females: 46.2%; $p = 0.46$) (Table 1). Age and sex significantly influenced the presence of resistance in the multivariate model ($p < 0.0001$, $p < 0.0001$, respectively), whereas the proportion of heteroresistant infections in resistant cases was found to be independent of age and sex ($p = 0.918$, $p = 0.429$). Regarding bacterial density, the resistance rate was significantly lower in 1+ cases than in 2+ or 3+ cases ($p < 0.001$ for both). Heteroresistance was also significantly less frequent (40.3%) among 1+ cases than in patients with 2+ bacterial density (50%) ($p = 0.0452$), whereas no significant difference was found between 1+ and 3+ (47.1%) or between 2+ and 3+ ($p = 0.189$ and $p = 0.478$, respectively) (Table 1).

Analysis of resistance data by age group revealed a resistance rate lower than 15% only in patients aged 70+ years (Table 1). In adult women, the prevalence of Cla-res was over 20% in all age groups except patients aged 70 years and over. In contrast, resistance rates in men were under 15%, except in those aged 30–39 years (Table 1; Fig. 2). Univariable analysis found significant differences between the female and male resistance rates in the 40–49, 50–59, and 60–69 age groups (20–29: $p = 0.184$; 30-39: $p = 0.146$; 40-49: $p = 0.0005$, RR 0.682–0.880, OR 0.329–0.753; 50–59: $p < 0.0001$, RR 0.683–0.846, OR 0.342–0.673; 60–69: $p = 0.0101$, RR 0.733-0.947, OR 0.453–0.910; $p = 0.9229$; Fisher's exact test).

**Primary and secondary resistance defined by prior Cla eradication.** According to the common definition of primary Cla-res in gastroenterology, the proportion of resistant patients with no prior history of clarithromycin-containing eradication treatment was 13.3% (Table 2). Resistance mutations were found in 51.4% of previously treated patients as secondary (acquired) resistance.

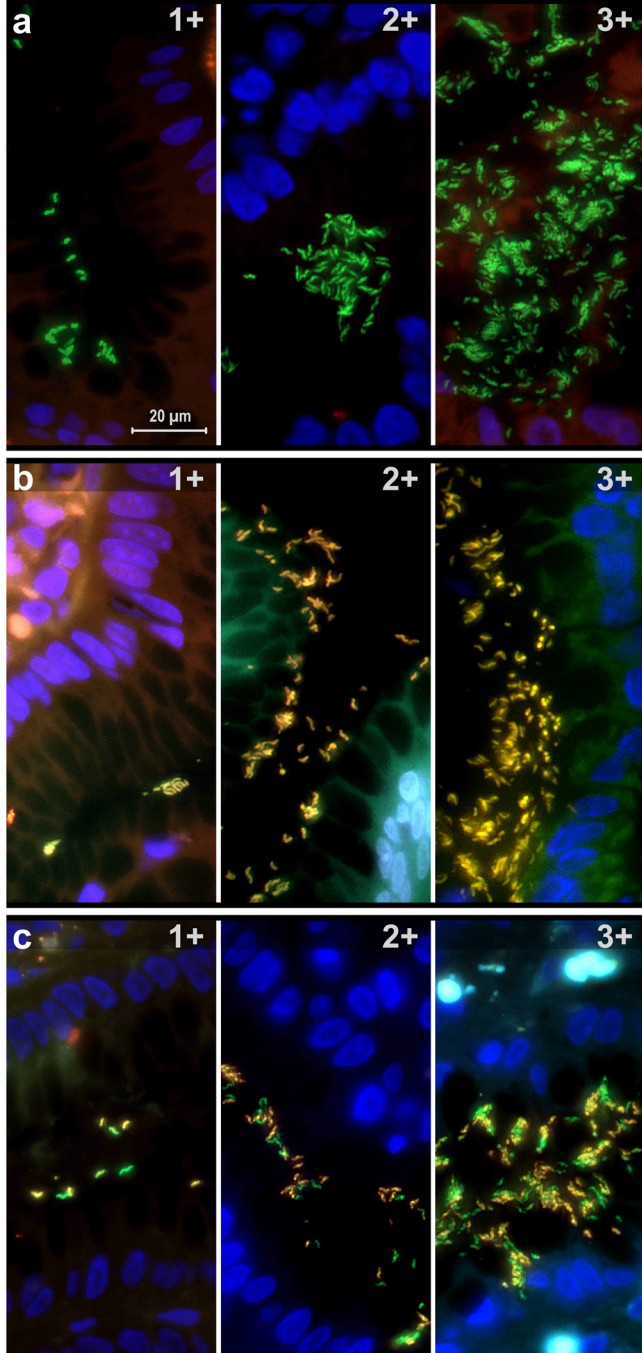

**Fig. 1 Detection of *Helicobacter pylori* bacteria with fluorescence in situ hybridization (FISH) in gastric tissue samples. a** Clarithromycin-susceptible bacteria appear in green. **b** In clarithromycin-homoresistant infection, resistant *H. pylori* bacteria show yellow fluorescence. **c** In clarithromycin-heteroresistant *H. pylori* infection, a mixed population of susceptible (green) and resistant (yellow) bacteria is present. 1+/2+/3+: low/moderate/high *H. pylori* density. Original magnification: 1000×. For further details, see "Clarithromycin susceptibility FISH test" in the Methods.

**Primary and secondary resistance defined by any prior macrolide use**. Compared to the above approach, a definition of primary Cla-res that excluded any prior macrolide consumption found a significantly reduced percentage, as only 5.5% of macrolide-naive patients showed resistance ($p < 0.001$) (Table 2). However, 30.6% of the macrolide-exposed group exhibited secondary resistance (Table 2).

**Analysis of primary and secondary resistance rates**. A significant sex difference was found in eradication-naive resistant cases: 15.1% of females with non-eradicated infection exhibited resistance, compared with only 11.1% of males ($p = 0.0001$) (Table 2). In contrast, no significant difference was found between macrolide-naive females and males when primary Cla-res rates were assessed by considering all prior macrolide consumption (Table 2). However, in the macrolide-exposed group, women showed a significantly higher prevalence of Cla-res than men (Table 2), with higher average numbers of macrolide-dispensing occasions and boxes dispensed per person (3.18 vs 2.61 and 3.46 vs 2.91 in females vs males, respectively). For a more detailed comparison of the primary and secondary Cla-res rates (including the homo-/heteroresistance rates) in women and men, see Supplementary Note 1, Supplementary Table S1. A significant difference was observed between rates of eradication-induced and other macrolide-treatment-related secondary resistance (Table 2). Regarding the latter, clarithromycin therapy for purposes other than *H. pylori* eradication was found to lead to Cla-res more frequently than the use of other macrolides (Table 2). The difference between secondary resistance rates observed after eradication and non-eradication use of clarithromycin was statistically significant (51.4% vs 36.2%, respectively; $p < 0.001$).

**Mathematical modeling of the transmission dynamics of *H. pylori***. Based on the numerical results predicted by the model from the input parameters (Table 3, with detailed explanation in Supplementary Notes 1 and 2) and displayed in Supplementary Note 2, Supplementary Table S7, transmission of resistant strains is responsible for only approximately 5.7% (4.67–7.43%) of all new infections (incidence, including all transitions from uninfected to infected compartments, whether infected with susceptible or resistant strains). Assuming that a heteroresistant infected individual exposes the infectee to both resistant and susceptible *H. pylori* with equal probability, the relative transmission fitness of the resistant *H. pylori* is $\delta = 0.72$ (0.58–0.95) relative to the susceptible bacteria. By applying the results of the model to the 5.5% primary resistant infection observed in our cohort, we predict the following sources: 98.7% of macrolide-naive Cla-res infections are derived from transmission of a resistant infection, including one-third in which the resistant strain overcolonizes a wild-type bacterial population (resulting in heteroresistant infection), and 1.3% are developed by spontaneous mutation (Fig. 3).

We have also modeled the effect of discontinuing macrolide use for indications other than *H. pylori* eradication. With the current rate of macrolide consumption, we predict an increase of approximately 0.1% (0.06–0.14%) per year in the rate of clarithromycin resistance in the future; however, the discontinuation of non-eradication macrolide use would keep the rate of clarithromycin resistance in the population on a very slowly increasing trajectory (Fig. 4).

**Discussion**

We conducted one of the largest studies on Cla-res *H. pylori*, involving 816 resistant cases out of 4744 *H. pylori*-infected patients. Uniquely, we investigated the primary Cla-res rate in the most accurate way available, by collecting data on individual patients' prior macrolide use from a nationwide medicine-dispensing database. We established a mathematical model to characterize the transmission dynamics of Cla-res *H. pylori* and reveal the origin of primary Cla-res infections.

Primary Cla-res of *H. pylori* infections is commonly defined as resistance in patients with no prior history of clarithromycin-

**Table 1 Prevalence of clarithromycin-resistant _H. pylori_ infections by sex, bacterial load, and age.**

|  | n | Cla-susceptible | Cla-resistant | Cla-res/total | Homoresistant | Heteroresistant | Homo-/heteroresistant |
|---|---|---|---|---|---|---|---|
| Total | 4744 | 3928 | 816 | 17.2% | 431 | 385 | 53:47 |
| Females | 2709 | 2172 | 537 | 19.8% | 289 | 248 | 54:46 |
| Males | 2035 | 1756 | 279 | 13.7% | 142 | 137 | 51:49 |
| Hp 1+ | 1196 | 1042 | 154 | 12.9% | 92 | 62 | 60:40 |
| Hp 2+ | 2100 | 1712 | 388 | 18.5% | 194 | 194 | 50:50 |
| Hp 3+ | 1448 | 1174 | 274 | 18.9% | 145 | 129 | 53:47 |
| Age groups (years) |  |  |  |  |  |  |  |
| 20–29 | 252 | 208 | 44 | 17.5% | 19 | 25 | 43:57 |
| 30–39 | 536 | 438 | 98 | 18.3% | 52 | 46 | 53:47 |
| 40–49 | 740 | 596 | 144 | 19.5% | 80 | 64 | 56:44 |
| 50–59 | 1199 | 991 | 208 | 17.3% | 114 | 94 | 55:45 |
| 60–69 | 986 | 807 | 179 | 18.2% | 95 | 84 | 53:47 |
| ≥70 | 979 | 850 | 129 | 13.2% | 63 | 66 | 49:51 |
| Female age groups (years) |  |  |  |  |  |  |  |
| 20–29 | 136 | 108 | 28 | 20.6% | 12 | 16 | 43:57 |
| 30–39 | 280 | 222 | 58 | 20.7% | 31 | 27 | 53:47 |
| 40–49 | 429 | 327 | 102 | 23.8% | 58 | 44 | 57:43 |
| 50–59 | 686 | 538 | 148 | 21.6% | 87 | 61 | 59:41 |
| 60–69 | 542 | 428 | 114 | 21.0% | 58 | 56 | 51:49 |
| ≥70 | 602 | 522 | 80 | 13.3% | 38 | 42 | 48:52 |
| Male age groups (years) |  |  |  |  |  |  |  |
| 20–29 | 116 | 100 | 16 | 13.8% | 7 | 9 | 44:56 |
| 30–39 | 256 | 216 | 40 | 15.6% | 21 | 19 | 53:47 |
| 40–49 | 311 | 269 | 42 | 13.5% | 22 | 20 | 52:48 |
| 50–59 | 513 | 453 | 60 | 11.7% | 27 | 33 | 45:55 |
| 60–69 | 444 | 379 | 65 | 14.6% | 37 | 28 | 57:43 |
| ≥70 | 377 | 328 | 49 | 13.0% | 25 | 24 | 51:49 |

_Cla_ clarithromycin, _res_ resistant.

containing eradication treatment. However, this approach has several limitations. Clarithromycin is used to treat various other bacterial infective diseases, which may lead to Cla-res of _H. pylori_, as may the use of other macrolides[16,19]. This bystander selection (unintended selection of resistant _H. pylori_ by an antibiotic therapy targeting another bacterial species) has been suggested to play an important role in the maintenance of the population level of Cla-res prevalence of _H. pylori_, including both primary and secondary resistance according to the above definition[24]. Nevertheless, most of the previously published studies on the epidemiology of Cla-res have used this definition of primary resistance because the prior antibiotic consumption of the patients for non-eradication purposes is frequently unknown. Some authors have assessed macrolide consumption at the population level by using antibiotic sales data[16,19], which is not appropriate for revealing acquired Cla-res in individual cases. Others asked patients to list prior antibiotic treatments or reviewed their medical records where available from treating hospitals, outpatient clinics, and family doctors[16,25,26]. However, human memory is prone to substantial gaps and errors[27]. Moreover, antibiotic treatment data from individual treatment sites do not necessarily cover all possible sources of antibiotics; therefore, these inaccurate approaches commonly ignore the use of macrolides for non-eradication purposes.

We found an overall _H. pylori_ Cla-res prevalence of 17.2% in our study (Table 2). Based on the common definition (no prior eradication attempt with a clarithromycin-containing regimen), the primary Cla-res rate reached 13.3% with a significant sex difference. The prevalence of Cla-res was significantly lower (5.5%) in patients with no history of prior macrolide use than in those with such a history, and there was no statistically significant difference between sexes. A possible role of macrolide exposure from macrolide remnants in food and water was excluded, as no macrolide traces (over the threshold) were detected in alimentary

products in the study period (personal communication with Dr. Melinda Nemes-Terényi from the Hungarian National Food Chain Safety Office, 2018). Moreover, food processing, especially heat treatment, mostly inactivates these antibiotics[28]. Therefore, we considered these macrolide-naive but Cla-res patients to be the real primary resistance cases.

However, the question arises as to how this primary resistance develops. Some patients may acquire a Cla-res _H. pylori_ infection (transmitted resistant strain)[29]. On the other hand, _H. pylori_ is known to have a high mutation rate[30–32]; therefore, Cla-res mutations can occur even in the absence of macrolide-related selection pressure (internal source, newly emerged resistant strain). Previous in vitro and experimental studies have determined the frequency of spontaneous mutation to Cla-res in _H. pylori_ isolates[32,33]. On this basis, we calculated that Cla-res bacteria continuously appear in chronic _H. pylori_ infections (Supplementary Note 4), but the rate of successfully emerging Cla-res clones is presumably low in the absence of clarithromycin-induced selection pressure. However, we found no published data on the prevalence of this spontaneous-mutation-induced primary resistance in the human population.

Thus, we established a mathematical model describing the population dynamics of Cla-res _H. pylori_. Among the macrolide-naive Cla-res patients, this model predicts the ratio of transmitted to mutation-originated resistant infections by considering several factors, e.g., the published mutation rate of _H. pylori_, the observed homoresistance/heteroresistance ratio (as a spontaneous mutation is expected to lead to heteroresistance, whereas transmitted resistant infections will more likely result in homoresistant bacterial populations), the assessed prevalence of _H. pylori_ and the observed Cla-res rate (affecting the transmitted resistance rate). By modeling the population dynamics of Cla-res _H. pylori_ with these observed, assessed and previously published parameters, we

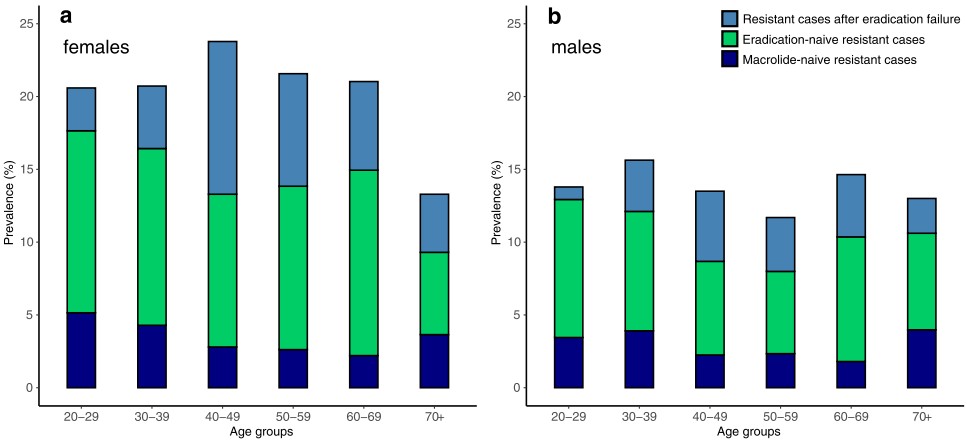

**Fig. 2 Prevalence of *Helicobacter pylori* clarithromycin resistance in adults by age group (relative to the total cohort). a** Cla-resistance was diagnosed in 530 cases out of the 2675 adult female patients. Treatment-related subcohorts (resistant cases after eradication failure/eradication-naive resistant cases/ macrolide-naive resistant cases) according to age groups: 20–29 years: 4/17/7, 30–39 years: 12/34/12, 40–49 years: 45/45/12, 50–59 years: 53/77/18, 60–69 years: 33/69/12, 70+ years: 24/34/22. **b** Cla-resistance was diagnosed in 272 cases out of the 2017 adult male patients. Treatment-related subcohorts (resistant cases after eradication failure/eradication-naive resistant cases/macrolide-naive resistant cases) according to age groups: 20–29 years: 1/11/4, 30–39 years: 9/21/10, 40–49 years: 15/20/7, 50–59 years: 19/29/12, 60–69 years: 19/38/8, 70+ years: 9/25/15. Deep blue: macrolide-naive resistant, green: clarithromycin-containing eradication-naive (but other macrolide treatment exposed) resistant, light blue: resistant with a history of *Helicobacter pylori* eradication with a clarithromycin-containing treatment regimen. Cla: clarithromycin.

**Table 2 Clarithromycin resistance rates of *H. pylori*-infected individuals according to prior eradication therapy and use of macrolides.**

|  | F/M | Cla-resistant | | Cla-susc | Total | p | OR/RR | CI (OR/RR) |
|---|---|---|---|---|---|---|---|---|
| Cla eradication history |  |  |  |  |  |  |  |  |
| Cla eradication attempt | 1.64 | 51.4% | 247 | 234 | 481 | <0.001 | 6.849 | 5.583–8.405 |
| Cla-eradication-naive | 1.30 | 13.3% | 569 | 3694 | 4263 |  | 3.847 | 3.427–4.320 |
| Total | 1.33 | 17.2% | 816 | 3928 | 4744 |  |  |  |
| Cla-eradication-naive |  |  |  |  |  |  |  |  |
| Females |  | 15.1% | 364 | 2046 | 2410 | <0.001 | 1.430 | 1.187–1.726 |
| Males |  | 11.1% | 205 | 1648 | 1853 |  | 1.365 | 1.163–1.602 |
| Total | 1.30 | 13.3% | 569 | 3694 | 4263 |  |  |  |
| History of macrolide use |  |  |  |  |  |  |  |  |
| Exposed to any macrolide | 1.71 | 30.6% | 676 | 1536 | 2212 | <0.001 | 7.516 | 6.184–9.185 |
| Macrolide-naive | 1.08 | 5.5% | 140 | 2392 | 2532 |  | 5.527 | 4.650–6.570 |
| Total | 1.33 | 17.2% | 816 | 3928 | 4744 |  |  |  |
| Macrolide-naive |  |  |  |  |  |  |  |  |
| Females |  | 6.4% | 84 | 1229 | 1313 | 0.055 | 1.419 | 0.990–2.048 |
| Males |  | 4.6% | 56 | 1163 | 1219 |  | 1.393 | 1.002–1.935 |
| Total | 1.08 | 5.5% | 140 | 2392 | 2532 |  |  |  |
| Exposed to any macrolide |  |  |  |  |  |  |  |  |
| Females |  | 32.4% | 453 | 942 | 1396 | 0.011 | 1.279 | 1.053–1.555 |
| Males |  | 27.3% | 223 | 593 | 816 |  | 1.188 | 1.038–1.360 |
| Total | 1.71 | 30.6% | 676 | 1536 | 2212 |  |  |  |
| Exposed to any macrolide |  |  |  |  |  |  |  |  |
| Eradication-purpose | 1.66 | 51.4% | 247 | 234 | 481 | <0.001 | 3.202 | 2.583–3.971 |
| Non-eradication-purpose | 1.73 | 24.8% | 429 | 1302 | 1731 |  | 2.072 | 1.838–2.335 |
| Total | 1.71 | 30.6% | 676 | 1536 | 2212 |  |  |  |
| Previous non-eradication-purpose macrolide use |  |  |  |  |  |  |  |  |
| Clarithromycin | 1.89 | 36.2% | 312 | 549 | 861 | <0.001 | 3.655 | 2.863–4.688 |
| Any other macrolide | 1.59 | 13.4% | 117 | 753 | 870 |  | 2.695 | 2.227–3.260 |
| Total | 1.73 | 24.8% | 429 | 1 302 | 1 731 |  |  |  |

Cla eradication: previous unsuccessful eradication attempt using a clarithromycin-containing *H. pylori* eradication regimen. Macrolide: clarithromycin or any other macrolide antibiotic.
*Cla* clarithromycin, *susc* susceptible, *F/M* female/male ratio, *p* p value from significance test (significant when <0.05), *OR* odds ratio, *RR* risk ratio, *CI* confidence interval.

concluded that only 1.3% of primary (macrolide-naive) Cla-res infections developed from spontaneous mutations. This strongly supports the idea that it is very rare for Cla-res *H. pylori* clones to compete successfully with Cla-susceptible bacteria in the absence of macrolide-induced selection pressure. Although the relative

transmission fitness of Cla-res bacteria is predicted to be 0.72 (0.58–0.95), meaning that resistance is somewhat costly for transmission, transmitted resistant strains are nonetheless responsible for 5.7% (4.67–7.43%) of all new infections. In another aspect, 98.7% of primary resistant cases are derived from

**Table 3 Parameters and results (outputs) of the mathematical model.**

| Notation | Parameter | Input value \| Output | Source |
|---|---|---|---|
| $\mu$ | Demographic turnover | 0.013 | Hungarian Central Statistical Office, 2005–2013, Supplementary Note 2 |
| $\eta$ | Development of heteroresistance due to macrolide use | 0.129 | Supplementary Note 1 |
| $\rho_u$ | successful eradication | 0.92 | Supplementary Note 1 |
| $\rho_w$ | No effect of Cla treatment | 0.039 | Supplementary Note 1 |
| $\rho_r$ | Cla-treatment-induced homoresistance | 0.029 | Supplementary Note 1 |
| $\rho_h$ | Cla-treatment-induced heteroresistance | 0.012 | Supplementary Note 1 |
| $\theta$ | Cla treatment rate | >0 \| 0.00295584 | Model output |
| $m$ | Rate of macrolide consumption | 0.00889 | Supplementary Note 1, 2 |
| $\omega$ | Spontaneous mutation rate | 0.0000155 | Reference: Linz et al.[30] |
| $\beta$ | Transmission rate | >0 \| 0.019 | Model output |
| $\delta$ | Relative transmission parameter | >0 \| 0.72 | Model output |
| $\lambda_r / \lambda_w$ | Force of infection with respect to resistant/wild-type strains | >0 \| 0.00040911 ($\lambda_r$) <br> >0 \| 0.00676164 ($\lambda_w$) | Model output |
| $H$ | *H. pylori* infection prevalence | 0.409 | Supplementary Note 1 |
| $\frac{dH}{dt}$ | Prevalence trend | −0.002 | Reference: Hooi et al.[1] |

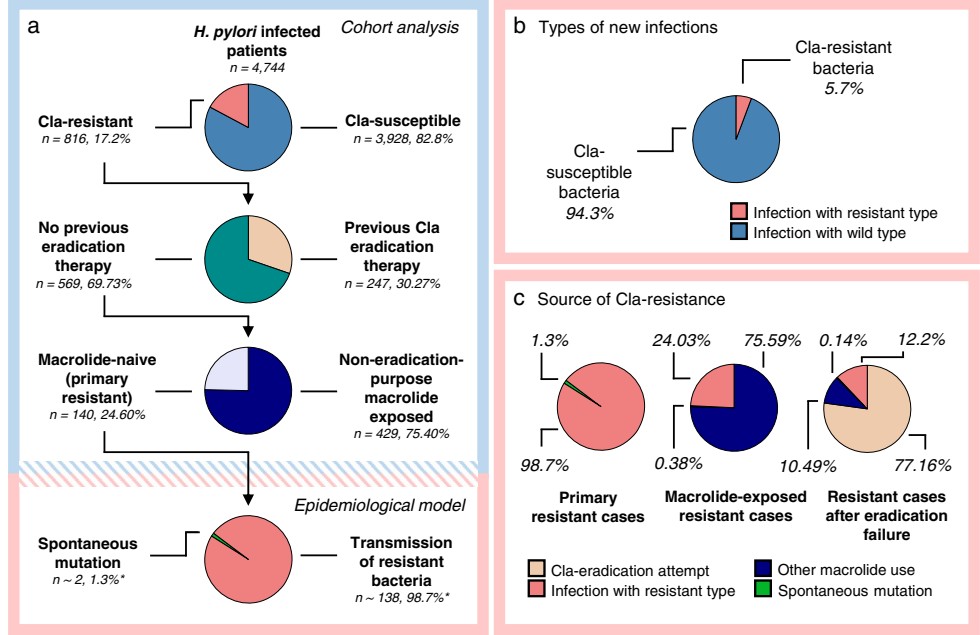

**Fig. 3 Population dynamics of clarithromycin resistant *H. pylori*.** The results of the retrospective cohort analysis are framed in light blue. The conclusions of the constructed epidemiological model are framed in light red. **a** Source of Cla-res in primary and secondary clarithromycin-resistant infections. Secondary resistance cases were divided according to their anamnestic drug exposure (prior clarithromycin-containing eradication therapy/non-eradication-purpose macrolide consumption) obtained from the cohort data (Table 2). The sources of primary resistance cases were inferred from the epidemiological model. Colors of the pie charts: Cla-resistant – light red, Cla-susceptible – light blue, No previous eradication therapy – dark cyan, Previous Cla eradication therapy – light brown, Macrolide-naive (primary resistant) – lavender, Non-eradication-purpose macrolide-exposed - dark blue, Spontaneous mutation – green, Transmission of resistant bacteria – light red. **b** Incidence of clarithromycin-susceptible and clarithromycin-resistant infections in uninfected individuals. Colors of the pie chart: Cla-susceptible bacteria – light blue, Cla-resistant bacteria – light red. **c** Sources of resistance in individuals with Cla-res infections. Colors of the pie charts: Previous Cla eradication attempt – light brown, Infection with resistant type – light red, Spontaneous mutation – green, Other macrolide use – dark blue. Cla = clarithromycin, res = resistance/resistant, *H. pylori = Helicobacter pylori*.

a transmitted resistant strain, according to our mathematical model, although one-third of those take hold by overcolonizing a Cla-susceptible infection. Thus, spontaneous mutation seems to have only marginal significance in the origin of primary Cla-res cases compared to the spread of resistant strains. This

underscores the importance of controlling the population level of clarithromycin resistance.

We have also modeled the effect of discontinuing the use of macrolides for indications other than *H. pylori* eradication. In the model, this approach noticeably slows the long-term growth rate

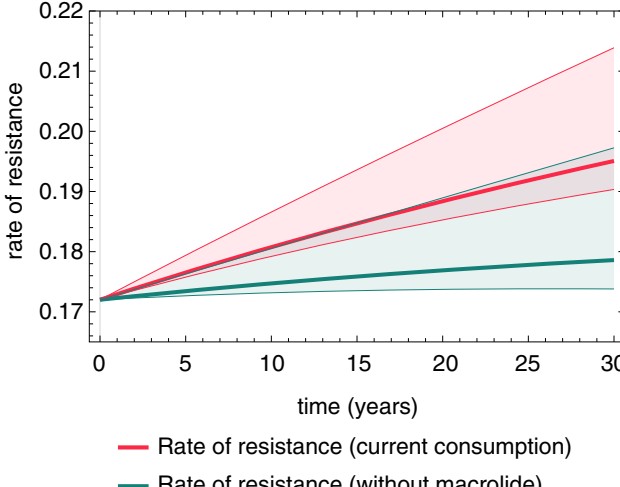

**Fig. 4 Modeling the future prevalence of *H. pylori* clarithromycin resistance with and without non-eradication-purpose macrolide use in the population.** The model shows a noticeably slower long-term growth rate of Cla-res prevalence among infected individuals with the discontinuation of non-eradication-purpose macrolide use (green line) than with the current situation (with a constant level of macrolide consumption), in which the model predicts a more rapidly increasing rate of Cla-res infections (red line). Cla = clarithromycin, res = resistance/resistant, *H. pylori* = *Helicobacter pylori*.

of Cla-res prevalence compared to the current practice, whereas if the current strategy is continued (with a constant level of macrolide consumption), the model predicts a faster increase in the prevalence of clarithromycin resistance. In order to explain the predicted outcome of the first strategy, it is important to note that *H. pylori* infection usually persists for life without medical eradication[3]; consequently, the existing Cla-res cases remain resistant in the long term, and so the prevalence of resistance will not drop substantially if the use of macrolides is discontinued. Moreover, the cumulative effect of transmission of the Cla-res *H. pylori* strains (which can generate new resistant cases not only by infecting previously uninfected persons but also by superinfection of individuals carrying susceptible strains), spontaneous-mutation-related primary resistance and unsuccessful eradication attempts with clarithromycin-containing regimens will further increase the prevalence of Cla-res. Consequently, the overall result of the effects discussed above is a slower but still increasing trajectory of the prevalence of clarithromycin resistance. This is in close agreement with the experiences in Taiwan, where a reimbursement regulation restricted the use of antibiotics from 1 February 2001 in patients with upper respiratory infection[15]. Since clarithromycin was frequently used for this indication before the regulations changed, consumption of macrolide antibiotics declined substantially (approximately by 70%) between 2000 and 2006, whereas the Cla-res rate did not drop considerably among the eradication-naive *H. pylori* infections but showed a slight increasing trend in the 2006–2011 period.

We used the FISH technique to detect point mutations in 23S rRNA, which are reported to be the genetic basis of phenotypic Cla-res in the great majority of resistant *H. pylori* infections[9]. Previous studies demonstrated a strong correlation between phenotypic and FISH-detected genotypic Cla-res[34,35]. Contrary to culture-based susceptibility testing methods, for which the success rate did not exceed 60–70% in certain studies[36], FISH is able to detect *H. pylori* Cla-res from any standard formalin-fixed, paraffin-embedded tissue sample. A further advantage is that

resistance status is diagnosed from morphologically identified, viable *H. pylori* bacteria, and mixed populations of susceptible and resistant *H. pylori* bacteria (intraniche heteroresistance) can be easily detected. Our results showed that nearly half of the resistant cases were heteroresistant. Such a high rate of heteroresistant infections has already been observed in other studies but it is not widely known, and even the Maastricht V/Florence Consensus Report does not include any particular treatment recommendation for these cases[2]. However, given the high prevalence of this type of resistance, further studies should clarify optimal therapeutic strategies for heteroresistant cases, in line with our recently published concept of heteroresistance[20].

In accordance with the literature[37,38], we found significantly higher Cla-res prevalence in females (19.8%) than in males (13.7%). The Cla-res rate exceeded 20% in all age groups of women under 70 years of age, while it was lower than 15% in men except those in the age group of 30–39 years, which exhibited a slightly higher rate (15.3%). To put this into context, women in our cohort were observed to have higher macrolide consumption than men, in accordance with the literature[39,40]. *H. pylori* eradication with a clarithromycin-containing protocol is not recommended without prior susceptibility testing when the Cla-res rate is greater than or equal to 15% in a population[2]. Therefore, based on our results, susceptibility testing is suggested prior to the use of clarithromycin for eradication in women under 70 years of age and men between 30 and 39 years of age in the studied population. Conversely, considering the need for cost-effectiveness, testing is not necessarily required in adult males except those aged 30–39 years or in women older than 70 if there is no history of prior macrolide use. Recent studies confirmed that susceptibility-guided clarithromycin-containing treatments can achieve eradication rates as good or even better than those of other anti-*H. pylori* regimens[41]. As clarithromycin-based conventional triple therapy is still the safest eradication treatment, especially with regard to side effects causing long-term changes in the gut microbiota, this therapy is worth keeping in use[41].

The prevalence of secondary resistance (the Cla-res rate of the macrolide-exposed group) was found to be 30.5% in our study. A significant difference was found between the rates of resistance caused by eradication (51.3%) and secondary resistance related to other macrolide treatments (24.8%), suggesting that macrolide therapies for purposes other than eradication have lower potential to cause Cla-res. This was confirmed by further analyses showing that both non-eradication use of clarithromycin and treatment with other macrolides led to Cla-res less frequently (36.2% and 13.4%, respectively) than clarithromycin-containing eradication therapy. Accordingly, the greatest risk factor for acquiring Cla-res is an unsuccessful eradication attempt, which underscores the need to find more effective therapeutic approaches for *H. pylori* infections. Moreover, by minimizing the use of clarithromycin for other indications, a reduced secondary resistance rate would be achieved[15]. As other macrolides were found to cause significantly less Cla-res, replacement of clarithromycin with other macrolides or nonmacrolide antibiotics is recommended for indications other than *H. pylori* eradication.

As with the majority of studies, our study also has some limitations. One of these is that the FISH assay detects only the three most prevalent and well-documented Cla-res mutations of *H. pylori*. However, several studies have demonstrated that there is an almost perfect (90–100%) correlation between these three mutations and phenotypic resistance[9], which was also confirmed by the FISH technique that we used[34,35,42]. Our test is unable to diagnose rare Cla-res mutations or resistance caused by uncommon genetic mechanisms, which is a limitation of this study. However, this seems to have very limited relevance, as further point mutations of the peptidyl transferase region of 23S rRNA

are very rare or associated with little or no phenotypic resistance[43,44]. Furthermore, due to the study design, the macrolide-use-related homo-/heteroresistance ratio is, unfortunately, not available in detail for each subgroup in this cohort; therefore, some of the model parameters were calculated by using the data of one of our previous publications. Regarding the prediction of the spontaneous mutation versus transmission origins of the primary (macrolide-naive) resistance cases (1.3% vs. 98.7%), it is important to note that the *H. pylori* bacteria—including the resistant bacteria in the heteroresistant cases—show an uneven and patchy mucosal distribution pattern, especially if the bacterial density is low[45]. Moreover, a long time period might be necessary for the spread of resistance from one mutated bacterium in the whole bacterial population in the absence of selection pressure from macrolides. Thus, the sampling error (no resistant bacteria in the biopsy specimen) might lead to underestimation of the presence of mutated (resistant) bacteria and consequently the prevalence of these spontaneous-mutation-originated cases. Our epidemiological model accounts for the most important state transitions due to infection, medication use, and emergence of resistance. To keep the equations tractable, we made several simplifications. We assumed that infections with different Cla-res statuses share certain parameters. We ignored the possibility of unintended *H. pylori* eradication, the effect of repeated macrolide use, and the differences in the applied macrolide doses, since we assume that these variables do not have a potentially significant impact on the overall epidemiological dynamics. We also ignored the differences in the resistance-inducing potential of non-eradication-purpose clarithromycin and other macrolides for lack of information on the homo-/heteroresistance ratio in Cla-res infections of these treatment-related subcohorts. We assumed that a heteroresistant infected host transmits either the wild-type (Cla-susceptible) isolate/strain or the resistant isolate/strain during an infection episode and cannot transmit both wild-type and resistant bacteria at once to the same host. We assumed that spontaneous mutation and macrolide use in a host infected with wild-type (Cla-susceptible) *H. pylori* give rise to heteroresistance and that the emerged resistant strain cannot completely eliminate the already established wild-type population from within the host. The model also has the usual limitations of compartmental models: it assumes a homogeneous, well-mixed population and mass-action incidence.

In conclusion, we have demonstrated an age-dependent preponderance of females in secondary (acquired) Cla-res of *H. pylori* infections. This resistance may be caused not only by unsuccessful eradication attempts but also by prior consumption of any other macrolide. Conversely, the prevalence of real primary Cla-res is significantly lower among *H. pylori* infections than previously assessed by the conventional (eradication-history-based) clinical approach. This type of resistance originates predominantly from transmitted Cla-res strains but occasionally from spontaneous mutations as well. In light of all this, it is highly important to identify any prior macrolide use by patients, as this increases the chance of secondary resistance to clarithromycin-containing eradication therapy. Thus, preliminary susceptibility testing is recommended for patients with a known history of macrolide antibiotic exposure as well as in those locally established age- and sex-based subgroups where the Cla-res rate is greater than or equal to 15%. We found a high prevalence of heteroresistant *H. pylori* infections, which demonstrates the need for further studies to optimize therapeutic strategies for these cases. Our unique mathematical model describing the transmission dynamics of Cla-susceptible and Cla-resistant *H. pylori* infections could be useful for predicting the Cla-res rates of different populations in further studies.

**Table 4 Cohort characteristics.**

| Parameter | n or mean | % or standard deviation |
|---|---|---|
| Sex | | |
| Female | 2709 | 57.1% |
| Male | 2035 | 42.9% |
| Age | | |
| Years | 55.56 | ±16.23 |
| Requested by | | |
| University clinic | 958 | 20.2% |
| General hospital/outpatient clinic (Budapest) | 3720 | 78.4% |
| General hospital/outpatient clinic (not Budapest) | 66 | 1.4% |
| Histological findings[a] | | |
| No gastritis | 16 | 0.3% |
| Inactive gastritis | 3299 | 69.5% |
| Active gastritis | 1429 | 30.1% |
| Mucosal structural alteration[b] | 465 | 9.8% |

[a]Overlap of diagnoses is possible (between gastritis and mucosal structural alterations). [b]Structural alterations of the gastric mucosa induced by chronic *H. pylori* gastritis, including intestinal metaplasia, atrophy/preatrophy, erosion, ulcers, foveolar hyperplasia, hyperplastic polyps, dysplasia, carcinoma, lymphoid follicles, and MALT lymphoma.

## Methods

**Study design.** For this retrospective study, *H. pylori*-infected patients with known FISH-based clarithromycin susceptibility test results were collected from 2005–2013 using our institutional electronic database. The exclusion criteria were cases with unknown, missing or false Hungarian social security identification numbers (TAJs) and negative *H. pylori* FISH test results (as small foci of *H. pylori* observed in the slides used for the histopathological diagnosis might disappear from further section planes, resulting in negative *H. pylori* FISH tests and thus inconclusive susceptibility test results). Patients who were sampled and tested repeatedly were included in the cohort, but only the first test result for each person was entered. Gastric tissue samples were obtained by mapping biopsy during upper digestive endoscopy in one of the 16 requesting institutions (university clinics, hospitals, and outpatient clinics) responsible for patient care in Budapest and the Central Hungary region. *H. pylori* clarithromycin susceptibility was successfully determined for 4,744 patients, whose cohort characteristics are described in Table 4. As the age group of 0–19 years contained only 51 patients (1.1%), these pediatric cases (which represent a distinct subgroup) were excluded from further age-group-based comparisons and calculations.

The study protocol followed the ethical guidelines of the 1975 Declaration of Helsinki and was approved by the Ethical Committee of Semmelweis University, Budapest (#148/2007 and #97/2012). Based on the current Hungarian law for scientific research, contacting patients to obtain informed consent is generally not requested for retrospective studies. Accordingly, the Ethical Committee of Semmelweis University, Budapest, waived the informed consent procedure for the study.

In the initial phase of data collection, authors from the 2nd Department of Pathology, Semmelweis University (É.K., I.S., I.K., Z.K., A.K., Z.S., and G.L.) were able to access the patients' clinical data in connection with the results of the histopathological examinations and the Cla-susceptibility FISH test, which were also available to the treating gastroenterologists (including authors G.M.B. and A. S.) in the case of their own patients. In the second phase, after a data collection request available for researchers with ethics approval, we collaborated with the National Health Insurance Fund Administration, Budapest, Hungary (NEAK), to obtain the patients' individual drug-dispensing data. For this purpose, we provided NEAK with a patient list containing the Hungarian social security identification number ("TAJ") and the result of the Cla-susceptibility FISH test for each patient. NEAK linked the patients' individual drug-dispensing data to the TAJ numbers and analyzed the data to determine Cla-res prevalence in treatment-related subcohorts of the patients (e.g., macrolide-exposed/macrolide-naive/prior Cla eradication attempt, etc.) according to age and sex groups. In this stage, only NEAK staff (including one of our co-authors, P.F.-F.) had access to the combined data. Subsequently, NEAK provided the results to the authors in the form of aggregated data of the investigated groups.

**Diagnosis of *H. pylori* infection.** According to the clinical sample collection practice of the study period, the mapping biopsy tissue samples of an individual arrived in the same collection tube for histopathological workup in the majority of

**Table 5 Model compartments.**

|  | Uninfected | Wild-type infected | Resistant infected | Heteroresistant |
|---|---|---|---|---|
| Macrolide-naive | $u^n$ | $i_w^n$ | $i_r^n$ | $i_h^n$ |
| Macrolide history | $u^m$ | $i_w^m$ | $i_r^m$ | $i_h^m$ |
| Clarithromycin history | $u^c$ | $i_w^c$ | $i_r^c$ | $i_h^c$ |

*u* uninfected, *i* infected, index *w* wild type (clarithromycin susceptible), index *r* resistant (clarithromycin homoresistant), index *h* heteroresistant infection, index *c* used clarithromycin for eradication purposes, index *m* used macrolides for other purposes, index *n* macrolide-naive.

cases (4,398/4,744). Accordingly, these samples were embedded in the same paraffin block. Samples from the antral and oxyntic gastric compartments were distinguishable during routine histological investigation based on the expertise of the pathologists. The *H. pylori* diagnostic staining results were summarized considering the staining patterns observed from all the individual tissue samples of a given patient, independent of whether they were embedded in the same or separate tissue blocks.

*H. pylori* positivity was determined in formalin-fixed, paraffin-embedded tissue material based on positivity by modified Giemsa staining and/or *Helicobacter* immunohistochemistry with B0471 polyclonal rabbit anti-*H. pylori* primary antibody (Dako, Glostrup, Denmark)[45] and confirmed by a BACTFish *H. pylori* Combi Kit (Izinta Kft., Budapest, Hungary)[20,45].

**Clarithromycin susceptibility FISH test.** FISH for the detection of *H. pylori* Clares was performed according to the instruction manual of the BACTFish *H. pylori* Combi Kit[20,45]. As an initial step, 4- to 5-µm-thick sections were cut from the paraffin-embedded tissue specimens. These sections were attached to glass slides by heating on a hot plate at 65 °C for 10 s and subsequently deparaffinized by sequential immersion in xylene twice for 30 min each and 96% ethanol twice 30 min each. Then, the air-dried slides were immersed in Vector Antigen Unmasking Solution (Cat. No. H-3300, Vector Laboratories) and microwaved at 400 W for 20 min as heat pretreatment, after which the slides were rinsed in distilled water three times and air-dried at room temperature. The tissue sections were then placed into a humidity chamber and hybridized for 90 min at 46 °C with the DNA Probe Mix of the BACTFish *H. pylori* Combi Kit, containing an *H. pylori*-specific probe (Hpy-1, targeting 16S-rRNA, labeled with green fluorochrome fluorescein isothiocyanate) and specific probes for the detection of Cla-res-specific distinct point mutations within the peptidyl transferase region of 23S-rRNA (ClaR1 [A2143G], ClaR2 [A2144G] and ClaR3 [A2143C], labeled with orange-red fluorescent Cy3) as well as an unlabeled wild-type probe (ClaWT) to avoid non-specific hybridization of the ClaR1-3 probes (probe sequences are displayed in Supplementary Table S8)[42]. After a posthybridization wash in prewarmed Wash Buffer (2 × 15 min at 46 °C), the slides were mounted with antifade fluorescent mounting medium containing the blue-fluorescent nuclear counterstain 4′,6-diamidino-2-phenylindole, or DAPI (Vectashield with DAPI, Cat. No. H-1200, Vector Laboratories, Burlingame, CA, USA). Then, the slides were examined using a Leica DM RXA epifluorescence microscope (Leica Microsystems, Wetzlar, Germany) equipped with DAPI (excitation 355–425 nm, emission >470 nm, Leica Microsystems, Wetzlar, Germany), Spectrum Green (excitation 460–500 nm, emission 512–542 nm, Vysis, Downers Grove, IL, USA), and Spectrum Orange (excitation 540–578 nm, emission 564–612 nm, Vysis, Downers Grove, IL, USA) filter sets for blue, green, and orange-red fluorescence, respectively. Images were taken using a Leica DFC365 FX camera and documented by Leica CW4000 FISH software.

*H. pylori* infections were categorized semiquantitatively as mild (1+; low bacterial density, scattered *H. pylori* distribution pattern), moderate (2+; moderate bacterial density, small groups of *H. pylori*), or severe (3+; high bacterial density, large groups/fields of *H. pylori*), according to the updated Sydney system[46,47], considering the FISH result of each gastric-mucosa-containing tissue specimen (Fig. 1).

Cla-res was diagnosed and classified according to the specific fluorescence shown by *H. pylori* bacteria. Cases with bacteria showing only green fluorescence (no resistance-specific probe binding) were considered susceptible to clarithromycin (Fig. 1a). Clarithromycin-homoresistant *H. pylori* infections were diagnosed by the exclusive presence of bacteria showing green and orange-red fluorescence simultaneously (as both the species-specific probe and one of the resistance-specific probes were bound to bacterial rRNA), which appeared as yellow fluorescent bacteria on the merged-channel images (Fig. 1b). A mixed population of susceptible and resistant bacteria in the same sample (with green and yellow bacteria on the merged image) was classified as clarithromycin heteroresistant (Fig. 1c). Cases exhibiting discrepant Cla-res status at different sampling sites were also considered to be Cla-heteroresistant. Both homo- and heteroresistant infections were classified as Cla-res.

**Analysis of prior antibiotic consumption.** The individual patients' antibiotic dispensing data, both related and unrelated to *H. pylori* eradication, were collected in cooperation with the National Health Insurance Fund Management of Hungary,

whose prescription database has been available since 2000. In this study, all medicines prescribed by a medical doctor for any patient in Hungary were recorded by their Anatomical Therapeutic Chemical classification codes. The medical procedures and related diagnoses of the individual patients are also recorded in this database by their OENO (Hungarian adaptation of the WHO International Classification of Procedures in Medicine) codes and WHO International Classification of Diseases codes, respectively. In Hungary, oral antibiotics are available only by prescription, as are the proton pump inhibitors that are used in eradication therapy. Therefore, individual drug-dispensing data, representing the exclusive source of these medicines, were found to be appropriate for assessing eradication-related and eradication-independent macrolide use. Clarithromycin was considered to be dispensed for eradication purposes if the prescription was related to gastroduodenoscopic biopsy or if the clarithromycin was dispensed within 1 week of other medicines from any known combination regimen for *H. pylori* eradication.

**Mathematical modeling of the transmission dynamics of *H. pylori*.** A detailed description of this mathematical model (including the full code used for calculations) is given in Supplementary Notes 1, 2, and 3. Briefly, our patients represent the infected individuals in the whole population of Central Hungary. We classified this population according to infection status (uninfected [*u*]; infected [*i*] with a wild-type [Cla-susceptible] strain [index *w*], a Cla-homoresistant strain [index *r*] or a heteroresistant strain [index *h*]) and medication history (used Cla for eradication purposes [index *c*]; used macrolides for other purposes [index *m*]; macrolide-naive [index *n*]). This produces a total of 12 compartments (Table 5). The notation is straightforward; for example, $i_r^c(t)$ denotes the fraction of the population that has a history of clarithromycin use for eradication and is infected with a resistant strain at time *t*.

Newborns in the modeled population appear in the macrolide-naive uninfected compartment. When uninfected individuals acquire infection from wild- or resistant-type-infected hosts, they move to the corresponding infected compartment, with transmission rates $\beta$ and $\delta\beta$; the factor $\delta$ expresses the relative transmission fitness of resistant strains. Wild- or resistant-type-infected individuals can be colonized by the other strain with the same transmission rates, in which case they become heteroresistant. Heteroresistant hosts generate wild-type infections with probability $\kappa$ and resistant infections with probability $1-\kappa$. Macrolide-naive individuals use macrolides with rate *m*, at which point they move to the corresponding compartment with a history of macrolide use; in a fraction $\eta$ of wild-type-infected individuals, resistance emerges, and they become heteroresistant. Infected individuals with or without a history of macrolide use undergo clarithromycin treatment for the purpose of eradication at a rate of $\theta$. Such treatment can have various outcomes: wild-type infections can be successfully eradicated (with probability $\rho_u$) so that the patients revert back to uninfected; treatments can fail so that the hosts remain infected with wild-type bacteria (with probability $\rho_w$); or resistance may emerge in the hosts, placing them into resistant or heteroresistant compartment (with probabilities $\rho_r$ and $\rho_h$, respectively), where $\rho_u + \rho_w + \rho_r + \rho_h = 1$. If clarithromycin treatment is administered to a heteroresistant host, it may eradicate the wild-type bacteria and cause a resistant infection (with probability $\rho_u + \rho_r$), or the host might remain heteroresistant (with probability $\rho_h + \rho_w$). Treatment has no effect on resistant infected hosts. We allow the possibility of spontaneous mutations (with rate $\omega$), such that these hosts move from a wild-type-infected compartment to the corresponding heteroresistant compartment. The transmission diagram is depicted in Fig. 5.

We obtained the current states of the known infected compartments ($i_r^n$, $i_w^n$, $i_w^m$, $i_w^c$, $i_h^n$) and one of the model parameters (*m*: rate of macrolide consumption) directly from the cohort. Further model parameters ($\mu$: demographic turnover; $\eta$: development of heteroresistance due to macrolide use; $\rho_u$: successful eradication; $\rho_w$: no effect of Cla treatment; $\rho_r$: Cla-treatment-induced homoresistance; $\rho_h$: Cla-treatment-induced heteroresistance; $\omega$: spontaneous mutation rate; *H*: *H. pylori* infection prevalence and prevalence trend) were assessed by using published data or publicly available databases[20,31,48,49]. We will now briefly explain some of these variables. A slowly declining prevalence of *H. pylori* has been observed in several European countries[1], corresponding to a decrease of ~0.2% per year. Therefore, we also modeled a slowly decreasing trend in the prevalence of *H. pylori* by using this −0.2% annual rate as baseline, allowing it to vary from −0.1% to −0.3%. The demographic parameters for the investigated years were taken from the public

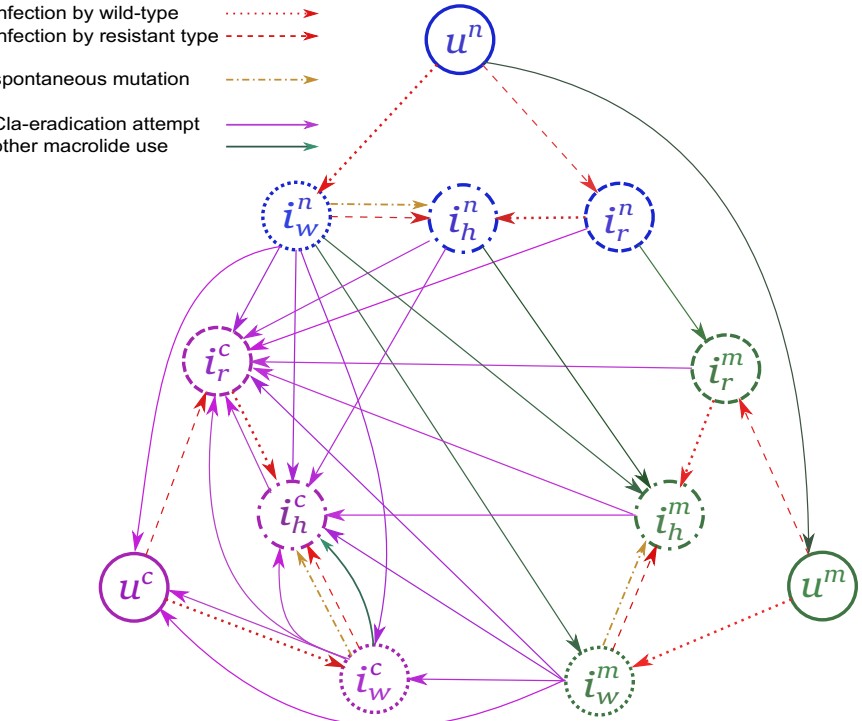

**Fig. 5 Simplified transfer diagram of the mathematical model describing the transmission dynamics of clarithromycin-resistant** *Helicobacter pylori*. Arrows indicating the transitions between the compartments: Infection with wild-type – red dotted arrow, Infection with resistant type – red dashed arrow, Spontaneous mutation – gold dashdotted arrow, Cla-eradication attempt – purple solid arrow, Other macrolide use – green solid arrow. The different compartments are indicated by the following notations: uninfected individuals as $u^n$ – macrolide-naive, $u^m$- macrolide-exposed, $u^c$- after Cla-containing eradication therapy, individulas with wild-type infection as $i_w^n$ - macrolide-naive, $i_w^m$ – macrolide-exposed, $i_w^c$ – after Cla-containing eradication attempt, individuals with homoresistant infection: $i_r^n$ – macrolide-naive, $i_r^m$ – macrolide-exposed, $i_r^c$ – after Cla-containing eradication attempt, individuals with heteroresistant infection: $i_h^n$ – macrolide-naive, $i_h^m$ – macrolide-exposed, $i_h^c$ – after Cla-containing eradication attempt. Cla = clarithromycin.

databases of the Hungarian Central Statistical Office (https://www.ksh.hu/?lang=en). We assumed that the use of macrolides for non-eradication purposes in the cohort was representative of the whole population (when adjusted for age), considering their non-gastroenterological indications. In contrast, we did not accept the annual rate of cases undergoing clarithromycin-containing eradication treatment in our cohort to be representative of the whole population because the cohort individuals are *H. pylori*-infected gastroenterology patients and consequently have a higher frequency of previous eradication treatment. A more detailed description of these parameters is displayed in Table 3 and in Supplementary Notes 1 and 2. The remaining parameters are derived from the compartmental model and are shown in Table 3 and Supplementary Note 2.

Furthermore, we used the inferred parameters and current state of the system to compare future scenarios by solving the system forward in time with the actual parameters and, for the sake of comparison, assuming that non-eradication-purpose macrolide use would be discontinued. All calculations and plotting of the mathematical model were performed by Wolfram Mathematica software (version 10.4.0.0). The used code is described in Supplementary Note 3, as well as provided as Supplementary Software 1.

**Statistical analysis.** Categorical data were analyzed using 2 × 2 or 2 × 3 contingency tables and compared using Fisher's exact probability test. Associations between age, sex, and proportions of resistant and heteroresistant infections were calculated using a generalized linear model with binary response variables. R software version 3.5.1 (www.r-project.org) with packages epitools (version 0.5–10.1) and ggplot2 (version 3.3.0) was used for the statistical analyses presented in the main text. All *p* values were calculated using a two-tailed method, and differences were considered significant when *p* < 0.05.

**Reporting summary.** Further information on research design is available in the Nature Research Reporting Summary linked to this article.

## Data availability

The authors declare that the data supporting the findings of this study are available within the paper and its supplementary information files. The patients' individual drug-dispensing data are not publicly available due to them containing information that could

compromise patients' privacy. Such data were analyzed by staff of the National Health Insurance Fund Administration of Hungary (NEAK), including one of the authors (P.F-F.), after a data collection request available for researchers with ethics approval. Only the aggregated data and the results of the analyses were available to the rest of the authors. The aggregated data provided by NEAK are available within the paper and its Supplementary Information files. Further information on the data collection process is available in the "Methods" (Study design) section. The demographic parameters of the Central Hungarian population were taken from the public database of the Hungarian Central Statistical Office (https://www.ksh.hu/?lang=en). For any further enquiries about the collection process or availability of the data, please contact the corresponding author.

## Code availability

Wolfram Mathematica code used for the mathematical model is provided as Supplementary Software 1 and is also available at Wolfram Cloud (https://www.wolframcloud.com/env/61da20c7-76f5-4fe9-87a6-3174076d6f8c). Further explanations about the numerical calculations and simulations are provided in Supplementary Notes 2 and 3.

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

## Acknowledgements

We would like to give special thanks to our technicians, Mrs. Magdolna Pekár and Mrs. Csilla Horváth. We also gratefully acknowledge all the support and work of Dr. Mihály Kramer, an excellent, enthusiastic scientist, coauthor, and colleague who was very active and contributed greatly to the success of this project but passed away recently. We would also like to show our gratitude to Prof. Francis Mégraud for his useful advice during the revision of the manuscript. This study was partly supported by the New National Excellence Program (ÚNKP-19-3-I-SE-65, ÚNKP-19-3-I-SE-73, and ÚNKP-20-3-II-SE-36) and the Thematic Excellence Programme (2020-4.1.1.-TKP2020) of the Ministry for Innovation and Technology (Hungary), the New National Excellence Program (ÚNKP-18-3-I-SE-44), and grant 20391-3/2018/FEKUSTRAT from the Ministry of Human Capacities (Hungary), the K_18 128881 and KKP129877 grants by the National Research, Development and Innovation Office of Hungary, as well as a start-up grant from Semmelweis University, Budapest, Hungary.

## Author contributions

G.L. and É.K. conceived and designed the study and wrote the manuscript. É.K., G.M.B., I.S., I.K., Z.K., A.S., P.F.-F., K.S., M.R., M.F., A.K., Z.S., G.R., and G.L. collected and/or analyzed the data. É.K. and G.L. performed the statistical analysis. G.R. led the mathematical modeling. A.K., Z.S., G.R., and G.L. oversaw the study. G.L. and G.R. jointly supervised this work. All authors contributed to interpreting the data and reviewing and editing the article, and all authors approved the final version of the manuscript. The funders of the study had no role in the study design; the data collection, analysis, or interpretation; or the writing or submission of the report. The corresponding author had full access to all the data used in the study and had final responsibility for the decision to submit the article for publication.

## Competing interests

The authors declare no competing interest.
