## [Peer Review File · Nature Communications]

Reviewers' comments:

Reviewer #1 (Remarks to the Author):

This is a large retrospective observational study of Clarithromycin resistance in *H.pylori* strains in a region in Hungary. Fluorescent in situ hybridization of gastric biopsies was used to determine Clarithromycin resistance strains in the population, including mixed strains (some resistant, some not) in the same patient (hetero resistance).

Because each patient's history of macrolide intake is available through a prescription database, the authors were able to establish Clarithromycin resistance in individual patients and develop a mathematical model to describe how many of the strains were linked to prior Clarithromycin usage by the individual and how many were not.

The novelty of the paper is that a mathematical model was developed that was able to determine how many of the Clarithromycin resistant strains were due to the development of resistance in vivo secondary to macrolide usage for other infections, and how many were due to the transmission of preexisting Clarithromycin resistant strains.

The study has been carefully performed and well described.

However, the clinical significance of utilization of such a model is unclear as the decision to use a Clarithromycin based regimen or not in any population is primarily based on the background clari-resistance in that population rather than in an individual. However, classical microbiological and molecular techniques are now available to determine Clarithromycin in an individual. With these tools, it is unclear how the mathematical model here adds to how antibiotic choice is determined.

Other Comments:

1. Page 6, first paragraph. How many cases had multiple biopsies and discrepant results?
2. Page 6, paragraph 2. Over how many prior years does the pharmacy database record prescription usage?
3. Page 8, figure 3. Are any of these differences across age groups statistically significant?
4. Page 9, first paragraph. "previously eradicated" should be changed to "previously treated".

Reviewer #2 (Remarks to the Author):

Review for: Primary and secondary clarithromycin resistance of *Helicobacter pylori*: role of macrolides and mathematical modelling of resistance independent of macrolide consumption, Kocsmar et al

This study combines epidemiological data analysis and mathematical modelling to elucidate the origin of resistance to the clarithromycin antibiotic in *Helicobacter pylori* bacteria. Notably, the authors aim to clarify the proportion of resistant infections which originate from the transmission of resistant bacteria from another person, and the proportion attributable to spontaneous mutation, as well as the correlation between clarithromycin-resistant *H. pylori* infections and previous exposure to clarithromycin or other macrolide antibiotics. This study is set in Hungary, where antibiotic prescribing is recorded for each person, therefore allowing the authors to link previous antibiotic exposure and current *H. pylori* infection status.

The retrospective cohort analysis is clearly presented in the Methods and Results sections in the main text and leads to strong and potentially important conclusions on the link between macrolide consumption and risk of clarithromycin-resistant infection. However, some crucial assumptions made in the model are not discussed, which prevents the results of the model from being truly

meaningful. In addition, some key information on *H. pylori* characteristics and epidemiology is missing from the Introduction section, which could cause confusion for the wider audience of Nature Communications.

We were asked specifically to look at the mathematical modelling included in this paper. The model structure itself is described well and is clear. However, we do not believe that the results are reported clearly – the headline results do not clarify the other results (e.g. equilibrium values for populations infected) that these results rely on. Single percentage values for the contribution of transmission are inappropriate and misleading. We do agree that a model such as this was an appropriate tool for investigating the question, but this was not made clear in the introduction nor the context for this model discussed. Has such a model been attempted before? How do the results compare to other *H. pylori* modelling literature? A big flaw is that no attempt is made in the main text to explore how the uncertainty in the model and parameters translates into uncertainty in the results.

Overall, we found it hard to follow much of the analysis – especially in the supplementary. A clearer narrative is needed with more context.

Major comments

1) Some key information is currently missing in the Introduction. We feel that this should be included to make this relevant to the non-specialist audience of Nature Communications. Firstly, it should be mentioned that *H. pylori* is estimated to be present in more than 50% of the global population (Hooi, J. K. et al. (2017). Global prevalence of *Helicobacter pylori* infection: systematic review and meta-analysis), and only causes an asymptomatic infection in the vast majority of cases (Kusters, J. G., van Vliet, A. H., & Kuipers, E. J. (2006). Pathogenesis of *Helicobacter pylori* infection). This should be clearly mentioned to avoid any reader confusion in the rest of the paper regarding the reported prevalence of infection. Secondly, it should also be mentioned that *H. pylori* infection usually persists for life, and that individuals can only clear the infection thanks to eradication treatment (Kusters, J. G., van Vliet, A. H., & Kuipers, E. J. (2006). Pathogenesis of *Helicobacter pylori* infection). This context is important to understand why natural clearance of the bacteria is not part of the mathematical model. Lastly, the capacity for *H. pylori* to perform horizontal gene transfer by transformation should be mentioned (Israel, D. A., Lou, A. S., & Blaser, M. J. (2000). Characteristics of *Helicobacter pylori* natural transformation). Although the dynamics of antimicrobial resistance gene transfer by transformation are admittedly still unclear there is good experimental evidence that this can occur at a significant rate and must therefore be mentioned, since this factor could also contribute to increasing rates of Cla-res.

2) There are multiple elements of the model that are currently unclear, and should be discussed and corrected. Importantly, the work as it is not reproducible despite having the model code (e.g. how were the parameters determined?). These are detailed in the list of revisions below, but broadly include:

- The parameterisation is not supported by sufficient calculation information or any evidence (e.g. reference to estimates from previous literature).
- A wider sensitivity analysis should be included to cover the key results as well as all unknown parameters.
- The effect of previous consumption of macrolides for non-eradication purposes. The model assumes that the effect would be the same regardless of whether these macrolides are clarithromycin or not, but the data in table 4 seems to suggest otherwise (previous users of clarithromycin for non-eradication purposes have a higher risk of Cla-res infection than users of other macrolides)
- Individuals infected by either one of the susceptible or resistant strain are considered to be fully susceptible to over-infection by the other strain. This is a strong assumption, yet it is not currently discussed or the subject of a sensitivity analysis.
- Individuals infected only by the susceptible strain can become infected by the resistant strain as a consequence of eradication treatment. The mechanism by which this can happen is unclear and might require either rethinking whether these individuals are actually hetero-resistant when receiving treatment, or linking the probability of resistant arising in this scenario to the mutation

rate of the bacteria.

- Hetero-resistant individuals can transmit either the susceptible or resistant strain, governed by the parameter κ . The consequence of changing κ for the main results presented in this manuscript (the origin of primary clarithromycin resistance) is not currently tested.
- The model assumes the system is at steady state despite evidence showing the dynamic nature of H pylori infection. What evidence is there to support this? This should be mentioned in the main text.

Minor comments

Abstract

Line 30: Do you mean that clarithromycin resistance is the major cause of unsuccessful treatment *for H. pylori*? In that case, the sentence should be rephrased, as it could be interpreted as "clarithromycin resistance in H. pylori is the major cause of unsuccessful treatment attempts for all bacteria".

Can you make it clear here that clarithromycin is a macrolide? And that it is the drug used for eradication treatment? It was unclear to me what eradication-naïve meant without this information. E.g. in Line 41: could it perhaps be rephrased to "clarithromycin-naïve (for purposes of eradication) patients?"

Line 39: To avoid confusion, I recommend rephrasing this sentence to "The overall rate of clarithromycin resistance amongst all H. pylori infections was 17.2%"

Line 41: the 64/36% split is a result from the modelling – can this be included in this sentence?

Line 47: what are these groups where is it above 15%?

Line 47: as mentioned above, I think you need to be clear what is a modelling result and what comes from the epidemiological cohort

Introduction

Line 55: highest rank where?

Line 57: "Helicobacter pylori" -> "H. pylori" abbreviation could be used

Line 60: What is the empirical triple therapy?

Line 61: "15% of all H. pylori infections"

Line 64: Can you make clear that clarithromycin is a macrolide: e.g. cross resistance exists in macrolide resistance such that the use of other macrolides can lead to clarithromycin resistance?

Line 68: Could you please provide one or two examples of "other purposes" why macrolides could be used? This would help to provide context on the magnitude of this problem.

Line 70: Can heteroresistant populations be found by your lab method or are they hypothesised populations?

Line 74: italics and abbreviation could be used here for Cla-res H. pylori

Line 74: could the authors please clarify whether H. pylori resistance has already been modelled? The reference included is not specific to H. pylori. How does this work add to the existing H. pylori work?

It needs to be clearer what the model was for. "to analyse the population dynamics" is not

sufficient. What could not be determined from the epidemiological information?

Materials and methods

We could not comment on the lab results and hence focused on the Mathematical modelling section only.

Line 148: It is unclear to what population you are referring in this first sentence – are these the patients across the study or in your model? In the model, the patients described in the rest of the study are not explicitly represented. Instead, the model infers the proportion of people infected by *H. pylori* in Central Hungary based on a previously published study.

Line 148: Can you add a link to the supplementary material early on? Otherwise the description of the model and the notation used is very clear.

Line 149: Are there any differences in the symptoms associated with *H. pylori* carriage? As in should we be thinking about those asymptomatic and symptomatic carriers? I assume treatment eradication exposure rates would then be higher for the latter? Is there a time delay – e.g. colonised and then progress to asymptomatic infection for this bacterium? By using the word “infected” here any link to symptoms is ignored which could impact on treatment exposure (and possibly transmission? E.g. higher bacterial load?).

Line 152: The “ m ” index assumes that whatever the effect of macrolides on *H. pylori* resistance may be, it is the same regardless of whether these macrolides are clarithromycin or not. However, Table 4 indicates that the proportion of Cla-res is higher in individuals who used clarithromycin for non-eradication purposes (36.2%), compared to those who used other macrolides for non-eradication purposes (13.4%). Can you clarify how this discrepancy is taken into account in the calculation of the “ η ” parameter?

Line 157: Is there any evidence of a fitness cost to clarithromycin resistance in *H. pylori*?

Line 158: it is assumed that individuals infected by either one of the susceptible or resistant strain are equally likely to be infected by the other strain as non-infected individuals. This is a strong assumption that is not supported by any reference, or discussed in the paper. Could the authors please expand on this? This might require further sensitivity analysis with a new parameter controlling the relative susceptibility to over-infection of already colonised individuals compared to non-infected individuals.

Line 161: Does macrolide use never clear *H. pylori* carriage? Are there no uses of macrolide that could lead to this?

Line 167: The model assumes that resistance can emerge in the hosts only infected by the susceptible strain in response to clarithromycin treatment (with probability pr). If this is because the host is actually also infected at low levels by the resistant strain, then shouldn't they be in the corresponding compartment? (index h) Alternatively, if this is because the susceptible bacteria develop resistance by spontaneous mutation during treatment, and this is then selected for by the presence of clarithromycin, then shouldn't pr be linked to the spontaneous mutation rate ω ?

Line 171: Is resistance always complete? They are not susceptible at all to the eradication treatment?

Figure 2: This is a very informative figure but hard to follow, especially with the current legend. It needs to be stand-alone with more information on all the index notation. Perhaps a simplified version with just the natural history states (e.g. all of iw) could be included. This could then go in the supplementary.

The model is explored at steady state: this needs to be discussed in the main paper. Why was it appropriate to assume that the system is at equilibrium when there has been such a change in *H.*

pylori prevalence as shown by the data in the supplementary?

Presumably, there is currently some level of drug susceptibility testing before eradication treatment is given. Can the estimates for the level of this in Hungary be given and used as a parameter in the model?

Results

It seems that age and sex were significant influencers of the results – why were they not included in the mathematical model? Similarly, should the 1+/2+/3+ categories be included? What impact does their lack of inclusion have on the modelling results?

Line 187: "Females showed a significantly higher Cla-res rate than males"; please include the percentages in the main text here.

Line 191: Should "ratio" be replaced by "proportion"?

Table 4: Clearly lays out all the results. More information could be given in the row titles.

Line 227: What is the denominator for this percentage? New infections just with the resistant strain or with any H. pylori? How does this link to the 5.5% below?

Line 228: What supports this $k = 0.5$ value? Did the lab work find that in heteroresistant cases the bacterial load of R/S was the same? Could you quantify the bacterial burden at all and use this here?

Line 229: Can you add "relative to the susceptible strain"? This is a large fitness cost. Is this supported by other measures in the literature?

Line 231: I would have thought that again this was not a parameter for the model to estimate but that there may be supporting evidence for this. How different is your estimate to this evidence? E.g. in references 21-24 cited in lines 266-8

Line 230: it is assumed that hetero-resistant individuals have a 50-50 chance of transmitting either the resistant or susceptible bacteria ($\kappa = 0.5$). The authors perform a sensitivity analysis by looking at the effect of varying κ between 0 (hetero-resistant individuals only transmit the susceptible strain) and 1 (hetero-resistant individuals only transmit the resistant strain) on the transmission rate β and the relative fitness of the resistant strain δ . However, since the main results are the origins of primary resistance (i.e. the percentage of macrolide-naïve resistant infection originating from transmission of the resistant strain, and the percentage from spontaneous mutation), the sensitivity of these results to changes in κ , which are much more relevant than β and δ to a clinical audience, should be shown.

Line 231: What is this 5.5% result? Can you make it clearer that this is the observed level of resistance? And that the next results are from the model?

Line 233: Here it would be good to emphasise that the 5.5% includes those with heteroresistant carriage.

There needs to be more analysis of the sensitivity of this model to the variation in the many unknown parameters and a description of the impact of this on the key results – the level of transmission vs. mutation.

Discussion

Line 255: can you add "in H. pylori infections"?

Line 276: How does this statement affect the assumption made in the model that transmission of resistant strains can result in heteroresistance from those infected with the wildtype strain? Did

the model fit to the homo / hetero ratio from the data?

Where is the discussion of the limitations of the model? The lack of uncertainty in the estimates and the uncertainty in other parameters and hence the impact on the results needs to be discussed.

Supplementary material 1

- why did the authors not consider the results from Balint et al to be representative of the whole population of the Central Hungarian region?
- First table – why did H pylori prevalence decline so much? Why is this data needed?
- Were these prevalence data really “fitted” or just applied to the model?
- How is the H pylori prevalence calculated in the second table?
- How does the extrapolation work for the more recent birth cohorts? Did you fit a trend?
- The last sentence of this material suggests that all this work was just to get at an average population H pylori prevalence? This needs to be made clearer at the start. How likely is this to be correct? Are there studies from other countries? What is the impact on the results of varying this result?
- How was this overall prevalence split into initial conditions for uninfected / heteroresistant / resistant?
- What are the coverages of the Hungarian population by these two regions (Budapest / Pest county)?

Supplementary material 2

- There is a lot of unneeded overlap here with the main text.
- What are the initial conditions of the model in terms of natural history states? Or do you start with everyone uninfected and move to equilibrium with the conditions now?
- It is unacceptable that there are no references in this. Many new parameters are stated and their context should at least be discussed if there are no estimates from the literature. E.g. is there any evidence for fitness costs to resistance in H pylori? Where do the ranges in table A2 come from?
- Is resistance in H pylori only mutational? This is what is suggested here but transformation can also occur – is this important for clarithromycin / macrolide resistance? What impact would this have on the observed / modelled dynamics?
- As in the main text – there should be a link between the spontaneous mutation rate and the fraction of exposed wild type individuals that become heteroresistant upon macrolide exposure
- There needs to be a description of why a deterministic system was chosen. What are the limitations of this?

Limitations section

- What other types of infection could be? Shouldn't the “categorised all strains” be “infection states” not “strains”?
- Do you have any data on repeat exposure?
- “the emerged resistant strain cannot completely wipe out the already established wild-type population”; could the authors further discuss this assumption in light of experimental evidence for the importance of horizontal gene transfer by transformation in H. pylori, and the fact that H. pylori infection generally persists for life except if eradication therapy is undertaken?
- The steady state assumption is a big one – how can you be sure that it is correct?
- This section (or somewhere else in the paper) needs to explore how these assumptions affect the model output not just be statements. For example, what might be the impact of heteroresistant dual transmission? Would it make your estimates under or over-estimates of transmission?
- As mentioned in the main text – why does the model not include age and gender stratifications? They had a big impact on resistance levels.

Calculations

- Remove A4 from the heading?
- Can you provide a reference for the mortality statistics? I assume that you just used the most recent mortality rate and kept this the same as the birth rate to keep a steady population size but

you need to state exactly what you did.

- Table A2 – where are the references? It needs to be clear where these numbers came from as this works need to be supported and reproducible.
- I would have thought that the antibiotic usage rates could have been taken from the data used in the main text?
- H pylori prevalence in supplementary 1 was said to be 40.3% - why are you using a range now? And how?
- The relationships in the Cj statements need to be supported by the calculations written out clearly e.g. number of people in this state / total number of people
- For C1 I think the reference should be to Table 3.
- How were the feasible ranges in Table A2 decided upon?
- What does "close" to an equilibrium mean? And "with some accuracy"?
- How do you know that you have a unique equilibrium state?
- What does the value of alpha tell us about the conditions in Cj? that they do / do not hold with your parameters? Seems that this leads to a large bias towards parameters that satisfy the equilibrium condition but not the data on the levels of wild type / resistant etc?
- Can you link to supplementary material 1 for the 40.3% H value?
- Can you perform a sensitivity analysis using this method? How can you convey the uncertainty shown in the parameter ranges in Table A2? For example, could you fix one parameter and vary the others?
- Table A3 – what is the value for kappa? 0.5?

Conclusions

- Does the proportion of new infections depend on the population size? E.g. this is true if all are un-infected, but what if the majority of individuals are infected with the resistant strain?
- Linked to the above comment on sensitivity analysis, how did you generate the ranges for delta and beta? Did you fix kappa at different values and minimize W again? Could you do this for other important parameters?
- So this 64% is at equilibrium and is dependent on the population values e.g. 5% of the population have experienced eradication treatment but are not infected. Is this realistic? These values all need to be stated (or at least those that drive the 64%, un, inw) and explored for clinical / epidemiological validity.
- Where does the one-third are from over colonisation come from?

Supplementary material 3

This needs an introduction paragraph.

Reviewer #3 (Remarks to the Author):

The manuscript by Kocsmár et al presents a strategy to determine the existence of clarithromycin resistance that is independent from macrolide consumption. The manuscript is well-written and understandable, but I have several concerns related to specific aspects of the methodology. The mathematical modelling of the manuscript is largely based on data obtained using FISH. However, I found it really difficult to find details on the BACTFish H. pylori Combi Kit, both on the suppliers webpage or on the mentioned references (9 and 11). Reference 9 refers only to the H. pylori detection, whereas ref. 11 does refer to the clarythromycin method but for methodological details also links to the original publication in 9. Without a full disclosure of the methodology or at least strong data on the specificity and sensitivity of the method, it is not possible to assess the accuracy of the data provided. In the discussion the authors refer references 25 and 26 to support the accuracy of FISH-based data, but were these two studies conducted in the exact same conditions? The original probes appear to be from Russmann, but details on the hybridization and washing solutions are lacking. Also, are all point mutations associated with CLA resistance picked by the probes? Was the hybridisation carried out without any fixation or permeabilization solution?

Lines 83-88: How many tissue sections from each patient were used for the study? From which section(s) of the stomach? If the authors only analysed one section of the stomach, how likely is it for other strains of *H. pylori* to be present and how would this impact the conclusions of this study?

Lines 108-110: Please provide details of the wavelengths of the mentioned filters

Figure 1 – Please include scale bars. Also, out of curiosity, it would be expectable that microbial cells would also be labelled blue by DAPI. Was this removed with image processing?

Reviewer #4 (Remarks to the Author):

In this manuscript Kocsmar et al. report on a study of primary and secondary *H. pylori* resistance against clarithromycin in Hungary.

This is an interesting study with a large number of patients, using fluorescence in situ hybridization (FISH) to detect *H. pylori* and its resistance, a technique which seems to provide a better detection of heteroresistance compared to other methods, and where information on a previous macrolide treatment was available through a national data base. They found a primary resistance rate of 13.3% in so-called naive patients according to the usual definition, and 5.5% in patients who were never prescribed macrolides. They also applied a mathematical model to follow the dynamics of clarithromycin resistant (ClariR) strains and claim that 2/3 come from the transmission of these ClariR strains and 36% from spontaneous mutations.

While it is difficult for a non-specialist to understand this model, a limit lies in the fact that a few clones of ClariR *H. pylori* may be present in a stomach following a spontaneous mutation (occurring by chance) and remain undetected until they reach a certain proportion of the whole population. It would be interesting to know the limit of detection of ClariR with the FISH method and how it affects the modeling.

The finding that non-eradication clarithromycin therapy leads to more ClariR than other macrolides is also in contradiction with a European study on this topic where the best fit between *H. pylori* ClariR and macrolide consumption was for long acting macrolides and not for clarithromycin.

It is not clear why a different therapeutic strategy should be proposed for heteroresistant vs homoresistant *H. pylori*.

It would be nice to present the resistance rate in females also for the so-called primary resistance and for those who were never prescribed macrolides.

We thank the Reviewers for the very detailed evaluation of our paper and the critical notes. We have really found the referees' comments useful. Based on these, we have substantially changed our mathematical model and revised our manuscript. This included a fundamental change in the determination of the values of model parameters in order to reduce the uncertainty of the model outputs and to ensure the correct sensitivity analyses. For this, instead of letting the values of parameters vary in relatively broad ranges, we have determined a fixed value for the maximum number of the input parameters as precisely as it was possible, based on the cohort data and/or data of published studies. This new approach addresses the questions and concerns raised about the mathematical model. From this revised model, we obtained somewhat different results highlighting the more dominant role of the transmitted resistant strains in the primary clarithromycin resistance of the *H. pylori* infections. We also answered the questions on the clinical relevance of the results. Although our results address several clinical questions (e.g. which age- and sex-related subgroups need preliminary susceptibility test) but the mathematical model was developed to

investigate the clarithromycin resistance on the population level. Therefore, it is not appropriate to use as a "risk calculator" for assessing the individual risk to have resistant *H. pylori* infection but applicable for testing different therapeutic strategies (e.g. for establishing new recommendations for clinical guidelines). However, considering the reviewers' critics and recommendations, we have integrated such an adaptation of our model in this paper. We have modelled the situation of withdrawing macrolides from the non-eradication-purpose treatment indications, which indicated a significant long-term benefit from the decreasing clarithromycin resistance in the population. We have also addressed the notices and concerns raised about the used susceptibility test method. Accordingly, we have described the FISH method in greater detail and added a new section about the limitations of the study, including both the model- and the FISH method-related limitations.

POINT-BY-POINT RESPONSES

REVIEWER #1 (REMARKS TO THE AUTHOR):

This is a large retrospective observational study of Clarithromycin resistance in *H.pylori* strains in a region in Hungary. Fluorescent in situ hybridization of gastric biopsies was used to determine Clarithromycin resistance strains in the population, including mixed strains (some resistant, some not) in the same patient (hetero resistance).

Because each patient's history of macrolide intake is available through a prescription database, the authors were able to establish Clarithromycin resistance in individual patients and develop a mathematical model to describe how many of the strains were linked to prior Clarithromycin usage by the individual and how many were not.

The novelty of the paper is that a mathematical model was developed that was able to determine how many of the Clarithromycin resistant strains were due to the development of resistance in vivo secondary to macrolide usage for other infections, and how many were due to the transmission of preexisting Clarithromycin resistant strains.

The study has been carefully performed and well described.

We thank the Reviewer for the careful review and comments on our study which greatly contributed to improve the manuscript.

I. However, the clinical significance of utilization of such a model is unclear as the decision to use a Clarithromycin based regimen or not in any population is primarily based on the background clari-resistance in that population rather than in an individual. However, classical microbiological and molecular techniques are now available to determine Clarithromycin in an individual. With these tools, it is unclear how the mathematical model here adds to how antibiotic choice is determined.

We agree with the reviewer that the most widely used approach for planning the eradication regimen is the consideration of the population Cla-res level. Our results suggest that better approach to determine those population subgroups – by considering the age and gender-related differences – in which the Cla-res level is high or low. However, recently, more and more evidence and experts call attention to the significance of the individual resistance testing. In line with this, the Maastricht V Consensus Report states that "It is recommended to perform clarithromycin susceptibility testing when a standard clarithromycin-

based treatment is considered as the first-line therapy, except in populations or regions with well documented low clarithromycin resistance (<15%).”¹ However, it was interpreted in many countries with high Cla-res rates in such a way that the gastroenterologists discontinued the conventional Cla-based triple therapy and started using another eradication protocols (most frequently one of the bismuth-containing combinations) instead of introducing the susceptibility testing. The problem of this approach is that the susceptibility-guided clarithromycin-containing conventional triple therapy can achieve just as good or even better eradication rates than the other non-targeted anti-*H. pylori* regimens but with better side-effect profile, especially regarding the long-term gut microbiota-changing effects.² Our mathematical model was developed to investigate the clarithromycin resistance on a large cohort. Therefore, it is applicable for testing different therapeutic strategies on population level (e.g. working out recommendations for the guidelines) but it is not appropriate to use for the estimation of the risk of resistance and determination of the antibiotic choice on an individual level. However, considering the reviewers’ critics and recommendations, we have integrated such an adaptation of our model in this paper. In this, we have modeled the situation when the non-eradication use of the macrolide antibiotics is banned. This has shown that the increasing tendency of clarithromycin resistance rate of the population will be significantly reduced in the long term while by keeping up the current situation (with constant level of macrolide consumption), the model predicts a more intensely increasing rate of the clarithromycin resistance. This is in good agreement with the Taiwan experiences where the Bureau of National Health Insurance of Taiwan issued a reimbursement regulation in 2001 to restrict the use of any antibiotics in patients with upper respiratory infection which was likely to be caused by viruses.³ Since the clarithromycin was frequently used in this indication, the prevalence of the primary clarithromycin resistance has been stabilized and remained as low as 11.2% in Taiwan.

Other Comments:

1. Page 6, first paragraph. How many cases had multiple biopsies and discrepant results?

According to the clinical gastroenterological protocols, multiple biopsy samples were obtained in each case. However, as a consequence of the clinical sample collection practice of the study period, the great majority of these samples arrived in the same collection tube for histopathological work-up. Accordingly, these were embedded in the same paraffin block. Samples from the antral and oxyntic gastric compartments were distinguishable during the routine histological investigation based on the expertise of the pathologists. However, it was not a requirement during the evaluation of *H. pylori* diagnostic stains (Giemsa and/or IHC) / FISH. The final Giemsa/IHC and FISH result was indeed a summary of all the staining patterns observed in the individual tissue samples of the same section. Therefore, we have detailed information about the Giemsa/IHC and FISH results of the individual tissue samples originating from different sites of the same stomach only from 346 cases in this cohort. This means that two different sites were sampled in 306 cases, three sites in 39 patients and four sites in one case. Heteroresistant *H. pylori* infection was found in 34 of these patients with biopsy samples from two sites in 28 cases and from three sites in six cases. From more recent periods, as the protocols have changed meanwhile, we have significantly more samples collected separately from the different gastric compartments of the same patient. Accordingly, we have modified the “Diagnosis of *H. pylori* infection” part by inserting the next:

*“According to the clinical sample collection practice of the study period, the mapping biopsy tissue samples of an individual arrived in the same collection tube for histopathological work-up in majority of the cases (4398/4744). Accordingly, these were embedded in the same paraffin block. Samples from the antral and oxyntic gastric compartments were distinguishable during the routine histological investigation based on the expertise of the pathologists. The *H. pylori* diagnostic stain results were summarized considering the staining*

patterns observed from all the individual tissue samples of a given patient, independently whether these were embedded in the same or separate tissue blocks.” (line 363-369)

2. Page 6, paragraph 2. Over how many prior years does the pharmacy database record prescription usage?

The prescription database of the National Health Insurance Fund Management of Hungary is available since the year 2000. As the study period has started from January 01, 2005, at least five prior years of prescription history were considered in each case. To discuss this issue, it is worth taking into consideration the results of a publication of Mégraud et al.⁴ They determined antibiotic (including clarithromycin) resistance rates of *H. pylori* from April 2008 to June 2009 in 18 European countries. They supposed that the chronic character and unknown time of acquisition of *H. pylori* infection justify considering total cumulative outpatient antibiotic use data in the analysis in addition to single yearly use. They found that the fit of their model for linking the antibiotic use and proportion of resistant *H. pylori* strains improved along with the cumulative use of antibiotics over the years between 2001 and 2008. However, the best fit between macrolide and quinolone use and the proportions of clarithromycin- and levofloxacin-resistant *H. pylori* isolates was obtained by using the yearly antibiotic use data for 2005. Summarizing this, considering the longer time of the macrolide use history is the better but at least data from the previous 3 years (of the susceptibility test time) seems to be essential. Accordingly, we believe that our method considering as long anamnestic period as available but at least a five-year history of the macrolide use is an appropriate approach in our study.

Accordingly, we added the next sentence to the “*Analysis of prior antibiotic consumption*” section of the “*Material and Methods*”: “*Their prescription database is available since the year 2000.*” (line: 416)

3. Page 8, figure 3. Are any of these differences across age groups statistically significant?

Yes, significant differences were found in the 40-49, 50-59 and 60-69 age groups between the female and male resistance rates. (20-29: $p=0.18412$; 30-39: $p=0.14604$; 40-49: $p=0.00048$, RR 0.682-0.880, OR 0.329-0.753; 50-59: $p<0.00001$, RR 0.683-0.846, OR 0.342-0.673; 60-69: $p=0.01005$, RR 0.733-0.947, OR 0.453-0.910; $p=0.9229$; Fisher’s exact test).

We have inserted this into the Results section of the manuscript. (line: 126-130)

4. Page 9, first paragraph. “previously eradicated” should be changed to “previously treated”.

We have changed this expression accordingly. (line: 147)

REVIEWER #2 (REMARKS TO THE AUTHOR):

Review for: Primary and secondary clarithromycin resistance of *Helicobacter pylori*: role of macrolides and mathematical modelling of resistance independent of macrolide consumption, Kocsmar et al

This study combines epidemiological data analysis and mathematical modelling to elucidate the origin of resistance to the clarithromycin antibiotic in *Helicobacter pylori* bacteria. Notably, the authors aim to clarify the proportion of resistant infections which originate from the transmission of resistant bacteria from another person, and the proportion attributable to spontaneous mutation, as well as the correlation between clarithromycin-resistant *H. pylori* infections and previous exposure to clarithromycin or other macrolide antibiotics. This study is set in Hungary, where antibiotic prescribing is recorded for each

person, therefore allowing the authors to link previous antibiotic exposure and current *H. pylori* infection status.

The retrospective cohort analysis is clearly presented in the Methods and Results sections in the main text and leads to strong and potentially important conclusions on the link between macrolide consumption and risk of clarithromycin-resistant infection. However, some crucial assumptions made in the model are not discussed, which prevents the results of the model from being truly meaningful. In addition, some key information on *H. pylori* characteristics and epidemiology is missing from the Introduction section, which could cause confusion for the wider audience of Nature Communications.

We were asked specifically to look at the mathematical modelling included in this paper. The model structure itself is described well and is clear. However, we do not believe that the results are reported clearly – the headline results do not clarify the other results (e.g. equilibrium values for populations infected) that these results rely on. Single percentage values for the contribution of transmission are inappropriate and misleading. We do agree that a model such as this was an appropriate tool for investigating the question, but this was not made clear in the introduction nor the context for this model discussed.

We thank the Reviewer for this very detailed evaluation of our paper and the critical notes. Based on these precise remarks, we have fundamentally changed our mathematical model. This includes a more detailed determination of the parameters which can be calculated from the cohort or other independent sources, the examination of the role of the non-eradication-purpose clarithromycin and other macrolides, moreover, the potential beneficial effect of their withdrawal from the non-eradication use. According to this, our manuscript was broadened with new sections and new conclusions have been drawn by these.

Has such a model been attempted before?

To the best of our knowledge, no such (or similar) model has been attempted before. The model by Rupnow et al.⁵ predicts the trends of *H. pylori* prevalence and the associated diseases in the United States but does not include any antibiotic resistance parameter. Moreover, the high majority of the published studies on the clarithromycin resistance of *H. pylori*, does not include the distinction between the homo- and heteroresistant infections. Furthermore, our model is also unique from the aspect that instead of a steady state, it assumes a slow populational change which is much closer to the real population dynamics of the *H. pylori* infections.

How do the results compare to other *H. pylori* modelling literature?

Our results are not directly comparable with other models as partly detailed above. The main difference between our model and that one described by Rupnow et al.⁵ that their primary aim was to model the natural course of *H. pylori* infection and the development of the associated diseases while we model the population dynamics of the clarithromycin-resistant *H. pylori* infections including the homo- and heteroresistance. Although the basic population dynamical and transmission rules are the same in both studies, but their model does not include any antibiotic resistance parameter while our model omits such parameters like age and sex, which do not directly influence the clarithromycin resistance (the primary clarithromycin resistance is independent of the age and sex, while the age- and sex-dependence of the secondary resistance is related to the age- and sex-dependent antibiotic consumption trends, and the population macrolide consumption is included in the model as parameter *m*). Although the model created by Rupnow et al. was a major step forward in the better understanding of natural course and population dynamics of the *H. pylori* infection, however the increasing antibiotic resistance rates became the leading

problem of the subsequent period. Therefore, we aimed to model this phenomenon and identify the sources of the clarithromycin resistance in the population.

A big flaw is that no attempt is made in the main text to explore how the uncertainty in the model and parameters translates into uncertainty in the results. Overall, we found it hard to follow much of the analysis – especially in the supplementary. A clearer narrative is needed with more context.

Thank you for these comments, we have tried to make clearer which parameter is observed by the cohort data, based on publicly available data or assumed based on the literature. We have made sensitivity analyses and modified the model description in order to provide more detailed information about the necessarily occurring uncertainties and to make it clearer for the broader audience.

MAJOR COMMENTS

1) Some key information is currently missing in the Introduction. We feel that this should be included to make this relevant to the non-specialist audience of Nature Communications.

We thank the Reviewer for these critical notes. We have modified the Introduction accordingly. Please find below the revised issues with the indication of the corresponding lines of the manuscript.

- Firstly, it should be mentioned that *H. pylori* is estimated to be present in more than 50% of the global population (Hooi, J. K. et al. (2017). Global prevalence of *Helicobacter pylori* infection: systematic review and meta-analysis), (line: 46-47)

- ...and only causes an asymptomatic infection in the vast majority of cases (Kusters, J. G., van Vliet, A. H., & Kuipers, E. J. (2006). Pathogenesis of *Helicobacter pylori* infection). This should be clearly mentioned to avoid any reader confusion in the rest of the paper regarding the reported prevalence of infection. (line: 47-48)

- Secondly, it should also be mentioned that *H. pylori* infection usually persists for life, and that individuals can only clear the infection thanks to eradication treatment (Kusters, J. G., van Vliet, A. H., & Kuipers, E. J. (2006). Pathogenesis of *Helicobacter pylori* infection). This context is important to understand why natural clearance of the bacteria is not part of the mathematical model. (line: 51-52)

- Lastly, the capacity for *H. pylori* to perform horizontal gene transfer by transformation should be mentioned (Israel, D. A., Lou, A. S., & Blaser, M. J. (2000). Characteristics of *Helicobacter pylori* natural transformation). Although the dynamics of antimicrobial resistance gene transfer by transformation are admittedly still unclear there is good experimental evidence that this can occur at a significant rate and must therefore be mentioned, since this factor could also contribute to increasing rates of Cla-res.

We have detailed the main clarithromycin resistance mechanism of the *H. pylori* in the Introduction and the potential role of the horizontal gene transfer in the spreading of the resistance. (line: 55-66)

2) There are multiple elements of the model that are currently unclear, and should be discussed and corrected. Importantly, the work as it is not reproducible despite having the model code (e.g. how were the parameters determined?). These are detailed in the list of revisions below, but broadly include:

- The parameterisation is not supported by sufficient calculation information or any evidence (e.g. reference to estimates from previous literature).

We thank the Reviewer for pointing out this critical flaw. In the revised manuscript, we provide detailed information about all input parameters in the Table 5 and Supplementary Material 1 and 2. However, we also provide sufficient information here for each parameter.

notation	parameter	input value output	Source
μ	demographic turnover	0.013	Hungarian Central Statistical Office, 2005-2013 Supplementary Material 2
η	developing heteroresistance by macrolide use	0.129	Supplementary Material 1
ρ_u	successful eradication	0.92	Supplementary Material 1
ρ_w	no effect of Cla-treatment	0.039	Supplementary Material 1
ρ_r	Cla-treatment-induced homoresistance	0.029	Supplementary Material 1
ρ_h	Cla-treatment-induced heteroresistance	0.012	Supplementary Material 1
θ	Cla-treatment rate	>0 0.00295584	Model output
m	rate of macrolide consumption	0.00889	Supplementary Material 1,2
ω	spontaneous mutation rate	0.0000155	References: Kennemann et al. ⁶ Linz et al. ⁷
β	transmission rate	>0 0.019	Model output
δ	relative transmission parameter	>0 0.72	Model output
λ_r / λ_w	Force of infection with respect to resistant/ wild-type	>0 0.00040911 (λ_r) >0 0.00676164 (λ_w)	Model output
H	H. pylori infection prevalence	0.409	Supplementary material 1
$\frac{dH}{dt}$	Prevalence trend	-0.002	Reference: Hooi et al. ⁸

Kennemann L, Didelot X, Aebischer T, et al. Helicobacter pylori genome evolution during human infection. Proceedings of the National Academy of Sciences. 2011;108(12):5033-5038. doi:10.1073/pnas.1018444108

Linz B, Windsor HM, McGraw JJ, et al. A mutation burst during the acute phase of Helicobacter pylori infection in humans and rhesus macaques. Nat Commun. 2014;5:4165. doi:10.1038/ncomms5165

Hooi JKY, Lai WY, Ng WK, et al. Global Prevalence of Helicobacter pylori Infection: Systematic Review and Meta-Analysis. Gastroenterology. 2017;153(2):420-429. doi:10.1053/j.gastro.2017.04.022

- Demographic turnover (μ)

For the demographic turnover (i.e. death and birth rates), we considered the average mortality rate of Hungary in the period 2005-2013, taken from the database of the Hungarian Central Statistical Office (KSH).

https://www.ksh.hu/docs/eng/xstadat/xstadat_annual/i_wdsd006a.html

https://www.ksh.hu/docs/eng/xstadat/xstadat_annual/i_wdsd006b.html

- Developing heteroresistance by macrolide use (η)

The prior macrolide use data obtained from the National Health Insurance Fund Management of Hungary. Their data protection / privacy policy allows them to provide only aggregated data. Therefore, they collected the information from the health insurance databases and analysed the individual patient's prior macrolide use. Then, based on this, they provided us the macrolide use data in the case of those groups and subgroups about which it was previously requested. Consequently, we have no direct access to the macrolide use data for each patient of our cohort and a re-analysis of these from new aspects would require very long processing time in the National Health Insurance Fund Management of Hungary (the original analyses took about two and a half years). Since the study was primarily designed to find the macrolide-naïve (primary resistant) infections, thus, from practical considerations, the "any prior macrolide use" was investigated during the data collection. Thus, obtaining the homo- and heteroresistant cases separately from each subgroup of the clarithromycin resistant infections was not included into our primary goals, therefore, unfortunately, the macrolide use-related homo/heteroresistance ratio is not available in detail for each subgroup. However, we can calculate this "Developing heteroresistance by macrolide use" (corresponding to the non-eradication-purpose macrolide use) parameter indirectly for our cohort data by considering one of our previous publications.

In this published study, we have investigated the homo/heteroresistance ratio among patients acquired the secondary resistance by unsuccessful clarithromycin-based eradication attempt.⁹ We found 70%/30% homo/heteroresistance ratio (23 homo- and 10 heteroresistant cases) among the unsuccessful eradication-related clarithromycin-resistant cases.

In the present cohort, we have observed 247 clarithromycin resistant cases with history of previous clarithromycin-based eradication attempt (Table 4) and 297 macrolide-exposed heteroresistant cases (secondary heteroresistance, Supplementary material 1, Table 1.). If we consider the 70%/30% ratio among the 247 patients with unsuccessful eradication treatment, then it will result in 173 homo- and 74 heteroresistant cases. Considering the 74 unsuccessfully eradicated heteroresistant cases and subtracting these from the 297 any macrolide exposed heteroresistant ones, it results in 223 heteroresistant cases out of the total 429 individuals of the non-eradication-related macrolide-exposed group. Based on this, the homo-/heteroresistance ratio among these patients is 48%/52% (206 homo- and 223 heteroresistant). Considering that the 429 resistant patients run to the 24.8% of the total 1731 non-eradication-related macrolide exposed persons, the "Developing heteroresistance by macrolide use" parameter would be 12.9% (0.129) based on the 48%/52% homo-heteroresistance ratio.

- Successful eradication (ρ_u)

This parameter (rate of the successful *H. pylori* eradication with clarithromycin-containing regimen) corresponds to the success rate of the clarithromycin-based eradication therapy in Cla-susceptible infections. To set this, we have read up studies with a susceptibility-guided approach in which the clarithromycin susceptible cases were treated by the classic triple therapy as well as intention-to-treat analysis was done. Based on these published data, it was set to 0.92, meaning that 92% of the clarithromycin-susceptible infections can be eradicated by the clarithromycin-containing regimens.

- No effect of Cla-treatment (ρ_w)

This parameter corresponds to the rate of cases in which the eradication treatment failed, the host remained infected and the remaining *H. pylori* population is clarithromycin-susceptible. This can be calculated from the previous parameter ("Successful eradication"). Since the rate of the successful *H. pylori* eradication with clarithromycin-containing treatment is 92%, the remaining 8% correspond to the 481 previously unsuccessfully eradicated patients (Table 2, first section, "Eradicated" / "Total"). This group is composed of the 234 clarithromycin susceptible cases (this corresponds the "No effect of Cla-treatment" parameter) as well as the 247 clarithromycin resistant cases which can be further divided into homo- and heteroresistant subgroups (Cla-treatment-induced homoresistance / heteroresistance; see below). As the 8% is shared in the ratio of 234:247, the "No effect of Cla-treatment" parameter was set to 3.89% (0.0389).

- Cla-eradication-induced homoresistance (ρ_r) / heteroresistance (ρ_h)

These parameters correspond to the rates of cases in which the clarithromycin-containing eradication treatment failed, the host remained infected and the remaining *H. pylori* population is either homoresistant or heteroresistant to clarithromycin. As we have discussed above in connection with the parameter "Developing heteroresistance by macrolide use", we have investigated the homo/heteroresistance ratio among patients who acquired the secondary resistance by unsuccessful clarithromycin-based eradication attempt in one of our previous studies.⁹ We found 70%/30% homo/heteroresistance ratio (23 homo- and 10 heteroresistant cases) among the unsuccessful eradication-induced clarithromycin resistant cases. Moreover, as we have deduced above, the group of the 8% previously unsuccessfully eradicated patients is shared in the ratio of 234:247 (no effect of Cla-treatment / clarithromycin treatment-induced resistant cases), consequently the clarithromycin-induced resistance (247 cases) corresponds to 4.11% . Accordingly, we divided this further in the ratio of 23:10, resulting in 2.86% (0.0286) Cla-treatment-induced homoresistance and 1.25% (0.0125) heteroresistance.

- Cla-eradication rate (θ)

This parameter corresponds to the annual rate of the clarithromycin-based *H. pylori* eradication treatment in the whole population (of Central Hungary). Although the rate of previous clarithromycin-containing eradication treatment is available for our cohort, we did not use it for setting this parameter of the model. Namely, our cohort represents patients with gastrointestinal complaints who were treated mainly in ambulance settings by gastroenterologists, so we did not accept the rate of the previous eradication therapy of these patients as representative of the whole population. Rather, these gastroenterology patients represent a special subset of the population displaying the highest rate of the previous eradication attempts with clarithromycin-containing regimen. Therefore, we do not constrain this parameter and infer its value from the model calculations. (For further details see the Supplementary Material 2.)

- Rate of macrolide consumption (m)

This parameter corresponds to the annual rate of cases undergoing non-eradication-purpose macrolide (non-eradication-purpose clarithromycin or other macrolide antibiotic) treatment. In the case of this parameter, we have accepted the non-eradication related macrolide consumption rates observed in our cohort as representative of the whole Central Hungarian population after fitting the female and male age group macrolide-consumption data to the age groups of the Hungarian population census 2011. (It is detailed in the Supplementary Material 1 including the S1 Table 2 which table is also inserted below.)

Gender	Age groups	Cohort total (n)	Cohort Macrolide exposed (n, %)	Cohort Eradicated (n, %)	Population total (n)	Population Macrolide exposed (n, %)	Population Eradicated (n, %)
Male	0-19	18	10 (55.56%)	2 (11.11%)	292,440	162,466.67 (55.56%)	32,493.33 (11.11%)
	20-29	116	46 (39.66%)	1 (0.86%)	191,642	75,995.97 (39.66%)	1,652.09 (0.86%)
	30-39	256	81 (31.64%)	13 (5.08%)	257,633	81,516.69 (31.64%)	13,082.93 (5.08%)
	40-49	311	76 (24.44%)	33 (10.61%)	188,944	46,172.81 (24.44%)	20,048.72 (10.61%)
	50-59	513	149 (29.04%)	50 (9.75%)	177,840	51,653.33 (29.04%)	17,333.33 (9.75%)
	60-69	444	153 (34.46%)	38 (8.56%)	152,178	52,439.72 (34.46%)	13,024.24 (8.56%)
	70-	377	119 (31.56%)	45 (11.94%)	117,358	37,044.04 (31.56%)	14,008.25 (11.94%)
Female	0-19	34	17 (50.0%)	4 (11.75%)	278,190	139,095.00 (50.0%)	32,728.24 (11.75%)
	20-29	136	58 (42.65%)	4 (2.94%)	194,402	82,906.74 (42.65%)	5,717.71 (2.94%)
	30-39	280	132 (47.14%)	30 (10.71%)	264,389	124,640.53 (47.14%)	28,327.39 (10.71%)
	40-49	429	176 (41.03%)	54 (12.59%)	195,133	80,054.56 (41.03%)	24,562.20 (12.59%)
	50-59	686	300 (43.73%)	89 (12.97%)	210,924	92,240.82 (43.73%)	27,364.78 (12.97%)
	60-69	542	218 (40.22%)	65 (11.99%)	203,754	81,952.72 (40.22%)	24,435.44 (11.99%)
	70-	602	196 (32.56%)	53 (8.80%)	221,689	72,177.81 (32.56%)	19,517.47 (8.80%)
Total		4744	1731 (36.49%)	481 (10.14%)	2,946,516	1,180,357.39 (40.06%)	274,296.11 (9.31%)

- Spontaneous mutation rate (ω)

This parameter (ω) was determined to be 1.55×10^{-5} on the basis of the publication from Kennemann et al.⁶ and Linz et al.⁷

- Transmission rate (β)

This parameter was not previously determined a-priori, but inferred from the model calculations.

- Relative transmission fitness (δ)

This parameter was not previously determined a-priori, but inferred from the model calculations.

- Force of infection with respect to resistant (λ_r)/ wild-type (λ_w)

This parameter was not previously determined a-priori, but inferred from the model calculations.

- *H. pylori* infection prevalence (H)

This parameter was calculated from the results of a previous publication. Buzás et al.¹⁰ collected the prevalence data from the population of the Central Hungary Region and their prevalence data grouped according to birth cohorts allowing us to fit these to the age-grouped population data of the Hungarian population census from the investigated years. This fitting was necessary because the birth cohorts from this population census and from the study of Buzás et al.¹⁰ both have 10-year / 5-year ranges (periods) but these were “shifted” relative to each other. Since no data were available for the 2007–2011, 2002–2006 and 1997–2001 birth cohorts, we extrapolated these prevalence rates by fitting a linearly increasing trend by age. We considered this solution as the closest estimation of the real prevalence data of the Central Hungary Region. We provide further details in the answers to the reviewer’s comments to the Supplementary Material 1 and in the revised Supplementary Material 1.

- A wider sensitivity analysis should be included to cover the key results as well as all unknown parameters. We thank the reviewer for this critical note. We have performed further sensitivity analyses as it has described in detail in the revised Supplementary Material 2.

- The effect of previous consumption of macrolides for non-eradication purposes. The model assumes that the effect would be the same regardless of whether these macrolides are clarithromycin or not, but the data in table 4 seems to suggest otherwise (previous users of clarithromycin for non-eradication purposes have a higher risk of Cla-res infection than users of other macrolides)

We did not consider the non-eradication-purpose clarithromycin and other macrolide use in the model as separate parameters since this would unnecessarily increase the complexity of the calculations resulting in a higher number of uncertain parameters. On the one hand, in most of the non-eradication purpose therapeutic regimens, the total dose of the macrolides is lower than in the eradication protocols. In addition, their effectiveness is highly affected by the number and length of the therapeutic attempts, about which we do not have information. Taking all this into consideration, we decided not to further divide the “non-eradication-purpose macrolide” group in order to reduce the uncertainty arising from precarious assumptions.

- Individuals infected by either one of the susceptible or resistant strains are considered to be fully susceptible to over-infection by the other strain. This is a strong assumption, yet it is not currently discussed or the subject of a sensitivity analysis.

We thank the Reviewer for this note. We have very limited data on this question. One of them is the publication from Mansour et al.¹¹ where the authors found 11 multiple infections (coinfection by different *H. pylori* strains) out of 42 *H. pylori* infected cases. Only susceptible strains were found in nine out of the 11 multiple infections while both strains (2/2) were clarithromycin-resistant in one case and three out of the four strains were resistant in another case. In a further publication by Seo et al.¹² they found ten multiple infections out of 80 *H. pylori*-infected patients. Only susceptible strains were detected in eight cases while both strains were clarithromycin resistant in one case and one out of the two strains were resistant in a further case. These studies strongly support the theory that overinfection by and coexistence of different strains of discrepant resistance status is possible. The rate of the clarithromycin resistance among the multiple infections did not differ significantly from the *H. pylori* prevalence of the given population. However, regarding the low number of studied patients, moreover the lack of the data about the transfer mechanisms, rates and fitness, we cannot say more than overinfection by either susceptible or resistant strains might be possible and to assume that it can happen by equal chance (since there is no additional information which confutes this).

- Individuals infected only by the susceptible strain can become infected by the resistant strain as a consequence of eradication treatment. The mechanism by which this can happen is unclear and might

require either rethinking whether these individuals are actually hetero-resistant when receiving treatment, or linking the probability of resistant arising in this scenario to the mutation rate of the bacteria.

We politely disagree with the Reviewer since we did not state that individuals could become infected by the resistant strain as a consequence of eradication treatment. As we have discussed above, overinfection by a resistant strain is possible but not as a consequence of the eradication therapy. However, according to the Supplementary Material 4, clarithromycin-resistant bacteria continuously appear in the susceptible *H. pylori* populations by spontaneous mutation. However, the rate of successfully emerging Cla-res clones is low in the absence of clarithromycin-induced selection pressure. From this point of view, it is indeterminable whether there is any resistant *H. pylori* present in the population at a given time point or not (if one mutant bacterium is actually present it is practically undetectable with most of the used techniques), therefore these infections are generally categorized as susceptible. Of course, external stress conditions like the selection pressure by antibiotics will increase the mutation rate.¹³ This will lead to an accelerated adaptation and when a resistance mutation occurs, this is able to spread in the population by horizontal gene transfer. The external stress condition also means an increased mutation rate which has also been described during the acute phase of *H. pylori* infection by Linz et al.⁷ Therefore, this adaptation-related mutation rate should be theoretically higher than the mutation rate observed from the model which is more likely a long-term mutation rate which is related to the chronic phase of the *H. pylori* infection. This is the reason why we do not link the mutation rate (ω) of the model to the appearing macrolide-caused resistance but use the "m" parameter instead of this.

- Hetero-resistant individuals can transmit either the susceptible or resistant strain, governed by the parameter κ . The consequence of changing κ for the main results presented in this manuscript (the origin of primary clarithromycin resistance) is not currently tested.

From the model we infer the forces of infection. These are consistent with many κ, δ pairs, where the parameter κ expresses the probability that a heteroresistant infector exposes the infectee by the susceptible or the resistant strain, and δ is the relative transmission parameter of the resistant strain. We explore the full possible range of κ (0-1) to obtain all possible δ , but the other main results do not depend on that, they depend only on the forces of infection.

- The model assumes the system is at steady state despite evidence showing the dynamic nature of *H. pylori* infection. What evidence is there to support this? This should be mentioned in the main text.

We politely disagree with the Reviewer since our model was designed for slow epidemiological change instead of steady state. We have clarified this point in the Supplementary material 2.

MINOR COMMENTS

Abstract

Line 30: Do you mean that clarithromycin resistance is the major cause of unsuccessful treatment *for *H. pylori**? In that case, the sentence should be rephrased, as it could be interpreted as "clarithromycin resistance in *H. pylori* is the major cause of unsuccessful treatment attempts for all bacteria".

We have rephrased this sentence accordingly. (line: 30-31)

Can you make it clear here that clarithromycin is a macrolide? And that it is the drug used for eradication treatment? It was unclear to me what eradication-naïve meant without this information.

We have inserted the following sentence into the abstract: "*Clarithromycin is a macrolide antibiotic widely used for eradication of the Helicobacter pylori (H. pylori) infection.*" (line: 29-30)

E.g. in Line 41: could it perhaps be rephrased to "clarithromycin-naïve (for purposes of eradication) patients?"

We have removed this sentence from the abstract since a significant shortening was needed.

Line 39: To avoid confusion, I recommend rephrasing this sentence to "The overall rate of clarithromycin resistance amongst all *H. pylori* infections was 17.2%"

We have removed this sentence from the abstract since a significant shortening was needed.

Line 41: the 64/36% split is a result from the modelling – can this be included in this sentence?

We have rephrased this sentence accordingly. (line: 34-35)

Line 47: what are these groups where is it above 15%?

We have removed this sentence from the abstract since a significant shortening was needed.

Line 47: as mentioned above, I think you need to be clear what is a modelling result and what comes from the epidemiological cohort

This has been clarified accordingly. (line: 34-35, 37-39)

Introduction

Line 55: highest rank where?

The source of this information is the next:

"Among bacteria typically responsible for community-acquired infections, the highest ranked were clarithromycin-resistant *Helicobacter pylori* at 44.8% (10.1) and fluoroquinolone-resistant *Campylobacter* spp at 41.0% (7.8), *Neisseria gonorrhoeae* at 35.8% (8.9), and *Salmonella typhi* at 37.6% (9.2)."¹⁴

Reference:

Tacconelli E, Carrara E, Savoldi A, et al. Discovery, research, and development of new antibiotics: the WHO priority list of antibiotic-resistant bacteria and tuberculosis. *Lancet Infect Dis.* 2018;18(3):318-327. doi:10.1016/S1473-3099(17)30753-3

Line 57: "*Helicobacter pylori*" -> "*H. pylori*" abbreviation could be used

We have changed this accordingly. (line: 53)

Line 60: What is the empirical triple therapy?

According to the Reviewer's recommendation, we have completed this sentence with the components of the empirical triple therapy. (line: 68-69)

Line 61: "15% of all *H. pylori* infections"

We have changed this expression accordingly. "*15% of all diagnosed H. pylori infections...*" (line: 74-75)

Line 64: Can you make clear that clarithromycin is a macrolide: e.g. cross resistance exists in macrolide resistance such that the use of other macrolides can lead to clarithromycin resistance?

We completed the text according to the Reviewer's recommendations. (line: 55-56, 77-84)

"Clarithromycin, a member of the macrolides family, is still the most potent antibiotic of the H. pylori eradication treatment protocols."

Line 68: Could you please provide one or two examples of "other purposes" why macrolides could be used? This would help to provide context on the magnitude of this problem.

We have inserted this into the text according to the Reviewer's recommendations. (line: 82-84)
"However, use of clarithromycin or other macrolides for other purposes (including treatment of upper and lower respiratory tract infections, pelvic inflammatory disease, skin and soft tissue infections), may also lead to Cla-res"

Line 70: Can heteroresistant populations be found by your lab method or are they hypothesised populations?

The used *H. pylori* FISH is able to identify simultaneous presence of the susceptible and resistant bacteria in the tissue sample (see Figure 1 C), thus identification of not only the interniche but also the intraniche heteroresistance is possible. Rate of the heteroresistant infections in the different age groups is displayed in the last column of Table 1.

Line 74: italics and abbreviation could be used here for Cla-res *H. pylori*

We have changed the criticised expression according to the Reviewer's recommendations. (line: 89)

Line 74: could the authors please clarify whether *H. pylori* resistance has already been modelled? The reference included is not specific to *H. pylori*. How does this work add to the existing *H. pylori* work? It needs to be clearer what the model was for. "to analyse the population dynamics" is not sufficient. What could not be determined from the epidemiological information?

To our best knowledge, the clarithromycin resistance of *H. pylori* has not been mathematically modelled so far. Although an analytical framework was created to model the *H. pylori* transmission dynamics and its subsequent disease progression in order to estimate the future trends of *H. pylori* and associated diseases in the United States⁵, however, this model does not consider the antibiotic resistance to be a factor influencing the natural course of *H. pylori* infection. In the review paper from Niewiadomska et al. the authors pointed out the problem of lacking antimicrobial resistance (AMR) models for certain infections: "Importantly, we were unable to find any published AMR models for the following serious threats: *Helicobacter pylori*, *Haemophilus influenzae*, fluconazole-resistant *Candida*, clindamycin-resistant group B strep, and erythromycin-resistant group A strep. While mathematical transmission models do exist for wild-type *H. pylori* [102], *H. influenzae* [103], and *Candida parapsilosis* [104], we are not aware of any models for resistant strains, which may have different transmission parameters than susceptible strains."¹⁵ Therefore, our study is a significant achievement in this field since this work models not merely the transmission dynamics of the clarithromycin resistant *H. pylori* but it also considers the distinct nature of the homoresistant and heteroresistant infections. Our primary goal was to determine the ratio of the transmitted and mutation-originated resistant infections which became possible by establishing this model. Furthermore, this has also enabled us to model the withdrawal of the macrolides from the indications other than *H. pylori* eradication, which we have inserted into the revised manuscript. Our model is applicable for testing different therapeutic strategies on population level and thereby support the development of recommendations and new guidelines. So, these are what we cannot determine from the epidemiological information.

We have inserted these references into the text. (line: 89-92)

Materials and methods

We could not comment on the lab results and hence focused on the Mathematical modelling section only.

Line 148: It is unclear to what population you are referring in this first sentence – are these the patients across the study or in your model? In the model, the patients described in the rest of the study are not explicitly represented. Instead, the model infers the proportion of people infected by *H. pylori* in Central Hungary based on a previously published study.

We have modelled the *H. pylori* clarithromycin resistance in Central Hungary based on the clarithromycin-resistance data observed from our 4744 *H. pylori*-infected patients. Comparing this number of infected patients to the European studies on the *H. pylori* clarithromycin-resistance organized by EHMSG in which one centre per 10-20 million inhabitants represented the given population (e.g. Megraud et al 2013)⁴, we feel that our studied patients – which is one of the largest study groups in the literature – from 16 different requesting institutions represent the clarithromycin-resistance status of the population of Central Hungary well. The connection between our retrospective cohort and the model describing the population dynamics of the clarithromycin resistant *H. pylori* infections in Central Hungary is that several essential input parameters of the model were obtained from the cohort. However, our cohort cannot provide or has only limited information about certain aspects, such as the population-level prevalence of *H. pylori* or the average eradication rate in the whole population.

Line 148: Can you add a link to the supplementary material early on? Otherwise the description of the model and the notation used is very clear.

We thank the Reviewer for this positive note. We have completed the text accordingly. (line: 429-430)

Line 149: Are there any differences in the symptoms associated with *H. pylori* carriage? As in should we be thinking about those asymptomatic and symptomatic carriers? I assume treatment eradication exposure rates would then be higher for the latter? Is there a time delay – e.g. colonised and then progress to asymptomatic infection for this bacterium? By using the word “infected” here any link to symptoms is ignored which could impact on treatment exposure (and possibly transmission? E.g. higher bacterial load?).

The vast majority of the patients suffering from *H. pylori* infection remain asymptomatic.¹⁶ Carriage of *H. pylori* bacteria do not have specific symptoms. Notwithstanding, long-term carriage of *H. pylori* bacteria predisposes various clinical conditions (e.g. peptic ulcer or gastric cancer). Interestingly, due to the changed intragastric conditions in these cases, the bacteria are often present in reduced number and/or redistributed patterns on the gastric mucosa.¹⁷ However, as we have no detailed information on how the clarithromycin resistance status of the *H. pylori* infection can be related to the natural course of the related diseases or the asymptomatic carriage status, it was not possible to integrate such information in the model.

Line 152: The “m” index assumes that whatever the effect of macrolides on *H. pylori* resistance may be, it is the same regardless of whether these macrolides are clarithromycin or not. However, Table 4 indicates that the proportion of Cla-res is higher in individuals who used clarithromycin for non-eradication purposes (36.2%), compared to those who used other macrolides for non-eradication purposes (13.4%). Can you clarify how this discrepancy is taken into account in the calculation of the “ η ” parameter?

We did not consider the non-eradication-purpose clarithromycin and other macrolide use in the model as separate parameters. On the one hand, in most of the non-eradication-purpose therapeutic regimens, the total dose of the macrolides is lower than in the eradication protocols. In addition, their effectiveness is highly affected by the number and length of the therapeutic attempts, about which we do not have information. Therefore, including such an uncertain parameter would unnecessarily increase the complexity of the calculations and uncertainty of the model. Taking all this into consideration, we decided not to further divide the “non-eradication-purpose macrolide” group in order to reduce the uncertainty arising from precarious assumptions. Consequently, there is no separate “ η ” for the non-eradication-

purpose clarithromycin and the use of other macrolides but the "η" parameter refers to the whole non-eradication-related macrolide use group.

Line 157: Is there any evidence of a fitness cost to clarithromycin resistance in *H. pylori*?

In the study of Kanai and his colleagues¹⁸, they examined clonally related macrolide sensitive- and resistant *H. pylori* isolates by culturing under laboratory conditions. If equal bacterial masses of the resistant and sensitive isolates were mixed together, the ratio of the resistant clone was gradually declined by increasing passage number suggesting that the resistance-conferring point mutation caused a significant disadvantage on the bacterial fitness.

However, somewhat different results were observed in the in vivo experiment performed by Björkholm et al.¹⁹ They infected mice with susceptible and resistant isolates of the same strain in 1:1 ratio. When the clarithromycin-resistant mutant was selected in vitro, this was outcompeted by the sensitive parent in the mice at all three doses tested. However, when the susceptible and resistant bacteria were clinically isolated from the same patient before and after an unsuccessful eradication attempt, the authors found that all the possible outcomes of the competition (the susceptible isolate overgrows the resistant in the stomach, the resistant overgrows the susceptible, or equilibrium between the two isolates) can occur. The authors concluded that the biological / fitness costs of the resistance mutations can be reduced in clinical isolates by compensatory mutations.

Line 158: it is assumed that individuals infected by either one of the susceptible or resistant strain are equally likely to be infected by the other strain as non-infected individuals. This is a strong assumption that is not supported by any reference, or discussed in the paper. Could the authors please expand on this? This might require further sensitivity analysis with a new parameter controlling the relative susceptibility to over-infection of already colonised individuals compared to non-infected individuals.

We thank the Reviewer for this note. As we have discussed above, we have very limited data on this question. One of them is the publication from Mansour et al. where the authors found 11 multiple infections (coinfection by different *H. pylori* strains) out of 42 *H. pylori* infected cases.¹¹ Only susceptible strains were found in nine out of the 11 multiple infections while both strains (2/2) were clarithromycin-resistant in one case and three out of the four strains were resistant in another case. In a further publication by Seo et al. they found ten multiple infections out of 80 *H. pylori* infected patients.¹² Only susceptible strains were detected in eight cases while both strains were clarithromycin-resistant in one case and one out of the two strains were resistant in a further case. These studies strongly support the theory that overinfection by and coexistence of different strains of discrepant resistance status is possible. The rate of the clarithromycin resistance among the multiple infections did not differ significantly from the *H. pylori* prevalence of the given population. However, regarding the low number of studied patients, moreover the lack of the data about the transfer mechanisms, rates and fitness, we cannot say more than overinfection by either susceptible or resistant strains might be possible and to assume that it can happen by equal chance (since there is no additional information which confutes this).

Line 161: Does macrolide use never clear *H. pylori* carriage? Are there no uses of macrolide that could lead to this?

Some other macrolides are not acid-stable (e.g. erythromycin) or their bioavailability is inferior in comparison with the clarithromycin (azithromycin: 37% vs. clarithromycin: 55-70%) as well as dosage of clarithromycin in several non-eradication-purpose indications is half of the eradication dose, moreover, the eradication effectiveness is strongly dependent on the control of intragastric pH by the concomitant proton-pump inhibitor therapy.^{20,21} Consequently, there is only a minimum chance to eradicate the *H. pylori* unintentionally. Indirectly, based on the presence of mucosal atrophy and intestinal metaplasia without current *H. pylori* infection in certain patients, Japanese and Vietnamese authors have concluded

that the unintended eradication rate can reach the 10-14%.²²⁻²⁴ Considering that other diseases can also lead to gastric atrophy and/or intestinal metaplasia, this prevalence can be even lower than this indirectly estimated rate. The other most known reason for gastric atrophy is the autoimmune gastritis, which can be often excluded histologically regarding that the atrophy is restricted to the corpus in the autoimmune disease while the antrum is preferentially affected when it is caused by *H. pylori*. Nevertheless, several other pathological process might result in an *H. pylori*-negative gastritis, for example lymphocytic, collagenous, eosinophilic and other forms of gastritis and gastropathy, complicating the diagnosis of *Helicobacter*-negative gastritis.²⁵ Infection with other *Helicobacter* species can also cause similar symptoms and consequences such as gastritis or mucosa-associated lymphoid tissue lymphoma and these species are frequently found in *H. pylori*-negative gastric specimens by using special molecular methods.^{26,27} Otherwise, technical obstacles during detection of *H. pylori* in patients suffering from gastritis can also influence the results of the frequently used urea breath tests. Urea breath tests often give false negative results in patients under therapy with proton pump inhibitors.²⁸ These are the main reasons why we think that the unintended eradication rate can be even lower than the 10-14% prevalence of the *H. pylori*-negative mucosal atrophy and intestinal metaplasia cases. Based on this, a) we have only limited knowledge on the exact rate of the unintentional eradication but it seems to be very low; b) introducing this in the model would make it much more complicated by increasing the number of the unknown/uncertain parameters; c) this has presumably minimal effect on the model outputs.

Line 167: The model assumes that resistance can emerge in the hosts only infected by the susceptible strain in response to clarithromycin treatment (with probability pr). If this is because the host is actually also infected at low levels by the resistant strain, then shouldn't they be in the corresponding compartment? (index h) Alternatively, if this is because the susceptible bacteria develop resistance by spontaneous mutation during treatment, and this is then selected for by the presence of clarithromycin, then shouldn't pr be linked to the spontaneous mutation rate ω ?

This is an interesting question. As we have discussed in Supplementary Material 4, clarithromycin resistant bacteria continuously appear in the susceptible *H. pylori* populations by spontaneous mutation. However, the rate of successfully emerging Cla-res clones are low in the absence of clarithromycin-induced selection pressure. From this point of view, it is indeterminable whether there is any resistant *H. pylori* present in the population at a given time point or not, therefore these infections are generally categorized as susceptible. Of course, external stress conditions like the antibiotic selection pressure will increase the mutation rate.¹³ This will lead to an accelerated adaptation and when a resistance mutation occurs, this will spread quickly throughout the population by horizontal gene transfer resulting usually in a homoresistant population.^{29,30} This requires a mutational burst with a necessarily higher mutation rate than the spontaneous mutation rate ω .⁷ Namely, ω corresponds to the long-term mutation rate of the *H. pylori* and is related to the chronic phase of the infection.

Line 171: Is resistance always complete? They are not susceptible at all to the eradication treatment?

Thank you for this important question. In a recent systematic review and meta-analysis, Savoldi et al. observed a seven-fold higher risk of failing eradication with clarithromycin-containing eradication regimens in the clarithromycin resistant cases.³¹ In one of our previous studies, we found two successful eradication attempts out of the 13 clarithromycin homoresistant cases in which clarithromycin-containing regimen was used (85% failure rate).⁹ Therefore, the treatment failure rate is considerably high; it can even reach the 100% as it was observed in different studies with higher case numbers, for example by Lee et al.³² and Chang et al.³³ Based on this, we have assumed that the treatment failure rate of the clarithromycin-containing regimen is 100% in the clarithromycin homoresistant cases since a) it is considerably high, converges to the 100%; b) introducing another varying parameter would make the

model more complicated by increasing the uncertainty; c) this has presumably minimal effect on the model outputs.

Figure 2: This is a very informative figure but hard to follow, especially with the current legend. It needs to be stand-alone with more information on all the index notation. Perhaps a simplified version with just the natural history states (e.g. all of iw) could be included. This could then go in the supplementary.

We have modified the Figure 2 accordingly by inserting a simplified version of this.

The model is explored at steady state: this needs to be discussed in the main paper.

We politely disagree with the Reviewer since our model was designed for slow epidemiological change instead of steady state. We have clarified this point in the Supplementary material 2.

Why was it appropriate to assume that the system is at equilibrium when there has been such a change in *H. pylori* prevalence as shown by the data in the supplementary?

The model was designed for slow change instead of steady state. Accordingly, we have modified the Supplementary material 2.

Presumably, there is currently some level of drug susceptibility testing before eradication treatment is given. Can the estimates for the level of this in Hungary be given and used as a parameter in the model?

Unfortunately, the susceptibility testing-guided eradication is infrequent in Hungary. To our best knowledge, our laboratory is the only one, which currently offers susceptibility testing in Central Hungary. For example, the National Public Health Center (NPHC) plans to introduce it only in the near future – in cooperation with us. Moreover, some of the gastroenterologists do not wait for the result of the susceptibility test. Furthermore, if a patient had more than one susceptibility testing than we have included only the first result into this database. Therefore, the preliminary susceptibility testing has only a marginal importance among the factors influencing the population dynamics of *Clara-res H. pylori* infections. Thus we did not use such a parameter in the model.

Results

It seems that age and sex were significant influencers of the results – why were they not included in the mathematical model? Similarly, should the 1+/2+/3+ categories be included? What impact does their lack of inclusion have on the modelling results?

We thank the Reviewer for the critical note. The primary aim of the model was to find an explanation to the background of the clarithromycin resistance in macrolide naive patients. However, we did not find a significant difference in the prevalence of clarithromycin-resistance between females and males who had been never exposed with macrolide antibiotics (Table 2). Notwithstanding, gender is a highly influencing factor of the macrolide consumption and therefore the secondary resistance. However, no data supports that mechanisms of the development of the primary resistance (e.g. spontaneous mutation, infection by a resistant strain) might be gender- or age-dependent. Furthermore, it seems that the primary resistance rate is also not influenced by other gender-dependent physiologic, immunologic or other circumstances. Regarding the bacterial density (1+/ 2+/ 3+ categories), no relationship was found with the proportion of clarithromycin resistant cases in this study. Of course, it seems to be reasonable that the higher the bacterial load the higher the chance to transmit the infection. However, no such experimental or human study was published which supports this theory. Accordingly, we did not integrate this in the model since a) we have only limited knowledge on the effect of the bacterial load to the infectiousness; b) it would make the model much more complicated by increasing the number of the unknown/uncertain parameters; c) this has presumably minimal effect on the model outputs.

Line 187: "Females showed a significantly higher Cla-res rate than males"; please include the percentages in the main text here.

We have changed the text accordingly. (line: 113)

Line 191: Should "ratio" be replaced by "proportion"?

We have changed this expression accordingly. (line: 117)

Table 4: Clearly lays out all the results. More information could be given in the row titles.

We thank the Reviewer for this note. We have modified the row titles to make these clearer. As the space in the row titles is limited, we clarified the titles as much as possible and explained them in more detail in the table legend.

Line 227: What is the denominator for this percentage? New infections just with the resistant strain or with any *H. pylori*? How does this link to the 5.5% below?

We thank the Reviewer for this note. The denominator for this value is indeed the number of the new (not overinfection-causing) *H. pylori* infections. These are not identical with those cases from the 5.5% of the all macrolide-naives who acquired the clarithromycin resistance by infection with a Cla-resistant *H. pylori* strain. Based on the revised model, this value is 5.7% [$\lambda_r / (\lambda_r + \lambda_w)$] where the denominator is the $\lambda_r + \lambda_w$, corresponding to the sum of resistant and wild-type infections of a given period (e.g. annual rate). Thus, this is an output of the model and not directly belongs to the 5.5% macrolide-naive resistant cases.

Line 228: What supports this $k = 0.5$ value? Did the lab work find that in heteroresistant cases the bacterial load of R/S was the same? Could you quantify the bacterial burden at all and use this here?

We thank the Reviewer for this comment. To the best of our knowledge, the average proportion of resistant bacteria in heteroresistant cases has not been published yet.

We set $k=0.5$ as a baseline value, but we explored the full range of possible k values (0-1) during our sensitivity analysis.

We have also not checked systematically the resistant/susceptible ratio of the heteroresistant cases yet. However, regarding our everyday diagnostic experience, it seemed to follow the normal distribution (Gaussian curve). To prove this, we have measured the average resistant/susceptible ratio from 68 different biopsy samples with heteroresistant *H. pylori* population. By checking this with the Kolmogorov-Smirnov test of normality, we got 0.10867 as the D value of the K-S test with a p-value of 0.37139 meaning that distribution of our data does not differ significantly from the normal distribution (unpublished result).

Measured proportion of the resistant *H. pylori* bacteria in the 68 heteroresistant cases:

0.00965, 0.02577, 0.04056, 0.04207, 0.04622, 0.05374, 0.05459, 0.05907, 0.06782, 0.09511, 0.09789, 0.10848, 0.11318, 0.13462, 0.14971, 0.15100, 0.16069, 0.16519, 0.17483, 0.19403, 0.21576, 0.22827, 0.23591, 0.25502, 0.28898, 0.30594, 0.32420, 0.32823, 0.36573, 0.38882, 0.39900, 0.40953, 0.42571, 0.42652, 0.45389, 0.45560, 0.47930, 0.49625, 0.51454, 0.53625, 0.54131, 0.54777, 0.61549, 0.64038, 0.66697, 0.67627, 0.69087, 0.69660, 0.69697, 0.71608, 0.76040, 0.76054, 0.77519, 0.79967, 0.83308, 0.84863, 0.88203, 0.88287, 0.89411, 0.89842, 0.89906, 0.90005, 0.93136, 0.93665, 0.94983, 0.96681, 0.97621, 0.97917

Another addition to this issue is that we investigated the distribution of the resistant and susceptible bacteria between the antral and corpus regions of the stomach in one of our previous publications. In this, we have found no difference in the regional distribution (upper / lower part of the stomach) of the resistant and susceptible bacteria, suggesting that both forms of *H. pylori* have equal chance to get out from the gastric environment.⁹

Line 229: Can you add "relative to the susceptible strain"? This is a large fitness cost. Is this supported by other measures in the literature?

We have changed the sentence accordingly. (line: 176)

To our best knowledge it has not been investigated by others yet. However, according to the revised model, this parameter has changed to 0.72, which corresponds to a lower fitness cost.

Line 231: I would have thought that again this was not a parameter for the model to estimate but that there may be supporting evidence for this. How different is your estimate to this evidence? E.g. in references 21-24 cited in lines 266-8

We thank the Reviewer for this question. In the revised model, we used the 1.55×10^{-5} as ω value published by Kennemann et al, PNAS 2011⁶ and Linz et al, Nat Commun 2014.⁷

Line 230: it is assumed that hetero-resistant individuals have a 50-50 chance of transmitting either the resistant or susceptible bacteria ($\kappa = 0.5$). The authors perform a sensitivity analysis by looking at the effect of varying κ between 0 (hetero-resistant individuals only transmit the susceptible strain) and 1 (hetero-resistant individuals only transmit the resistant strain) on the transmission rate β and the relative fitness of the resistant strain δ . However, since the main results are the origins of primary resistance (i.e. the percentage of macrolide-naïve resistant infection originating from transmission of the resistant strain, and the percentage from spontaneous mutation), the sensitivity of these results to changes in κ , which are much more relevant than β and δ to a clinical audience, should be shown.

Let us offer some more detailed explanation: the 50-50 is the chance that the infector exposes the infectee with resistant or susceptible bacteria. The chance of transmission is also modulated by delta. Let us note that the main results do not depend on κ . The reason is that from the calculations we directly infer the lambdas.

Line 231: What is this 5.5% result? Can you make it clearer that this is the observed level of resistance? And that the next results are from the model?

We have changed the sentence accordingly. (line: 177)

Line 233: Here it would be good to emphasise that the 5.5% includes those with heteroresistant carriage.

We have modified this according to the Reviewer's recommendation. (line: 179-180)

There needs to be more analysis of the sensitivity of this model to the variation in the many unknown parameters and a description of the impact of this on the key results – the level of transmission vs. mutation.

Several of the parameters we have now determined from other results in the literature. For the remaining ones, we performed a sensitivity analysis.

Discussion

Line 255: can you add "in H. pylori infections"?

We have changed this sentence accordingly. (line: 212)

Line 276: How does this statement affect the assumption made in the model that transmission of resistant strains can result in heteroresistance from those infected with the wildtype strain? Did the model fit to the homo / hetero ratio from the data?

Indeed, our model predicted that a proportion of the primary resistant cases appeared as the consequence of overinfection by a resistant strain affecting a host with clarithromycin susceptible *H. pylori* infection. In this case, it will result in a heteroresistant (multiple) infection. Unfortunately, the homo/heteroresistance-related macrolide use is not available for each subgroup, as we have detailed above (this is why we calculated the “Developing heteroresistance by macrolide use” parameter indirectly by considering one of our previous publications). Therefore, we cannot use these data directly to address such a question but we introduced the known and closely predictable homo- / heteroresistance rate data as deeply into the model subgroups and parameters as it was possible. Based on these data, the model is able to predict such missing parameters like the ratio of transmitted to mutation-originated resistant infections.

Where is the discussion of the limitations of the model? The lack of uncertainty in the estimates and the uncertainty in other parameters and hence the impact on the results needs to be discussed.

Accordingly, we have inserted a new section in the Discussion about the limitations of the model. (line: 309-323)

Supplementary material 1

- Why did the authors not consider the results from Balint et al to be representative of the whole population of the Central Hungarian region?

We thank the Reviewer for his remark. Our decision to omit the *H. pylori* prevalence data of Balint et al.³⁴ based mostly on the fact that their study included healthy volunteers only. The cohort of Balint et al. colleagues, therefore, has a significantly lower mean age than our cohort (40 vs 55 years) which can substantially influence the prevalence of *H. pylori*. We could not consider this cohort to be representative of our cohort or of the whole population of the Central Hungary Region as the age is a significant influencer of the *H. pylori* prevalence. Other reasons were the regional differences between the cohort of Balint and her colleagues and the population of the Central Hungary Region. They investigated Csongrád and Békés Counties situated in the southeast part of Hungary with a higher number of agricultural workers and lower population density than the capital city of Hungary and Pest County. On the contrary, Buzás et al.¹⁰ had prevalence data directly from the Central Hungary Region and their grouped data published according to birth cohorts allows us to extrapolate their prevalence data to the age-grouped population data of the Hungarian population census from the investigated years. We considered this solution as the closest estimation of the real prevalence data of the Central Hungary Region.

- First table – why did *H. pylori* prevalence decline so much? Why is this data needed?

These data were used for the calculation of *H. pylori* prevalence by age groups. Since, the age distribution of the study of Buzás et al.¹⁰ and the Central Hungarian population was different, thus the rate of *H. pylori* infections was calculated from the age-grouped prevalence data of Buzás et al. for each female and male age group of the Central Hungarian population.

- Were these prevalence data really “fitted” or just applied to the model? How is the *H. pylori* prevalence calculated in the second table?

The birth cohort *H. pylori* prevalence data from the study of Buzás et al.¹⁰ were applied for calculating the infection rate in the Budapest and Pest County birth cohort populations of the Population census 2011, Hungary. A “fitting” was made because the birth cohorts from this population census and from the study of Buzás et al. both have 10 year / 5 year ranges (periods) but these are “shifted” relative to each other. It means that in the study of Buzás et al. these birth cohorts are from, for example, 1940-49, 1950-59, or 1960-64, 1965-69, etc. while the population census used five year periods like 1952-56, 1957-1961, 1962-66, 1967-71, etc. Consequently, there are certain 5-year periods of the population census which

completely overlap with one of the 10-year periods of study of Buzás et al. (e.g. 1932-1936 with 1930-39). In these 5-year periods we used the prevalence data from the related 10-year period of Buzás et al. However, regarding those 5-year population census periods, which overlap with two periods of the study from Buzás et al. (e.g. 1937-1941 with 1930-39 and 1940-49 or 1972-1976 with 1970-74 and 1975-79), we calculated a weighted average by obtaining the prevalence for 3 of the 5 years from the earlier period and for 2 of the 5 years from the later period (e.g. 1930-39: 62.8%, 1940-49: 63.0%, 1937-1941: $(62.8 \times 0.6) + (63.0 \times 0.4) = 62.88\%$). The only exception was the 1992-1996 period since here, we had no data for the later 2 years and therefore we have applied the last available 1990-94 prevalence data (12.2%) for the whole 1992-1996 period.

- How does the extrapolation work for the more recent birth cohorts? Did you fit a trend?

We thank the Reviewer for this note. Prevalence data in age groups younger than 55-59 years shows a linear trendline on the line plot, which allows us to assume a similar trend in the three unknown age groups. In lack of real data, the previously submitted version of this calculation included assumptions for the more recent birth cohorts. In the revised version, we modified this by fitting a linear trend, which resulted in slightly different *H. pylori* prevalence data in case of these age groups.

- The last sentence of this material suggests that all this work was just to get at an average population *H. pylori* prevalence? This needs to be made clearer at the start.

We thank the Reviewer for this note. These supplementary calculations were really made to get an *H. pylori* prevalence value since it was needed for the mathematical model. We have completed the introduction part of the Supplementary Material 1 accordingly.

How likely is this to be correct? Are there studies from other countries? What is the impact on the results of varying this result?

Our calculations are based on the prevalence data published by Buzás et his colleagues¹⁰ who investigated the *H. pylori* prevalence covering a similar time interval (1997-2012) to our study (2005-2013) also in the Central Hungarian Region. We fitted their prevalence data to the age-grouped / birth cohort data of the Hungarian population census 2011. According to this, we assume that these prevalence data approximate well the real *H. pylori* prevalence of the Central Hungarian region. Taking the declining prevalence of *H.*

pylori into consideration, our calculated prevalence is also supported by the results of previous publications investigating the prevalence in Hungary, which had found higher prevalence than we calculated.³⁵⁻³⁷

H. pylori prevalence is lower in Northern and Western Europe than in Hungary, while higher in Eastern and Southern Europe. As Hungary is a Central European country, our calculated prevalence data fit into this trend accordingly. The closest Central European country where data are available is the Czech Republic where 41.2% prevalence was published.⁸

- How was this overall prevalence split into initial conditions for uninfected / heteroresistant / resistant?

We did not make such a split. The overall prevalence equally influences the different compartments of the model.

- What are the coverages of the Hungarian population by these two regions (Budapest / Pest county)?

Total population by the population census 2011, Hungary: 9,937,628. Budapest and Pest County: 2,946,516 which is 29.65% of the total Hungarian population.

Supplementary material 2

- There is a lot of unneeded overlap here with the main text.

We thank the Reviewer for this critical note. We have reduced the overlaps.

- What are the initial conditions of the model in terms of natural history states? Or do you start with everyone uninfected and move to equilibrium with the conditions now?

We did not use any initial conditions, as the model was not integrated in time. We just used the model structure to infer the unknown current states and parameters for better understanding of the transmission of the clarithromycin resistance in the population. Hence, we did not integrate the actual state of the natural history of the *H. pylori* related disease(s) since a) we have only limited knowledge on the effect of the different states on the transmission of the infection; b) it would make the model much more complicated by increasing the number of the unknown/uncertain parameters; c) this has presumably minimal effect on the model outputs. In the revision, however, we are making some predictions with our model by solving it in time for two different scenarios. For them, the initial conditions were exactly those values of the state parameters that have been inferred.

- It is unacceptable that there are no references in this. Many new parameters are stated and their context should at least be discussed if there are no estimates from the literature. E.g. is there any evidence for fitness costs to resistance in *H. pylori*? Where do the ranges in table A2 come from?

We have added references into a new "Source" column of this table (see in the "Major "Comments" part), and expanded the discussion of parameters in the Supplementary Material 1 (general discussion of sources) and Supplementary Material 2 (discussion of the parameters in the context of the model). We do not assume any a-priori any fitness cost, for the relative transmission fitness we use >1 (but the results suggest that there is such a cost).

- Is resistance in *H. pylori* only mutational? This is what is suggested here but transformation can also occur – is this important for clarithromycin / macrolide resistance? What impact would this have on the observed / modelled dynamics?

To the best of our knowledge, clarithromycin / macrolide resistance is practically only mutational in the *H. pylori*. Moreover, there is an efflux pump mechanism but it is not able to effectively excrete the clarithromycin in the presence of the wild-type 23S rRNA. On the contrary, it can further increase the

efficacy of the 23S rRNA resistance mutations. Namely, in the rRNA mutate strains, clarithromycin is preferentially excreted by the efflux pumps because of its low affinity with the mutated ribosomal site. Consequently, the efflux pump mechanism does not act as an independent resistance system but synergistically with the mutational resistance.³⁸⁻⁴⁰

The horizontal gene transfer is an existing mechanism in *H. pylori* which can be involved in the fast spread of the antibiotic resistance in a given bacterial population. Although there are data supporting that this phenomenon can occur between different *Helicobacter* species as well but this was found to be unidirectional in the case of such a closely related bacterium like the *Campylobacter jejuni* (*H. pylori* → *C. jejuni*). Considering this, we can conclude that the horizontal gene transfer can contribute to the spreading of the clarithromycin resistance within one *H. pylori* strain or between different strains when a person is infected by multiple strains rather than transmit the resistance between different individuals.⁴¹⁻⁴³

Accordingly, we did not integrate this in the model since a) we have only limited knowledge on the effect of the horizontal gene transfer in vivo; b) it would make the model much more complicated by increasing the number of the unknown/uncertain parameters; c) this has presumably minimal effect on the model outputs.

Accordingly, we have inserted some sentences about the efflux pump and the horizontal gene transfer mechanisms into the Introduction. (line: 55-66)

- As in the main text – there should be a link between the spontaneous mutation rate and the fraction of exposed wild type individuals that become heteroresistant upon macrolide exposure

According to the Supplementary Material 4, clarithromycin resistant bacteria continuously appear in the susceptible *H. pylori* populations by spontaneous mutation. However, the rate of successfully emerging Cla-res clones are low in the absence of clarithromycin-induced selection pressure. The external stress conditions like the selection pressure by antibiotics will increase the mutation rate.¹³ This will lead an accelerated adaptation and when a resistance mutation occurs, this is able to spread in the population by horizontal gene transfer. The external stress condition also means an increased mutation rate which has also been described during the acute phase of *Helicobacter pylori* infection by Linz et al.⁷ Therefore this adaptation-related mutation rate theoretically should be higher than the mutation rate observed from the model which is more likely a long-term mutation rate relating to the chronic phase of the *H. pylori* infection. This is the reason why we do not link the mutation rate (ω) of the model to the appearing macrolide-caused resistance but use the “m” parameter instead of this. In order to reduce the number of the uncertain parameters, we applied a fixed value for this in the revised model, based on the publication of Kennemann et al. (PNAS 2011)⁶ and Linz et al (Nat. Comm).⁷

- There needs to be a description of why a deterministic system was chosen. What are the limitations of this?

We think this is not relevant for us, since we were not investigating particular model trajectories, and we considered a large population.

Limitations section

- What other types of infection could be? Shouldn't the “categorised all strains” be “infection states” not “strains”?

We thank the Reviewer for catching this oversight. We rephrased this section in the revised Supplementary material 2.

- Do you have any data on repeat exposure?

We received the information on the prior macrolide use of our patients from the National Health Insurance Fund Management of Hungary as aggregated data only, according to their privacy policy. Since the study was primarily designed to find the macrolide-naïve (primary resistant) infections, thus, from practical considerations, the “any prior macrolide use” was investigated during the data collection, independently of the number of the treatment events, therefore we have no information on the repeated exposures.

- “the emerged resistant strain cannot completely wipe out the already established wild-type population”; could the authors further discuss this assumption in light of experimental evidence for the importance of horizontal gene transfer by transformation in *H. pylori*, and the fact that *H. pylori* infection generally persists for life except if eradication therapy is undertaken?

Basically, we do not have information on it. Our assumption is based on the next arguments:

1) *H. pylori* bacteria with macrolide exposure-induced Cla-res are supposed to develop compensatory mutations to overcome fitness cost of the Cla-res mutation.¹⁹

2) Spontaneously emerging Cla-res *H. pylori* bacteria are supposed to have fitness cost but there is no data on the effectiveness or even the existence of the compensatory mutations in the absence of the selection pressure of macrolides.

3) Horizontal gene transfer helps these mutations (Cla-res / compensatory) to spread quickly throughout the bacterial population under the selection pressure of the macrolides (clarithromycin) resulting predominantly in a homoresistant population (as our results showed a 70%/30% homo-/heteroresistant rate after an unsuccessful eradication attempt).⁹

4) We have no information on the role of the horizontal gene transfer in the spontaneously emerging Cla-res infections.

5) In the in vivo experiments of Björkholm et al¹⁹, they infected mice by 1:1 mixture of susceptible and Cla-res *H. pylori* isolates sampled clinically from the same patient before and after an unsuccessful clarithromycin-containing eradication attempt the authors found that in one of the mice the susceptible isolate has overgrown the resistant, in another one the resistant has overgrown the susceptible and equilibrium between the two isolates has been identified in a further animal. (Björkholm et al, PNAS, 2001; Figure 3) The authors concluded that the biological / fitness costs of the resistance mutations can be reduced in clinical isolates by compensatory mutations. However, in this experiment, the Cla-res was induced by in vivo clarithromycin treatment of the human patients.¹⁹

Consequently, as we suppose that the compensatory mutations are less and/or less effective (if any of them exists) in a spontaneously emerging Cla-res population, thus the heteroresistance seems to be more possible than the resistant subpopulation is able to overgrow the susceptible bacteria resulting in homoresistant bacterial population.

- The steady state assumption is a big one – how can you be sure that it is correct?

We politely note that we do not assume anything about steady state, only that the dynamics is changing very slowly. For this we used the phrase “near an equilibrium”, which was a bit misleading hence we have rephrased it.

- This section (or somewhere else in the paper) needs to explore how these assumptions affect the model output not just be statements. For example, what might be the impact of heteroresistant dual transmission? Would it make your estimates under or over-estimates of transmission?

- As mentioned in the main text – why does the model not include age and gender stratifications? They had a big impact on resistance levels.

These are good points, however, such investigations would constitute an additional paper each. Furthermore, many new compartments and parameters should be introduced, of which we have no information whatsoever. Stratifying by gender and age would multiply the number of compartments, and we have no idea about relative transmission parameters between these many subgroups (for example, what is the transmission rate from young females to elderly males etc.). Allowing heteroresistant dual transmission means three new transition routes, and we would need the probability of heteroresistant transmission. We have added some remarks to the limitations, and we hope that our work will inspire future research when these factors will be investigated in detail.

Calculations

- Remove A4 from the heading?

Thank you for catching this oversight. We have changed this expression accordingly.

- Can you provide a reference for the mortality statistics? I assume that you just used the most recent mortality rate and kept this the same as the birth rate to keep a steady population size but you need to state exactly what you did.

We do not assume anything about steady state, since hypothesized that the population dynamics is changing very slowly. For the demographic turnover (i.e. death and birth rates), we considered the average mortality rate of Hungary in the period 2005-2013, taken from the database of the Hungarian Central Statistical Office (KSH).

https://www.ksh.hu/docs/eng/xstadat/xstadat_annual/i_wdsd006a.html

https://www.ksh.hu/docs/eng/xstadat/xstadat_annual/i_wdsd006b.html

- Table A2 – where are the references? It needs to be clear where these numbers came from as this works need to be supported and reproducible.

We thank you for this note. We have described all parameters in detail in the Supplementary 1 and 2, as well as we added a new "Source" column in the identical table of the main text (Table 3).

- I would have thought that the antibiotic usage rates could have been taken from the data used in the main text?

We have detailed the source of each parameter in the revised Supplementary Material 1 including this information as well. (and also here above, among the first answers to the Major comments)

- H pylori prevalence in supplementary 1 was said to be 40.3% - why are you using a range now? And how?

In the previous version of the model, we used the value of 40.3% but let it vary in a range around this. In the revised model, we do not let the variation of this parameter, moreover recalculate it according to the comments of the Reviewer, resulting in 40.9% *H. pylori* prevalence. We provide further details in the answers to the reviewer's comments to the Supplementary Material 1 and in the revised version of the Supplementary Material 1.

- The relationships in the Cj statements need to be supported by the calculations written out clearly e.g. number of people in this state / total number of people

We thank the Reviewer for pointing out this flaw. We have written out clearly the number of people in the certain infection state / the total number of people in the revised version of Supplementary material 2.

- For C1 I think the reference should be to Table 3.

In the revised version we have simplified the labeling by omitting "C" labels and linked the corresponding tables as references to the equations.

- How were the feasible ranges in Table A2 decided upon?

We have substantially changed our mathematical model including a fundamental change in the determination of the values of model parameters. In order to reduce the uncertainty of the model outputs and to ensure the correct sensitivity analyses we omit to let the values of parameters vary in ranges and have determined a fixed value for the maximum number of the input parameters as precisely as it was possible, based on the cohort data and/or data of published studies.

- What does "close" to an equilibrium mean? And "with some accuracy"?

We have rephrased this misleading sentence.

- How do you know that you have a unique equilibrium state?

In the revised version we are not concerned with any equilibrium: we set the prevalence trend based on empirical observations in the literature, and we are looking for parameters for which the epidemiological dynamics is slow while satisfying the decreasing prevalence trend.

- What does the value of alpha tell us about the conditions in Cj? that they do / do not hold with your parameters? Seems that this leads to a large bias towards parameters that satisfy the equilibrium condition but not the data on the levels of wild type / resistant etc?

In the revised version we incorporated much more a-priori information, and do not use Cj neither alpha any more.

- Can you link to supplementary material 1 for the 40.3% H value?

We modified this accordingly.

- Can you perform a sensitivity analysis using this method? How can you convey the uncertainty shown in the parameter ranges in Table A2? For example, could you fix one parameter and vary the others?

We performed a sensitivity analysis with respect to three parameters, varying them parallel.

- Table A3 – what is the value for kappa? 0.5?

From the minimizer, we infer some values for λ_w and λ_r . These values are consistent with a variety of beta, delta and kappa combinations. Thus we use the whole possible range [0,1] for kappa, and each kappa determines a beta and a delta value. This way we obtain a possible range for beta and delta.

Conclusions

- Does the proportion of new infections depend on the population size? E.g. this is true if all are uninfected, but what if the majority of individuals are infected with the resistant strain?

According to the results of the revised model, the fitness cost of the resistance is smaller than in the previous version but still exists ($\delta = 0.72$). Therefore, the number of the new infections is dependent on the resistance rate, as the transmission of the Cla-res infections is reduced in comparison with the susceptible strains. However, δ may vary in a relatively large range as our sensitivity analysis showed.

- Linked to the above comment on sensitivity analysis, how did you generate the ranges for delta and beta? Did you fix kappa at different values and minimize W again? Could you do this for other important parameters?

First we find the lambdas from the minimizer, then vary kappa from 0 to 1 to produce these ranges. In the revision, delta and beta are varying also due to the sensitivity analysis.

- So this 64% is at equilibrium and is dependent on the population values e.g. 5% of the population have experienced eradication treatment but are not infected. Is this realistic? These values all need to be stated (or at least those that drive the 64%, un, inw) and explored for clinical / epidemiological validity.

This is an interesting question. However, unfortunately we do not have published or any other reliable data about this. However, we asked Hungarian gastroenterology specialists and based on these expert opinions, this proportion seems to be realistic.

- Where does the one-third are from over colonisation come from?

This is simply from the ratio of inw and un, we have added this comment to the text.

Supplementary material 3

This needs an introduction paragraph.

We completed the Supplementary material 3 with an introduction paragraph.

REVIEWER #3 (REMARKS TO THE AUTHOR):

The manuscript by Kocsmár et al presents a strategy to determine the existence of clarithromycin resistance that is independent from macrolide consumption. The manuscript is well-written and understandable, but I have several concerns related to specific aspects of the methodology.

The mathematical modelling of the manuscript is largely based on data obtained using FISH. However, I found it really difficult to find details on the BACTFish H. pylori Combi Kit, both on the suppliers webpage or on the mentioned references (9 and 11). Reference 9 refers only to the H. pylori detection, whereas ref. 11 does refer to the clarythromycin method but for methodological details also links to the original publication in 9. Without a full disclosure of the methodology or at least strong data on the specificity and sensitivity of the method, it is not possible to assess the accuracy of the data provided. In the discussion the authors refer references 25 and 26 to support the accuracy of FISH-based data, but were these two studies conducted in the exact same conditions? The original probes appear to be from Russmann, but details on the hybridization and washing solutions are lacking.

We thank the Reviewer for the careful review and comments on our study.

1. Also, are all point mutations associated with CLA resistance picked by the probes?

The used FISH test detects three distinct point mutations in the peptidyl transferase region of 23S rRNA.⁴⁴ These mutations are situated in two gene positions, actually numbered as 2142 or 2143 (formerly named 2058 and 2059, respectively, which are the cognates in *E. coli*, then 2143 and 2144 according to the *H. pylori* 23S rRNA sequence GenBank U27270 and subsequently these have changed according to the current nomenclature).^{45,46} These are the most prevalent and well-documented clarithromycin resistance mutations in *H. pylori* corresponding to an adenine-to-guanine transition at either position 2142 (A2142G) or 2143 (A2143G), or, less frequently, an adenine-to-cytosine transversion at position 2142 (A2142C). Mutation at position 2143 is usually associated with different levels of resistance (MICs ranging from 2 to

256 mg l⁻¹), while mutation at position 2142 commonly leads to a more restricted resistance (MIC of 64 mg l⁻¹).⁴⁶

Several studies proved that there is an almost perfect (90-100%) correlation between these three mutations and the phenotypic resistance⁴⁷, which was also proved by the FISH technique used by us.⁴⁸⁻⁵⁰ Although further point mutations were also identified in this ribosomal RNA region but these: 1) are able to confer clarithromycin resistance but found to be very rare (e.g. A2115G, G2141A, A2144T and T2289C); 2) have been associated with low resistance levels (e.g. C2694A and T2717C); 3) were not found to be manifested in phenotypic resistance (e.g. T2182C).^{46,51} Of course, our test is unable to diagnose rare mutations leading to clarithromycin resistance or resistance caused by uncommon genetic mechanisms which is a limitation of this study (thus we now display it in the Discussion, line: 299-306). However, based on the above, we are convinced that our FISH test detects the clarithromycin resistance of *H. pylori* precisely, with enough sensitivity for the main goals of this study.

2. Was the hybridisation carried out without any fixation or permeabilization solution?

Originally, this *H. pylori* clarithromycin susceptibility FISH test was developed and described by a group from the Max von Pettenkofer Institute for Hygiene and Medical Microbiology, Ludwig Maximilians University Munich, Germany.⁴⁴ Subsequently, this technique was commercialized and distributed under different names including creaFAST, seaFAST and BACTFish *H. pylori* Combi Kit. Initially, the protocol did not include any permeabilization step. This was appropriate for the microbiological smears and the small biopsy specimens for which relatively short fixation time might be sufficient. However, it turned out subsequently that the routine fixation practice of many pathology laboratories varies greatly (6-24 hour fixation in 10% formalin solution is recommended for the small biopsies but the fixation time is acceptable up to 48 hours). As a consequence, specimens with longer fixation times require a permeabilizing pretreatment step, similarly to other FISH protocols for FFPE samples. The instruction manual of the BACTFish *Helicobacter pylori* Combi kit recommended 400W for 10 min microwaving in Vector Antigen Unmasking Solution (Cat. No. H-3300, Vector Laboratories) as pretreatment (see in the BACTFish instruction manual attached as Related Manuscript File). Our experience showed that longer microwaving time might be needed for FFPE tissue samples received from certain sending departments. Thus, as a pretreatment step in our laboratory, the slides were microwaved at 400W for 20 minutes in Vector Antigen Unmasking Solution H-3300 (Vector Laboratories, Burlingame, CA, USA). The detailed FISH protocol is now included in the Methods section.

3. Lines 83-88: How many tissue sections from each patient were used for the study? From which section(s) of the stomach? If the authors only analysed one section of the stomach, how likely is it for other strains of *H. pylori* to be present and how would this impact the conclusions of this study?

As we have also indicated in our answer to the "Other comments" of Reviewer 1, multiple biopsy samples were obtained in each case. However, the samples from the different gastric regions of a given patient were usually collected and sent for histopathological work-up in the same sample collection tube. Accordingly, the different samples of the same patient were embedded together into one paraffin block. Samples from the antral and oxyntic gastric compartments were distinguishable during the routine histological investigation based on the expertise of the pathologists. However, it was not a requirement to record the results of the evaluation of *H. pylori* IHC / FISH separately for each individual sample of the same tissue block. Therefore, the final IHC / FISH result was indeed a summary of all the IHC / FISH patterns observed in the separate tissue samples of the same section. In the minority of the cases (346 patients) the samples of the different sites of the stomach were sent in separate collection tubes and consequently, handled separately. Therefore, we have detailed information about the IHC / FISH results of the individual tissue samples originating from different sites of the same stomach in these cases of our cohort. From a given patient, all of the individual samples in which *H. pylori* was diagnosed were further

processed for *H. pylori* clarithromycin susceptibility FISH test. Regarding the questioned issue, we have analyzed this in one of our recent publications.⁹ Based on this, we can say that about 20% of the resistant cases can remain hidden with the one-site-testing approach. Of course, it is possible that no *H. pylori* can be found in a section plain due to the low bacterial density, as we have discussed earlier.¹⁷ Therefore, even though that both the antrum and the corpus were sampled, *H. pylori* clarithromycin susceptibility was analyzed by FISH test in some cases only from one gastric compartment. By comparing the heteroresistance rate of the present manuscript (431 homo: 53%, 385 hetero: 47%) with that of the above mentioned paper (*Helicobacter pylori* heteroresistance to clarithromycin in adults - New data by in situ detection and improved concept: 35 homo; 48%, 38 hetero; 52%) in which at least one sample was analyzed from both the antral and corpus compartments in each cases, we found no significant difference between the prevalence of the clarithromycin heteroresistance. This indicates that the lack of one of the gastric regions available for susceptibility FISH test in some cases was not such a factor which seriously biased the results of our study.

4. Lines 108-110: Please provide details of the wavelengths of the mentioned filters

The wavelengths of the fluorescence filter cubes were inserted into the following sentence in the Material and Methods section:

"Then the slides were examined using epifluorescence microscope Leica DM-RXA (Leica Microsystems, Wetzlar, Germany) equipped with DAPI (excitation 355-425 nm, emission 470- nm, Leica Microsystems, Wetzlar, Germany), Spectrum Green (excitation 460-500 nm, emission 512-542 nm, Vysis, Downers Grove, IL, USA), and Spectrum Orange (excitation 540-578 nm, emission 564-612 nm, Vysis, Downers Grove, IL, USA) filter sets for blue, green, and orange-red fluorescence, respectively." (line: 391-396)

5. Figure 1 – Please include scale bars. Also, out of curiosity, it would be expectable that microbial cells would also be labelled blue by DAPI. Was this removed with image processing?

A scale bar was included in the first image of Figure 1 (in the first image only as the further images have the same magnification in this figure).

The bacterial cells are visible by DAPI staining but the intensity of this fluorescence is very low in comparison with the nuclear staining of the epithelial cells. So, bacterial fluorescence was not removed artificially but not visible under normal conditions. In order to demonstrate this phenomenon we attached a picture in which the DAPI channel was exposed twice with different exposure times. For the first, the exposure time was shorter in order to avoid the overexposure of the nuclei (this is the normal setting which was used for the images of Figure 1 as well). In this case, the bacterial cells are poorly visible (especially in dark blue) since their fluorescence intensity is lower by magnitudes. Moreover, when we merge this DAPI channel with the Spectrum Green and Spectrum Orange channels this faint DAPI signal of the bacteria becomes invisible since the specific signal from the green (and the red - in the case of clarithromycin resistance) channel(s) is much more intense. The second time we used an extended exposure time for the better visualization of the *H. pylori* cells in the DAPI channel. However, in this case, the nuclei are strongly overexposed while the bacteria are visible, especially if we change the blue color of DAPI to white.

1. Heteroresistant *H. pylori* infection. The image was created by merging of channels of DAPI (blue), green and red. (630x)

2. DAPI (blue) channel

3. DAPI channel in white

4. Green channel

4. Red channel

6. Overexposed DAPI channel (blue)

7. Overexposed DAPI channel in white

REVIEWER #4 (REMARKS TO THE AUTHOR):

In this manuscript Kocsmar et al. report on a study of primary and secondary *H. pylori* resistance against clarithromycin in Hungary.

This is an interesting study with a large number of patients, using fluorescence in situ hybridization (FISH) to detect *H. pylori* and its resistance, a technique which seems to provide a better detection of heteroresistance compared to other methods, and where information on a previous macrolide treatment was available through a national data base. They found a primary resistance rate of 13.3% in so-called naive patients according to the usual definition, and 5.5% in patients who were never prescribed macrolides. They also applied a mathematical model to follow the dynamics of clarithromycin resistant (ClariR) strains and claim that 2/3 come from the transmission of these ClariR strains and 36% from spontaneous mutations.

While it is difficult for a non-specialist to understand this model, a limit lies in the fact that a few clones of ClariR *H. pylori* may be present in a stomach following a spontaneous mutation (occurring by chance) and remain undetected until they reach a certain proportion of the whole population.

We are thankful to the Reviewer for the accurate analysis of our manuscript.

1. It would be interesting to know the limit of detection of ClariR with the FISH method and how it affects the modeling.

As we also discussed a similar question for the Reviewer 3, we answer the same:

The used FISH test detects three distinct point mutations in the peptidyl transferase region of 23S rRNA.⁴⁴ These mutations are situated in two gene positions, actually numbered as 2142 or 2143 (formerly named

2058 and 2059, respectively, which are the cognates in *E. coli*, then 2143 and 2144 according to the *H. pylori* 23S rRNA sequence GenBank U27270 and subsequently these have changed according to the current nomenclature).^{45,46} These are the most prevalent and well-documented clarithromycin resistance mutations in *H. pylori* corresponding to an adenine-to-guanine transition at either position 2142 (A2142G) or 2143 (A2143G), or, less frequently, an adenine-to-cytosine transversion at position 2142 (A2142C). Mutation at position 2143 is usually associated with different levels of resistance (MICs ranging from 2 to 256 mg l⁻¹), while mutation at position 2142 commonly leads to a more restricted resistance (MIC of 64 mg l⁻¹).⁴⁶

Several studies proved that there is an almost perfect (90-100%) correlation between these three mutations and the phenotypic resistance⁴⁷, which was also proved by the FISH technique used by us.⁴⁸⁻⁵⁰ Although further point mutations were also identified in this ribosomal RNA region but these: 1) are able to confer clarithromycin resistance but found to be very rare (e.g. A2115G, G2141A, A2144T and T2289C); 2) have been associated with low resistance levels (e.g. C2694A and T2717C); 3) were not found to be manifested in phenotypic resistance (e.g. T2182C).^{46,51} Of course, our test is unable to diagnose rare mutations leading to clarithromycin resistance or resistance caused by uncommon genetic mechanisms which is a limitation of this study (thus we now display it in the Discussion, line: 299-306). However, based on the above, we are convinced that our FISH test detects the clarithromycin resistance of *H. pylori* precisely, with We have also proved previously that the *H. pylori* diagnostic accuracy of the used FISH test is similar to the gold-standard immunohistochemical detection of the bacteria.¹⁷ Regarding the connection of the diagnostic accuracy of both the *H. pylori* bacteria and the clarithromycin resistance with the model, it is obvious that a low sensitivity method would bias the model in some degree by influencing the input parameters. However, we are persuaded that our method achieve such a high level of accuracy that this is negligible.

2. The finding that non-eradication clarithromycin therapy leads to more ClariR than other macrolides is also in contradiction with a European study on this topic where the best fit between *H. pylori* ClariR and macrolide consumption was for long acting macrolides and not for clarithromycin.

There are significant differences in the methods of the two studies. In the mentioned study from Mégraud et al., population level macrolide use data was used while they tested about 100 patients/10 million inhabitants in a Pan-European multicenter study.⁴ From Hungary, one center was included which provided 100 consecutive non-duplicate clinical isolates of *H. pylori*. This site was the 2nd Department of Medicine from Semmelweis University (personal communication with Prof. Mégraud). Our data have shown that the clarithromycin resistance rate was usually higher among the patients of the clinics of Semmelweis University than in other hospitals and outpatient clinics. But we can anyway suppose that 100 patients from one site might not be representative for 10 million people. We can easily apprehend that comparing data from this kind of sample collection with the population level macrolide use data is much less representative than our approach where thousands of patients were included from the same geographical region and individual macrolide use data were available. The authors of the mentioned study themselves suppose that the insufficient sample size might contribute to the strange findings that the long acting macrolides show better correlation with the clarithromycin resistance than the clarithromycin itself: "For macrolides, the lack of a statistically significant association between outpatient use data in 2005 and the proportion of clarithromycin-resistant strains could have been due to the insufficient sample size in this survey. However, this association became significant when the use of long-acting macrolides only (azithromycin) was considered in the analysis." [Megraud F, et al. *Helicobacter pylori* resistance to antibiotics in Europe and its relationship to antibiotic consumption Gut 2013;62:34-42.]. Another factor that the proportion of the clarithromycin among the used macrolides is much higher in Hungary than in other countries like Italy, Portugal, France, Poland, Slovenia, Croatia, UK, Germany, Spain, Norway and Finland, moreover, the long-acting macrolides are infrequently used in Hungary in comparison to others e.g. Italy, Portugal, Slovenia, Croatia, Spain, Finland (see in Figure 3 of the mentioned study, Megraud F, et

a. *Helicobacter pylori* resistance to antibiotics in Europe and its relationship to antibiotic consumption Gut 2013;62:34-42.).

3. It is not clear why a different therapeutic strategy should be proposed for heteroresistant vs homoresistant *H. pylori*.

We discussed it in detail in one of our recently published studies (Kocsmár É et al. *Helicobacter pylori* heteroresistance to clarithromycin in adults - New data by in situ detection and improved concept. *Helicobacter*. 2020 Feb;25(1):e12670. doi: 10.1111/hel.). Clarithromycin has been used for *H. pylori* eradication treatment traditionally because of its low minimum inhibitory concentration, good mucosal diffusion and relatively small effect on gastric acidity.⁴⁶ As the majority of the infections are still susceptible, these patients can benefit from clarithromycin-containing regimens by using appropriate susceptibility testing. Moreover, homoresistant cases certainly need non-clarithromycin-based protocols. However, our results, in agreement with published data from De Francesco et al.⁵² indicate that heteroresistant cases represent a distinct subgroup with an intermediate cure rate of *H. pylori* infection when treated by a clarithromycin-containing eradication regimen. Heteroresistant *H. pylori* infections shall be reported as resistant to clarithromycin, and non-clarithromycin-containing regimens are still the first-choice treatment preferences for these patients. Nevertheless, personalized therapeutic options may be considered in certain heteroresistant cases, including the additional use of clarithromycin as well, since non-Cla-containing protocols can also fail the eradication in heteroresistant cases. Hence, further investigations are needed to clarify the exact role of clarithromycin and other antibiotics in the treatment of Cla-heteroresistant infections including the potential efficacy of clarithromycin-containing quadruple therapies.

4. It would be nice to present the resistance rate in females also for the so-called primary resistance and for those who were never prescribed macrolides.

We thank the Reviewer for this recommendation. We have changed the figure accordingly. (Figure 2)

Figure 2 Prevalence of *H. pylori* clarithromycin resistance in females and males by age group (relative to the total cohort). Deep blue: macrolide-naive resistant, Green: Clarithromycin-containing eradication-naive (but other macrolide treatment exposed) resistant, Light blue: Resistant with history of *H. pylori* eradication with clarithromycin-containing treatment regimen.

1. Malfertheiner, P. *et al.* Management of *Helicobacter pylori* infection-the Maastricht V/Florence Consensus Report. *Gut* **66**, 6–30 (2017).
2. O'Connor, A., Liou, J.-M., Gisbert, J. P. & O'Morain, C. Review: Treatment of *Helicobacter pylori* Infection 2019. *Helicobacter* **24 Suppl 1**, e12640 (2019).
3. Liou, J.-M. *et al.* The Primary Resistance of *Helicobacter pylori* in Taiwan after the National Policy to Restrict Antibiotic Consumption and Its Relation to Virulence Factors-A Nationwide Study. *PLoS ONE* **10**, e0124199 (2015).
4. Megraud, F. *et al.* *Helicobacter pylori* resistance to antibiotics in Europe and its relationship to antibiotic consumption. *Gut* **62**, 34–42 (2013).
5. Rupnow, M. F. T., Shachter, R. D., Owens, D. K. & Parsonnet, J. A Dynamic Transmission Model for Predicting Trends in *Helicobacter pylori* and Associated Diseases in the United States - Volume 6, Number 3—June 2000 - Emerging Infectious Diseases journal - CDC. doi:10.3201/eid0603.000302.
6. Kennemann, L. *et al.* *Helicobacter pylori* genome evolution during human infection. *Proceedings of the National Academy of Sciences* **108**, 5033–5038 (2011).
7. Linz, B. *et al.* A mutation burst during the acute phase of *Helicobacter pylori* infection in humans and rhesus macaques. *Nat Commun* **5**, 4165 (2014).
8. Hooi, J. K. Y. *et al.* Global Prevalence of *Helicobacter pylori* Infection: Systematic Review and Meta-Analysis. *Gastroenterology* **153**, 420–429 (2017).
9. Kocsmár, É. *et al.* *Helicobacter pylori* heteroresistance to clarithromycin in adults-New data by in situ detection and improved concept. *Helicobacter* **25**, e12670 (2020).
10. Buzás, G. M., Lotz, G., Schneider, F. & Józán, J. [Changing prevalence of *Helicobacter pylori* infection in the 9th district of Budapest. A retrospective endoscopic study, 1997-2012]. *Orv Hetil* **154**, 900–907 (2013).

11. Ben Mansour, K. *et al.* Multiple and mixed *Helicobacter pylori* infections: Comparison of two epidemiological situations in Tunisia and France. *Infect. Genet. Evol.* **37**, 43–48 (2016).
12. Seo, J. W., Park, J. Y., Shin, T.-S. & Kim, J. G. The analysis of virulence factors and antibiotic resistance between *Helicobacter pylori* strains isolated from gastric antrum and body. *BMC Gastroenterology* **19**, 140 (2019).
13. Martinez, J. L. & Baquero, F. Mutation Frequencies and Antibiotic Resistance. *Antimicrob Agents Chemother* **44**, 1771–1777 (2000).
14. Tacconelli, E. *et al.* Discovery, research, and development of new antibiotics: the WHO priority list of antibiotic-resistant bacteria and tuberculosis. *Lancet Infect Dis* **18**, 318–327 (2018).
15. Niewiadomska, A. M. *et al.* Population-level mathematical modeling of antimicrobial resistance: a systematic review. *BMC Medicine* **17**, 81 (2019).
16. White, J. R., Winter, J. A. & Robinson, K. Differential inflammatory response to *Helicobacter pylori* infection: etiology and clinical outcomes. *J Inflamm Res* **8**, 137–147 (2015).
17. Kocsmár, É. *et al.* Sensitivity of *Helicobacter pylori* detection by Giemsa staining is poor in comparison with immunohistochemistry and fluorescent in situ hybridization and strongly depends on inflammatory activity. *Helicobacter* **22**, e12387 (2017).
18. Kanai, K., Shibayama, K., Suzuki, S., Wachino, J. & Arakawa, Y. Growth competition of macrolide-resistant and -susceptible *Helicobacter pylori* strains. *Microbiol. Immunol.* **48**, 977–980 (2004).
19. Björkholm, B. *et al.* Mutation frequency and biological cost of antibiotic resistance in *Helicobacter pylori*. *Proc Natl Acad Sci U S A* **98**, 14607–14612 (2001).
20. LeBel, M. Pharmacokinetic properties of clarithromycin: A comparison with erythromycin and azithromycin. *Can J Infect Dis* **4**, 148–152 (1993).
21. Ierardi, E. *et al.* Optimizing proton pump inhibitors in *Helicobacter pylori* treatment: Old and new tricks to improve effectiveness. *World J Gastroenterol* **25**, 5097–5104 (2019).

22. Ono, S. *et al.* Frequency of *Helicobacter pylori* -negative gastric cancer and gastric mucosal atrophy in a Japanese endoscopic submucosal dissection series including histological, endoscopic and serological atrophy. *Digestion* **86**, 59–65 (2012).
23. Boda, T. *et al.* Advanced method for evaluation of gastric cancer risk by serum markers: determination of true low-risk subjects for gastric neoplasm. *Helicobacter* **19**, 1–8 (2014).
24. Hiyama, T. *et al.* Rate of unintended *Helicobacter pylori* eradication in the Vietnamese. *Helicobacter* **20**, 156–157 (2015).
25. El-Zimaity, H., Choi, W.-T., Lauwers, G. Y. & Riddell, R. The differential diagnosis of *Helicobacter pylori* negative gastritis. *Virchows Arch* **473**, 533–550 (2018).
26. Nakamura, M. *et al.* PCR analysis and specific immunohistochemistry revealing a high prevalence of non-*Helicobacter pylori* *Helicobacters* in *Helicobacter pylori*-negative gastric disease patients in Japan: High susceptibility to an Hp eradication regimen. *Helicobacter* **n/a**, e12700.
27. Kiss, S., Zsikla, V., Frank, A., Willi, N. & Cathomas, G. *Helicobacter*-negative gastritis: polymerase chain reaction for *Helicobacter* DNA is a valuable tool to elucidate the diagnosis. *Alimentary Pharmacology & Therapeutics* **43**, 924–932 (2016).
28. Graham, D. Y. *et al.* Studies regarding the mechanism of false negative urea breath tests with proton pump inhibitors. *Am. J. Gastroenterol.* **98**, 1005–1009 (2003).
29. Fernandez, A. F. *et al.* A DNA methylation fingerprint of 1628 human samples. *Genome Res* **22**, 407–419 (2012).
30. Israel, D. A., Lou, A. S. & Blaser, M. J. Characteristics of *Helicobacter pylori* natural transformation. *FEMS Microbiol. Lett.* **186**, 275–280 (2000).
31. Savoldi, A., Carrara, E., Graham, D. Y., Conti, M. & Tacconelli, E. Prevalence of Antibiotic Resistance in *Helicobacter pylori*: A Systematic Review and Meta-analysis in World Health Organization Regions. *Gastroenterology* **155**, 1372-1382.e17 (2018).

32. Lee, J. H. *et al.* Impact of Clarithromycin Resistance on Eradication of *Helicobacter pylori* in Infected Adults. *Antimicrob Agents Chemother* **49**, 1600–1603 (2005).
33. Chang, Y. W. *et al.* Clarithromycin resistance and female gender affect *Helicobacter pylori* eradication failure in chronic gastritis. *Korean J. Intern. Med.* **34**, 1022–1029 (2019).
34. Bálint, L. *et al.* Epidemiologic characteristics of *Helicobacter pylori* infection in southeast Hungary. *World J Gastroenterol* **25**, 6365–6372 (2019).
35. Tamássy, K., Simon, L. & Francis, M. [Epidemiology of *Helicobacter pylori* infection in Hungary (comparative sero-epidemiologic study)]. *Orv Hetil* **136**, 1387–1391 (1995).
36. Iszlai, E. *et al.* [Seroprevalence of *Helicobacter pylori* infection and anti-CagA positivity in the county Szabolcs-Szatmár-Bereg]. *Orv Hetil* **144**, 1713–1718 (2003).
37. Prónai, L. & Tulassay, Z. A *Helicobacter pylori* eradikációjának sikertelensége: szempontok a további kezelés megítéléséhez. *Orvosi hetilap* **144**, 1299–1302 (2003).
38. Hirata, K. *et al.* Contribution of efflux pumps to clarithromycin resistance in *Helicobacter pylori*. *Journal of Gastroenterology and Hepatology* **25**, S75–S79 (2010).
39. Francesco, V. D. *et al.* Mechanisms of *Helicobacter pylori* antibiotic resistance: An updated appraisal. *World J Gastrointest Pathophysiol* **2**, 35–41 (2011).
40. Nishizawa, T. & Suzuki, H. Mechanisms of *Helicobacter pylori* antibiotic resistance and molecular testing. *Front Mol Biosci* **1**, (2014).
41. Fernandez-Gonzalez, E. & Backert, S. DNA transfer in the gastric pathogen *Helicobacter pylori*. *J. Gastroenterol.* **49**, 594–604 (2014).
42. Pot, R. G. J. *et al.* Interspecies Transfer of Antibiotic Resistance between *Helicobacter pylori* and *Helicobacter acinonychis*. *Antimicrob Agents Chemother* **45**, 2975–2976 (2001).
43. Bińkowska, A., Biernat, M. M., Łączmański, Ł. & Gościński, G. Molecular Patterns of Resistance Among *Helicobacter pylori* Strains in South-Western Poland. *Front Microbiol* **9**, (2018).

44. Trebesius, K. *et al.* Rapid and specific detection of *Helicobacter pylori* macrolide resistance in gastric tissue by fluorescent in situ hybridisation. *Gut* **46**, 608–614 (2000).
45. Mégraud, F. Epidemiology and mechanism of antibiotic resistance in *Helicobacter pylori*. *Gastroenterology* **115**, 1278–1282 (1998).
46. Marques, A. T., Vítor, J. M. B., Santos, A., Oleastro, M. & Vale, F. F. Trends in *Helicobacter pylori* resistance to clarithromycin: from phenotypic to genomic approaches. *Microbial Genomics*, **6**, e000344 (2020).
47. Chen, D. *et al.* Phenotypic and Molecular Antimicrobial Susceptibility of *Helicobacter pylori*. *Antimicrob Agents Chemother* **61**, (2017).
48. Rüssmann, H., Kempf, V. A. J., Koletzko, S., Heesemann, J. & Autenrieth, I. B. Comparison of Fluorescent In Situ Hybridization and Conventional Culturing for Detection of *Helicobacter pylori* in Gastric Biopsy Specimens. *J Clin Microbiol* **39**, 304–308 (2001).
49. Jüttner, S. *et al.* Reliable detection of macrolide-resistant *Helicobacter pylori* via fluorescence *in situ* hybridization in formalin-fixed tissue. *Modern Pathology* **17**, 684–689 (2004).
50. Demiray-Gürbüz, E. *et al.* Rapid identification of *Helicobacter pylori* and assessment of clarithromycin susceptibility from clinical specimens using FISH. *J Pathol Clin Res* **3**, 29–37 (2016).
51. Burucoa, C., Landron, C., Garnier, M. & Fauchère, J.-L. T2182C Mutation Is Not Associated with Clarithromycin Resistance in *Helicobacter pylori*. *Antimicrob Agents Chemother* **49**, 868–870 (2005).
52. De Francesco, V. *et al.* Phenotypic and genotypic *Helicobacter pylori* clarithromycin resistance and therapeutic outcome: benefits and limits. *J Antimicrob Chemother* **65**, 327–332 (2010).

REVIEWER COMMENTS

Reviewer #1 (Remarks to the Author):

The authors have been responsive to my concerns and improved the clinical significance of the manuscript significantly.

Comments

1. Reference 13: the journal name is American Journal of Gastroenterology
2. The manuscript would benefit from being edited for grammar and common usage by a native English speaker.

Reviewer #2 (Remarks to the Author):

The paper has changed substantially since the previous version so I have read the manuscript with fresh eyes providing a new overview below.

General

- It would be better to link to epidemiological language: incidence and prevalence rather than rates
- More inclusion of the uncertainty in the modelling predictions needs to be included in the main manuscript
- Needs some English corrections / proof reading

Abstract

- Cannot "transmit a resistant infection"
- I don't understand the sentence starting "An age-dependent female...". Is this now referring to non-macrolide naïve populations?
- There needs to be some results in the final sentence – not just that modelling can do this

Introduction

- Unclear: "rRNA mutate strains"
- Rewrite sentence beginning "Albeit the horizontal gene transfer"... of what?
- Unclear what the triple therapy is? (clar / amox + metro / PPI to me suggests a dual therapy)
Why is the triple therapy efficacy affected by cla-res if use amoxicillin instead of clarithromycin?
- Line 71: highest rank where?
- What will be the impact of these results? Are you suggesting that you might reduce clarithromycin as an eradication treatment? How do you stop transmission of H pylori?
- Is infection with HP lifelong? There is never any spontaneous clearance?

Results

- Line 172: "overinfection" needs explaining: has not been used in main text before
- Line 173: don't you need to remove prior infections then in the sum for the denominator? And make this just new infections?
- Line 169-180: Where is the uncertainty in your estimates?
- Line 177: Could you provide a figure to break the values in this paragraph down? Some kind of stacked bar chart? These are confusing percentage to combine. I think this would help to line up the 5.7% transmission in new vs 98.7 in just those macrolide naïve Cla-res infections.
- Line 182: 0.1% of what?
- Figure 3: I would not say that that is a significant reduction: 2% difference over 30 years? What is the denominator of the rate? All new infections?
- Figure 3: why doesn't the cla-res decrease with your estimated lower relative fitness? When you stop using clarithromycin for eradication?

Discussion

- You could refer here to the literature on bystander selection: Tedijanto et al for example from Marc Lipsitch's group. This is what you are describing for macrolide use for non-eradication purposes and will help the reader understand the differences in use and effect
- Line 212: 17.2% of all HP infections were with Cla-Res? Is this prevalence then not a "rate"?
- Line 220: the use of "primary" is confusing throughout this section – can you make it clearer? E.g. line 238: is this primary as in transmitted?
- Line 239: or that mutation rate is low? How big is the population size? Could be a small population => unlikely that mutation arises?
- Line 241: This is a very low relative fitness (see literature from Dan Andersson et al).
- Line 242: but this is in people who may have taken clarithromycin right? So combination here of resistance appearance / transmission and selection from drug exposure?
- Line 243: the one third seems very high: what is the prevalence of HP infection? i.e. what are the chances of someone with susceptible HP being infected by a transmitted resistant strain that then overcolonises the resident HP?
- Line 252: is this really significant? 2% change over 30 years? Significant to me would be a 50%+ change.
- Line 252: So what does this say about the impact of eradication purpose macrolide use? And how does this match the high fitness cost to resistance. Is this just a fitness cost to transmission? Shouldn't numbers with Cla-res drop substantially without bystander selection?
- Line 255: How can I be a good match to Taiwan if there is a "significant" reduction whilst Taiwan has stabilised? I think the language needs changing in this paragraph. I agree that a flat lining in your results does match a stable level in Taiwan but the value you reach is higher: did it drop in Taiwan to this level or, like you, just continue to increase at a slower rate?
- Line 324: An interesting model extension could be to use your model for an "all female" or "all male" population. Applying the antibiotic usage rates that vary by sex to match the resistance rate differences you see. With such an alternative model fit you could explore the generalisable nature of your output parameters (e.g. same estimate for relative fitness?).

Supplementary 4

- Conclusions: the ranges should be mentioned and discussed in the main text
- What is the implication of the relative fitness of the resistant strain going above 1? This would affect your future predictions right?
- Could you perhaps report 2 or 3 scenarios for the k value (onward transmission from hetero)? Why is 0.5 the base value?

Reviewer #3 (Remarks to the Author):

I was asked to provide input on the FISH sections. The authors have addressed my main concerns related to the FISH methodology and included the limitations of the methodology in their manuscript.

Nuno F. Azevedo

Reviewer #4 (Remarks to the Author):

The manuscript by Kocsmar et al. has benefitted from a tremendous amount of work, notably following the evaluation of the mathematical model by a specialist, and has been significantly improved.

Nevertheless the result for primary resistance, only 1.3% of cases occurring by spontaneous mutation versus 98.7% by transmission, should be tempered by 1) a limitation linked to the possibility of detection which underestimates the presence of bacteria with mutations in the stomach, given that a tiny part of the gastric mucosa is explored and 2) the long time period

necessary between the emergence of resistance in one bacterium and the spread of this clone in a proportion which is detectable, in the absence of selection pressure.

It would also be helpful to add a flow chart for primary resistance (numbers & %): Clari R/S - Previous macrolide therapy or not - Transmission vs mutation

Answers to the Referees

Reviewer #1 (Remarks to the Author):

The authors have been responsive to my concerns and improved the clinical significance of the manuscript significantly.

Comments

1. Reference 13: the journal name is American Journal of Gastroenterology

We thank the Reviewer for this note; we have modified the journal name accordingly.

2. The manuscript would benefit from being edited for grammar and common usage by a native English speaker.

We thank the Reviewer for this recommendation. Accordingly, we have sent the manuscript for professional proofreading.

Reviewer #2 (Remarks to the Author):

The paper has changed substantially since the previous version, so I have read the manuscript with fresh eyes, providing a new overview below.

General

- It would be better to link to epidemiological language: incidence and prevalence rather than rates

We thank the Reviewer for this remark. The phrases “rate of resistance” and “prevalence of resistance” are both widely used in the epidemiological literature to express the proportion of resistant infections among all infections (for example, ^{2,3}); hence, we use these terms interchangeably in our manuscript as well. Moreover, regarding the subgroups of a cohort, the terminology is also not consistent in the epidemiological literature, but “rate” is more likely to be used for describing the proportion of resistant infections in a given subgroup. In the description of the epidemiological model, we also use the word “rate” in the mathematical sense for some parameters (i.e., transmission rate). We have reviewed our references to the “rate” vs “prevalence” of resistance, and we hope our usage will not cause confusion.

- More inclusion of the uncertainty in the modelling predictions needs to be included in the main manuscript

We thank the Reviewer for this remark. Accordingly, the main text was completed with the uncertainty boundaries derived from the sensitivity analysis. The uncertainty of the modelling predictions was also integrated into Figure 4 (lines 149-166). In the main manuscript, we also inserted a more detailed description of the input parameters, clarifying their origin (retrospective cohort analysis, literature or inferred from the compartmental model) (lines 452-471).

- Needs some English corrections / proof reading

We thank the Reviewer for this recommendation. Accordingly, we have sent the manuscript for professional proofreading.

Abstract

- Cannot “transmit a resistant infection”

We thank the Reviewer for this note. We have changed this sentence accordingly:

“Cla-res was found in 5.5% of macrolide-naïve patients (primary Cla-res), with no significant sex difference, and was predicted by the model to originate from transmission of resistant bacteria in 98.7% of cases and to derive from spontaneous mutations in the other 1.3%.” (lines 32-34)

- I don’t understand the sentence starting “An age-dependent female...”. Is this now referring to non-macrolide naïve populations?

Yes, it is. We have modified the sentence for clarity:

“An age-dependent preponderance of females was found in secondary (macrolide-exposed) clarithromycin-resistant *H. pylori* infections, predominantly due to prior use of macrolides for non-eradication purposes.” (lines 34-36)

- There needs to be some results in the final sentence – not just that modelling can do this

We thank the reviewer for this recommendation. We have modified the final sentence accordingly:

“Mathematical modeling not only revealed the source of primary resistant cases but also predicted that the growth rate of Cla-res prevalence would decrease if macrolides were no longer used for purposes other than *H. pylori* eradication.” (lines 37-39)

Introduction

- Unclear: “rRNA mutate strains”

We thank the Reviewer for this recommendation. We modified the text as follows:

“Moreover, there is an efflux pump mechanism, but it is able to effectively excrete clarithromycin only in 23S rRNA mutant strains, suggesting that it is not an independent resistance system but acts synergistically with mutational resistance.” (lines 62-64)

- Rewrite sentence beginning “Albeit the horizontal gene transfer”... of what?

We thank the Reviewer for this recommendation. We modified this sentence as follows: “Albeit the horizontal gene transfer of **mutant genetic sequences** is also ...” (lines 64-65)

- Unclear what the triple therapy is? (clar / amox + metro / PPI to me suggests a dual therapy) Why is the triple therapy efficacy affected by cla-res if use amoxicillin instead of clarithromycin?

We thank the Reviewer for this note. The classic triple therapy has three components 1. Clarithromycin, 2. amoxicillin or metronidazole, and 3. A proton pump inhibitor (PPI). We have modified the sentence for clarity:

“The efficacy of the conventionally used empirical triple therapy (a combination of clarithromycin, amoxicillin (or metronidazole) and an oral proton pump inhibitor (PPI), administered without susceptibility testing) is being dramatically reduced by the increase in Cla-res.” (lines 71-73)

- Line 71: highest rank where?

In the WHO priority list of antibiotic-resistant bacteria and tuberculosis:

“Among bacteria typically responsible for community-acquired infections, the highest ranked were clarithromycin-resistant *Helicobacter pylori* at 44.8% (10.1) and fluoroquinolone-resistant *Campylobacter* spp at 41.0% (7.8), *Neisseria gonorrhoeae* at 35.8% (8.9), and *Salmonella typhi* at 37.6% (9.2).”

(Tacconelli E, Carrara E, Savoldi A, et al. Discovery, research, and development of new antibiotics: the WHO priority list of antibiotic-resistant bacteria and tuberculosis. *Lancet Infect Dis.* 2018;18(3):318-327. doi:10.1016/S1473-3099(17)30753-3)

This priority list described by Tacconelli and her colleagues is an international, global effort to prioritize research and development of new antibiotics according to bacterial drug resistance.

- What will be the impact of these results? Are you suggesting that you might reduce clarithromycin as an eradication treatment? How do you stop transmission of H pylori?

It is not completely clear which sentence of the Introduction is the object of this question. Therefore, we will answer generally:

The increasing prevalence of Cla resistance has led to a decreasing efficacy of Cla-containing eradication treatments worldwide. Although other antibiotics are also used for eradication therapy of *H. pylori* infection, clarithromycin, owing to its antimicrobial effectiveness and infrequent side effects, remains the recommended drug for first-line eradication therapy of *H. pylori* infection in the leading international guidelines^{4,5}. Against this background, reduction of clarithromycin use would be recommended in non-*H. pylori* eradication-motivated antibiotic treatments to decrease population-level resistance. By continuing the successful eradication strategy, we might reduce the prevalence of *H. pylori* infections at the

population level. This would also reduce the transmission of resistant bacteria (less *H. pylori* infection will result in less clarithromycin-resistant infection and, consequently, in less transmission of resistant *H. pylori* strains) and would therefore limit the increase in the prevalence of clarithromycin resistance. Finally, this would contribute to reducing the overall prevalence of *H. pylori* infection and associated diseases.

- Is infection with HP lifelong? There is never any spontaneous clearance?

The Reviewer gave us the following recommendation during the first round of review for our manuscript:

“It should also be mentioned that *H. pylori* infection usually persists for life, and that individuals can only clear the infection thanks to eradication treatment (Kusters, J. G., van Vliet, A. H., & Kuipers, E. J. (2006). Pathogenesis of *Helicobacter pylori* infection). This context is important to understand why natural clearance of the bacteria is not part of the mathematical model.” We completely agree with this statement, since spontaneous clearance is believed to be extremely rare; in other words, it is practically a nonexistent outcome of the natural course of *H. pylori* infection.

Results

- Line 172: “overinfection” needs explaining: has not been used in main text before

We thank the Reviewer for this remark. Accordingly, we have modified this sentence to avoid using the term “overinfection”.

“transmission of resistant strains is responsible for only approximately 5.7% (4.67-7.43%) of all new infections (**incidence, including all transitions from uninfected to infected compartments, whether infected with susceptible or resistant strains**)” (lines 152-154)

- Line 173: don't you need to remove prior infections then in the sum for the denominator? And make this just new infections?

We thank the Reviewer for this remark. The denominator included only transitions from uninfected to infected compartments (new infections), independent of the resistance status of the infections. Accordingly, we have clarified the text in brackets.

- Line 169-180: Where is the uncertainty in your estimates?

We thank the Reviewer for this note. We have inserted the uncertainty boundaries of the estimates into the text and into Figure 4 (former Figure 3).

- Line 177: Could you provide a figure to break the values in this paragraph down? Some kind of stacked bar chart? These are confusing percentage to combine. I think this would help to line up the 5.7% transmission in new vs 98.7 in just those macrolide naïve Cla-res infections.

We thank the Reviewer for this note. We have inserted a new figure (Figure 3) in the manuscript to clarify these different proportions.

- Line 182: 0.1% of what?

This value refers to the annual increase in Cla resistance in the next 30-year period starting from the current value (17.2%). This is also seen in Figure 4 (former Figure 3). We have changed this sentence accordingly.

“we predict an increase of approximately 0.1% (0.06-0.14%) per year in the rate of clarithromycin resistance in the future...” (lines 163-164)

- Figure 3: I would not say that that is a significant reduction: 2% difference over 30 years? What is the denominator of the rate? All new infections?

We thank the Reviewer for this note. In the figure legend, “reduce” refers to the reduction in the rate of increase in prevalence and not the prevalence of Cla-resistant infections in general. We have changed this sentence to avoid misunderstandings.

The denominator of this value is all *H. pylori* infections at a given time. According to the results, the current prevalence of Cla resistance among all *H. pylori*-infected individuals (17.2%) will increase by 0.1% every year. In line with this upward trend, the proportion of Cla-resistant infections in the *H. pylori*-infected subpopulation will be close to 20% after 30 years.

The term “significant” refers to the clinical significance of this value rather than statistical significance. In the leading guidelines for Europe and North America, 15% of population-level clarithromycin resistance is the breakpoint value for the use of clarithromycin as an empirical first-line drug in the treatment of *H. pylori* infections.^{4,5} In line with this, if non-eradication-purpose macrolide use were discontinued, population-level Cla resistance could remain under this threshold in several populations; therefore, Cla-containing eradication regimens would be applicable in clinical practice.

Accordingly, we have modified the questioned sentence as follows:

“The model shows a noticeably slower long-term growth rate of Cla-res prevalence among infected individuals with the discontinuation of non-eradication-purpose macrolide use (green line) ...”

(lines 664-665)

- Figure 3: why doesn't the cla-res decrease with your estimated lower relative fitness? When you stop using clarithromycin for eradication?

We note that we did not model the scenario in which clarithromycin is no longer used for eradication; instead, as displayed in (former) Figure 3 (now Figure 4), we show the predicted result of discontinuing macrolide use for purposes other than eradication. Our advice is that clarithromycin should be used exclusively for *H. pylori* eradication and, if possible, replaced with other macrolides or nonmacrolide antibiotics for other indications.

Regarding the trends displayed in (former) Figure 3 (now Figure 4), clarithromycin resistance will not drop substantially if macrolides cease to be used for non-eradication purposes. By way of explaining this prediction, it is important to note that *H. pylori* infection usually persists for life (as we have already discussed above); consequently, the existing clarithromycin-resistant cases remain resistant for a long time, and the cumulative effect of transmission of the resistant strains (which can generate new resistant cases not only by infection of previously uninfected persons but also by superinfection of individuals carrying susceptible strains), spontaneous mutation-related primary resistance and unsuccessful eradication attempts with clarithromycin-containing regimens will slightly increase the prevalence of clarithromycin resistance. Our model predicts a relative transmission fitness of 0.72, which does not seem to be extremely low; for example, in the publication of Shrestha *et al.* (Drivers and Trajectories of Resistance to New First-Line Drug Regimens for Tuberculosis)⁶ the authors predicted relative transmission fitness values of 0.4-0.6 for drug-resistant tuberculosis. In the current situation, secondary resistance related to macrolide use contributes more to the increasing prevalence of clarithromycin

resistance than the transmission of resistant strains. However, if this macrolide-use-related source were eliminated, the hosts of Cla-res *H. pylori* infections might still contribute substantially to the spread of clarithromycin resistance even with this 0.72 value of relative transmission fitness. Consequently, the overall result of the effects discussed above is a slower but still increasing trajectory of the prevalence of clarithromycin resistance in the absence of non-eradication-purpose macrolide use, as displayed in Figure 3.

If clarithromycin-containing eradication treatments were discontinued, the slowdown of the current increasing trend might be slight. Specifically, clarithromycin-containing eradication therapy poses the highest selection advantage for resistant bacteria; therefore, an unsuccessful eradication attempt represents the highest risk of acquiring clarithromycin resistance (51.4% of the remaining infections were resistant in our cohort). Both non-eradication use of clarithromycin and treatment with other macrolides led to clarithromycin resistance less frequently (36.2% and 13.4%, respectively) in our cohort. However, clarithromycin-containing regimens are able to eradicate the vast majority of *H. pylori* infections, while the unintended eradication rate (eradication of *H. pylori* by non-eradication-purpose macrolide use, whether clarithromycin monotherapy or other macrolide treatment) is considered to be very low⁷⁻⁹. Although the rate of posttreatment clarithromycin resistance related to non-eradication macrolide use is lower, currently, these antibiotic treatments are much more frequently used in the population than clarithromycin-containing eradication therapies. Consequently, the absolute number of these cases (and therefore the population prevalence) is considerably higher than the rate of eradication-related posttreatment resistance. Therefore, it is not worthwhile to stop the clarithromycin-containing eradication treatments, as the benefit to clarithromycin resistance prevention would be subtle, and susceptibility-guided clarithromycin-containing conventional triple therapy can achieve eradication rates just as good as or even better than those of other nontargeted anti-*H. pylori* regimens, but with a better side-effect profile⁷.

Discussion

- You could refer here to the literature on bystander selection: Tedijanto et al for example from Marc Lipsitch's group. This is what you are describing for macrolide use for non-eradication purposes and will help the reader understand the differences in use and effect

We thank the Reviewer for suggesting that we include an explanation of bystander selection in the Discussion. Accordingly, we have inserted a sentence on this topic, with a reference to Chris Kenyon's publication entitled "Population-level macrolide consumption is associated with clarithromycin resistance in *Helicobacter pylori*: An ecological analysis", in which the author discusses bystander selection in terms of the correlation between the clarithromycin resistance of *H. pylori* and population-level macrolide consumption:

"This bystander selection (unintended selection of resistant *H. pylori* by an antibiotic therapy targeting another bacterial species) has been suggested to play an important role in the maintenance of the population level of Cla-res prevalence of *H. pylori*, including both primary and secondary resistance according to the above definition."¹⁰

(lines 176-179)

- Line 212: 17.2% of all HP infections were with Cla-Res? Is this prevalence then not a "rate"?

We thank the Reviewer for this remark. As we have discussed above, the phrases "rate of resistance" and "prevalence of resistance" are both widely used in the epidemiological literature to express the proportion of resistant infections among all infections (for example,^{2,3}); hence, we use these terms interchangeable in

our manuscript as well. The 17.2% value refers to all *H. pylori* infections. For clarification, we also refer to Table 2, from which the meaning of 17.2% becomes clear, and we changed the term “rate” to “prevalence”.

- Line 220: the use of “primary” is confusing throughout this section – can you make it clearer? E.g. line 238: is this primary as in transmitted?

This might be confusing due to the common (and, in our opinion, inappropriate) use of the term “primary” in the literature. Specifically, this term is commonly used for clarithromycin-resistant *H. pylori* infections in patients who have never received previous clarithromycin-containing eradication therapy, but prior use of macrolides for other purposes is not taken into account. However, we have clearly described in line 220 that we consider macrolide-naïve but Cla-res patients to be the real primary resistant cases. Therefore, we do not feel that further clarification is needed. However, for clarity, we have modified the questioned sentence by inserting an explanatory phrase in parentheses:

“... we concluded that only 1.3% of primary (macrolide-naïve) Cla-res infections developed from spontaneous mutations.” (lines 215-216)

- Line 239: or that mutation rate is low? How big is the population size? Could be a small population => unlikely that mutation arises?

The mutation rate of *H. pylori* is considered to be relatively high among bacteria, as stated by several publications^{11,12}. In general, however, clarithromycin resistance has a biological cost, since the protein synthesis of mutant bacteria is less effective than that of wild-type bacteria.¹¹ This is the reason why the newly appearing mutant bacteria will usually be selected out. There are compensatory mutations that can balance out the biological cost of antibiotic resistance by at least partly restoring the effectiveness of bacterial protein synthesis.^{11,13} However, these are usually not present at the initial time when the clarithromycin resistance mutation appears, contributing to the fast disappearance of the clarithromycin-resistant mutants except when macrolide antibiotics are present. Regarding the question of population size, our cohort contains 140 macrolide-naïve primary resistance cases, which is not a small population for this investigation. The size of the bacterial population of the stomach in chronic *H. pylori* gastritis is also discussed in Supplementary Material 4 and estimated to vary between 1.118×10^9 and 6.237×10^{12} *H. pylori* bacteria. On this basis, the bacterial population size is large enough to develop a high number of clarithromycin-resistant mutant bacteria. Therefore, neither the population size nor the spontaneous mutation rate is too low, but the long-term survival of the new resistant clones is a rare event, except when the presence of a macrolide antibiotic supports it.

- Line 241: This is a very low relative fitness (see literature from Dan Andersson et al).

We thank the Reviewer for this note. It is important to emphasize that the “bacterial fitness” (ability of bacteria to replicate in a given environment) and the “transmission fitness” (ability to infect other individuals) are different entities. What predicted was not bacterial fitness but relative transmission fitness, for which a value of 0.72 does not seem to be extremely low; for example, a publication by Shrestha et al.⁶ predicted relative transmission fitness values of 0.4-0.6 for drug-resistant tuberculosis.

- Line 242: but this is in people who may have taken clarithromycin right? So combination here of resistance appearance / transmission and selection from drug exposure?

This 5.7% is the proportion of resistant transmission among new infections independent of prior clarithromycin use; therefore, this includes patients with and without prior clarithromycin treatment. Of these, patients who had never taken clarithromycin or other macrolides corresponded to two-thirds of the

transmitted primary resistance cases that were derived from resistant transmission and infected previously uninfected people (and these transmitted primary resistance cases account for 98.7% of all primary clarithromycin resistant cases). However, it is important to note that 5.7% is an incidence, while 98.7% is a prevalence. We also refer to our new Figure 3.

- Line 243: the one third seems very high: what is the prevalence of HP infection? i.e. what are the chances of someone with susceptible HP being infected by a transmitted resistant strain that then overcolonises the resident HP?

We thank the Reviewer for this question. According to the literature, the prevalence of *H. pylori* infection in Central Hungary was assessed to be 40.9%¹⁴ [see Supplementary Material 1]. Considering that the proportion of cases with clarithromycin-resistant infection was 17.2% in our cohort, this accounts for 7% of the overall prevalence. Therefore, for a given case of resistant transmission, there is approximately a 59% chance that the recipient had no preexisting infection, a 34% chance that susceptible *H. pylori* are already present, and a 7% chance that a patient with one resistant infection is superinfected with another. Regarding the question, this latter should be left out of the consideration, and the ratio of 59% to 34% is close to the predicted two-thirds uninfected and one-third infected targets of resistant transmission among primary (macrolide-naïve) resistant cases.

- Line 252: is this really significant? 2% change over 30 years? Significant to me would be a 50%+ change.

This question is addressed above. We have modified this sentence in the Discussion to avoid the use of the term “significant”. (lines 228-232)

- Line 252: So what does this say about the impact of eradication purpose macrolide use? And how does this match the high fitness cost to resistance. Is this just a fitness cost to transmission? Shouldn't numbers with Cla-res drop substantially without bystander selection?

We thank the Reviewer for this question. As we have already partly addressed this question in connection with our former Figure 3 (now Figure 4), it is important to emphasize that we modeled the situation where macrolide use for non-eradication purposes is discontinued. If clarithromycin-containing eradication treatments are stopped, the slowdown of the current increasing trend might be slight. Specifically, clarithromycin-containing eradication therapy provides the highest selective advantage for resistant bacteria; therefore, an unsuccessful eradication attempt represents the highest risk of acquiring clarithromycin resistance (51.4% of the remaining infections were resistant in our cohort). Both non-eradication use of clarithromycin and treatment with other macrolides led to clarithromycin resistance less frequently (36.2% and 13.4%, respectively) in our cohort. However, clarithromycin-containing regimens are able to eradicate the vast majority of *H. pylori* infections, while the unintended eradication rate (eradication of *H. pylori* by non-eradication-purpose macrolide use, either by clarithromycin monotherapy or by other macrolide treatment) is considered to be very low^{7,8}. Although the rate of posttreatment clarithromycin resistance related to non-eradication macrolide use is lower, these antibiotic treatments are currently much more frequently used in everyday medical practice than clarithromycin-containing eradication therapies. Consequently, the absolute number of these cases (population prevalence) is considerably higher than the number with eradication-related posttreatment resistance. Therefore, the use of macrolides for eradication plays a much smaller part in maintaining the population level of clarithromycin resistance.

Regarding the mentioned “high fitness cost”, we also addressed this issue above, and we would like to politely note that our model predicted the fitness cost of the transmission (ability to infect other

individuals) instead of the “bacterial fitness” (ability of bacteria to replicate in a given environment). Our model predicts a relative transmission fitness of 0.72. Regarding line 241, the Reviewer noted that “This is a very low relative fitness (see literature from Dan Andersson et al).” However, in the publication by Andersson et al.¹³, the authors show examples of bacterial fitness for which the 0.72 value would be very low in comparison with the displayed mutational effects. However, we predicted not the bacterial fitness but the relative transmission fitness, for which a value of 0.72 does not seem to be extremely low; for example, a publication by Shrestha *et al.* (Drivers and Trajectories of Resistance to New First-Line Drug Regimens for Tuberculosis)⁶ predicted relative transmission fitness values of 0.4-0.6 for drug-resistant tuberculosis. Therefore, we are convinced that this 0.72 relative transmission fitness does not imply an extremely “high fitness cost”.

Regarding the scenario of stopping bystander selection, this is what we modeled and displayed in (former) Figure 3 (now Figure 4). In this situation, clarithromycin resistance will not drop substantially after the discontinuation of non-eradication macrolide use. In order to explain this, it is important to note that *H. pylori* infection usually persists for life (as we have already discussed above); consequently, the existing clarithromycin-resistant cases remain resistant in the long term, and the cumulative effect of transmission of the resistant strains (which can generate new resistant cases not only by infecting previously uninfected persons but also by superinfection of individuals carrying susceptible strains), spontaneous-mutation-related primary resistance and unsuccessful eradication attempts with clarithromycin-containing regimens will further increase the prevalence of clarithromycin resistance. In the current situation, macrolide-related secondary resistance contributes more to the increasing prevalence of clarithromycin resistance than the transmission of resistant strains. However, if this macrolide-use-related mechanism were eliminated, the hosts of the Cla-res *H. pylori* infections might still contribute substantially to the spread of clarithromycin resistance even with this 0.72 value of relative transmission fitness. Consequently, the overall result of the above-discussed effects is a slower but still increasing trajectory of the prevalence of clarithromycin resistance in the absence of non-eradication-purpose macrolide use, as displayed in Figure 4 (former Figure 3).

- Line 255: How can I be a good match to Taiwan if there is a “significant” reduction whilst Taiwan has stabilised? I think the language needs changing in this paragraph. I agree that a flat lining in your results does match a stable level in Taiwan but the value you reach is higher: did it drop in Taiwan to this level or, like you, just continue to increase at a slower rate?

In Taiwan, consumption of macrolide antibiotics substantially reduced (1.55→0.49 DDD /1000 persons /day) in the 2000-2006 period, and the rate of clarithromycin resistance did not drop considerably but rather showed a slight increasing trend from 2006 to period, as shown in Fig 1 in the publication of Liou JM et al.¹⁵ This is in close agreement with our prediction displayed in Figure 4 (former Figure 3).

We agree with the Reviewer that some changes in the language could help clarify the legend of this figure. The “significant reduction” was used in the context of the increasing trend and not the population prevalence. Therefore, we have changed the legend and the related part of the main text accordingly.

(lines 228-246)

- Line 324: An interesting model extension could be to use your model for an “all female” or “all male” population. Applying the antibiotic usage rates that vary by sex to match the resistance rate differences you see. With such an alternative model fit you could explore the generalisable nature of your output parameters (e.g. same estimate for relative fitness?).

Thank you for this interesting remark. A transmission model with “all female” or “all male” populations would leave out a significant portion of transmissions (male to female etc.), and the resulting estimates of parameters could be misleading. For this reason, we decided to simulate scenarios where the whole population uses macrolides at a calculated female consumption rate and compared it with the male consumption rate. The results are depicted in Supplementary Material 2 Figure 2, supporting the idea that a higher consumption rate will result in a higher rate of resistance, aligned with the observed variation by sex.

Supplementary 4

- Conclusions: the ranges should be mentioned and discussed in the main text

We thank the Reviewer for this remark. During the revision process, we have inserted the ranges into the relevant parts of the manuscript.

- What is the implication of the relative fitness of the resistant strain going above 1? This would affect your future predictions right? Could you perhaps report 2 or 3 scenarios for the k value (onward transmission from hetero)? Why is 0.5 the base value?

The fitness of the resistant strain exceeds 1 only in an extreme situation (when kappa is near zero). One can think of kappa as relative bacterial shedding of the susceptible/resistant types by a heteroresistant infected host (we clarified this also in the text of Supplementary Material 2). Since we could not observe a significant difference in the numbers of susceptible vs resistant bacteria in our samples, we assumed that 0.5 as a baseline value. However, we simulated scenarios with varying kappa values and plotted the cases kappa=0.3 and kappa= 0.7 in S2 (Figure 3). The onward transmission is determined by both kappa (shedding) and delta (relative transmission fitness, i.e., ability to infect a new host), as seen from the formula for the force of infection.

Reviewer #3 (Remarks to the Author):

I was asked to provide input on the FISH sections. The authors have addressed my main concerns related to the FISH methodology and included the limitations of the methodology in their manuscript.

Nuno F. Azevedo

We thank Dr. Nuno F. Azevedo for taking time to review our paper and making very helpful suggestions.

Reviewer #4 (Remarks to the Author):

The manuscript by Kocsmar et al. has benefitted from a tremendous amount of work, notably following the evaluation of the mathematical model by a specialist, and has been significantly improved.

Nevertheless the result for primary resistance, only 1.3% of cases occurring by spontaneous mutation versus 98.7% by transmission, should be tempered by 1) a limitation linked to the possibility of detection which underestimates the presence of bacteria with mutations in the stomach, given that a tiny part of the gastric mucosa is explored and 2) the long time period necessary between the emergence of resistance in one bacterium and the spread of this clone in a proportion which is detectable, in the absence of selection pressure.

It would also be helpful to add a flow chart for primary resistance (numbers & %): Clari R/S - Previous macrolide therapy or not - Transmission vs mutation

We thank the Reviewer for the helpful comments. According to the Reviewer's recommendation, we have inserted these statements into the limitations section of the main text. (lines 297-305) Additionally, we have added a flowchart to the main text (Figure 3/a.) to show the exact numbers and percentages of resistant cases based on the cohort and modeling data.

References

1. Hooi, J. K. Y. *et al.* Global Prevalence of Helicobacter pylori Infection: Systematic Review and Meta-Analysis. *Gastroenterology* **153**, 420–429 (2017).
2. Mégraud, F. Epidemiology and mechanism of antibiotic resistance in Helicobacter pylori. *Gastroenterology* **115**, 1278–1282 (1998).
3. Gonzalez-Hormazabal, P. *et al.* Prevalence of clarithromycin resistance in Helicobacter pylori in Santiago, Chile, estimated by real-time PCR directly from gastric mucosa. *BMC Gastroenterol* **18**, 91 (2018).
4. Chey, W. D., Leontiadis, G. I., Howden, C. W. & Moss, S. F. ACG Clinical Guideline: Treatment of Helicobacter pylori Infection. *Am. J. Gastroenterol.* **112**, 212–239 (2017).
5. Malfertheiner, P. *et al.* Management of Helicobacter pylori infection-the Maastricht V/Florence Consensus Report. *Gut* **66**, 6–30 (2017).
6. Shrestha, S. *et al.* Drivers and Trajectories of Resistance to New First-Line Drug Regimens for Tuberculosis. *Open Forum Infect Dis* **1**, (2014).
7. O'Connor, A., Liou, J.-M., Gisbert, J. P. & O'Morain, C. Review: Treatment of Helicobacter pylori Infection 2019. *Helicobacter* **24 Suppl 1**, e12640 (2019).
8. Hiyama, T. *et al.* Rate of unintended Helicobacter pylori eradication in the Vietnamese. *Helicobacter* **20**, 156–157 (2015).
9. Ono, S. *et al.* Frequency of Helicobacter pylori -negative gastric cancer and gastric mucosal atrophy in a Japanese endoscopic submucosal dissection series including histological, endoscopic and serological atrophy. *Digestion* **86**, 59–65 (2012).

10. Kenyon, C. Population-level macrolide consumption is associated with clarithromycin resistance in *Helicobacter pylori*: An ecological analysis. *International Journal of Infectious Diseases* **85**, 67–69 (2019).
11. Björkholm, B. *et al.* Mutation frequency and biological cost of antibiotic resistance in *Helicobacter pylori*. *Proc Natl Acad Sci U S A* **98**, 14607–14612 (2001).
12. Linz, B. *et al.* A mutation burst during the acute phase of *Helicobacter pylori* infection in humans and rhesus macaques. *Nat Commun* **5**, 4165 (2014).
13. Andersson, D. I. & Hughes, D. Antibiotic resistance and its cost: is it possible to reverse resistance? *Nature Reviews Microbiology* **8**, 260–271 (2010).
14. Buzás, G. M., Lotz, G., Schneider, F. & Józán, J. [Changing prevalence of *Helicobacter pylori* infection in the 9th district of Budapest. A retrospective endoscopic study, 1997-2012]. *Orv Hetil* **154**, 900–907 (2013).
15. Liou, J.-M. *et al.* The Primary Resistance of *Helicobacter pylori* in Taiwan after the National Policy to Restrict Antibiotic Consumption and Its Relation to Virulence Factors-A Nationwide Study. *PLoS ONE* **10**, e0124199 (2015).

REVIEWERS' COMMENTS

Reviewer #2 (Remarks to the Author):

The authors have responded well to my comments.

I like the new Figure 3. My only comment on it would be that I'm not entirely clear that "Eradicated resistant cases" is the correct phrase to use in part C.

Reviewer #2 (Remarks to the Author):

The authors have responded well to my comments.

I like the new Figure 3. My only comment on it would be that I'm not entirely clear that "Eradicated resistant cases" is the correct phrase to use in part C.

We thank the Reviewer for the positive response. We agree with the Reviewer's comment. Accordingly, we have modified the legend on Figure 3 to "Resistant cases after eradication failure". The same phrase appeared in the guide of Figure 2, we have changed that one as well.